# Mutation bias reflects natural selection in *Arabidopsis thaliana*

J. Grey Monroe[1,2 ✉], Thanvi Srikant[1], Pablo Carbonell-Bejerano[1], Claude Becker[1,10], Mariele Lensink[2], Moises Exposito-Alonso[3,4], Marie Klein[1,2], Julia Hildebrandt[1], Manuela Neumann[1], Daniel Kliebenstein[2], Mao-Lun Weng[5], Eric Imbert[6], Jon Ågren[7], Matthew T. Rutter[8], Charles B. Fenster[9] & Detlef Weigel[1 ✉]

Since the first half of the twentieth century, evolutionary theory has been dominated by the idea that mutations occur randomly with respect to their consequences[1]. Here we test this assumption with large surveys of de novo mutations in the plant *Arabidopsis thaliana*. In contrast to expectations, we find that mutations occur less often in functionally constrained regions of the genome—mutation frequency is reduced by half inside gene bodies and by two-thirds in essential genes. With independent genomic mutation datasets, including from the largest *Arabidopsis* mutation accumulation experiment conducted to date, we demonstrate that epigenomic and physical features explain over 90% of variance in the genome-wide pattern of mutation bias surrounding genes. Observed mutation frequencies around genes in turn accurately predict patterns of genetic polymorphisms in natural *Arabidopsis* accessions ($r = 0.96$). That mutation bias is the primary force behind patterns of sequence evolution around genes in natural accessions is supported by analyses of allele frequencies. Finally, we find that genes subject to stronger purifying selection have a lower mutation rate. We conclude that epigenome-associated mutation bias[2] reduces the occurrence of deleterious mutations in *Arabidopsis*, challenging the prevailing paradigm that mutation is a directionless force in evolution.

The random occurrence of mutations with respect to their consequences is an axiom upon which much of biology and evolutionary theory rests[1]. This simple proposition has had profound effects on models of evolution developed since the modern synthesis, shaping how biologists have thought about and studied genetic diversity over the past century. From this view, for example, the common observation that genetic variants are found less often in functionally constrained regions of the genome is believed to be due solely to selection after random mutation. This paradigm has been defended with both theoretical and practical arguments: that selection on gene-level mutation rates cannot overcome genetic drift; that previous evidence of non-random mutational patterns relied on analyses in natural populations that were confounded by the effects of natural selection; and that past proposals of adaptive mutation bias have not been framed in the context of potential mechanisms that could underpin such non-random mutations[3–6].

Yet, emerging discoveries in genome biology inspire a reconsideration of classical views. It is now known that nucleotide composition, epigenomic features and bias in DNA repair can influence the likelihood that mutations occur at different places across the genome[7–13]. At the same time, we have learned that specific gene regions and broad classes of genes, including constitutively expressed and essential housekeeping genes, can exist in distinct epigenomic states[14]. This could in turn provide opportunities for adaptive mutation biases to evolve by coupling DNA repair with features enriched in constrained loci[2]. Indeed, evidence that DNA repair is targeted to genic regions and active genes has been found[15–20]. Here we synthesize these ideas by investigating the causes, consequences and adaptive value of mutation bias in the plant *Arabidopsis thaliana*.

## De novo mutations in *Arabidopsis*

The greatest barrier to investigating gene-level mutation variability has been a lack of data characterizing new mutations before they experience natural selection. We addressed this limitation by compiling large sets of de novo mutations in *A. thaliana* (hereafter referred to as *Arabidopsis*), for which there is rich information on sequence and epigenomic features plausibly linked to mutation rates. We first reanalysed existing *Arabidopsis* mutation accumulation lines[12], combining putative germline and somatic mutations (Fig. 1a, Extended Data Figs. 1, 2, Supplementary Data 1; Methods). A filtering pipeline to eliminate false positives and based on mapping quality, depth and variant frequency retained less than 10% of called variants in a final high-confidence set of mutations. We found no evidence of selection on these mutations. The germline mutations had accumulated

[1]Department of Molecular Biology, Max Planck Institute for Biology Tübingen, Tübingen, Germany. [2]Department of Plant Sciences, University of California Davis, Davis, CA, USA. [3]Department of Plant Biology, Carnegie Institution for Science, Stanford, CA, USA. [4]Department of Biology, Stanford University, Stanford, CA, USA. [5]Department of Biology, Westfield State University, Westfield, MA, USA. [6]ISEM, University of Montpellier, Montpellier, France. [7]Department of Ecology and Genetics, EBC, Uppsala University, Uppsala, Sweden. [8]Department of Biology, College of Charleston, Charleston, SC, USA. [9]Oak Lake Field Station, South Dakota State University, Brookings, SD, USA. [10]Present address: Faculty of Biology, Ludwig Maximilian University, Martinsried, Germany. ✉e-mail: gmonroe@ucdavis.edu; weigel@weigelworld.org

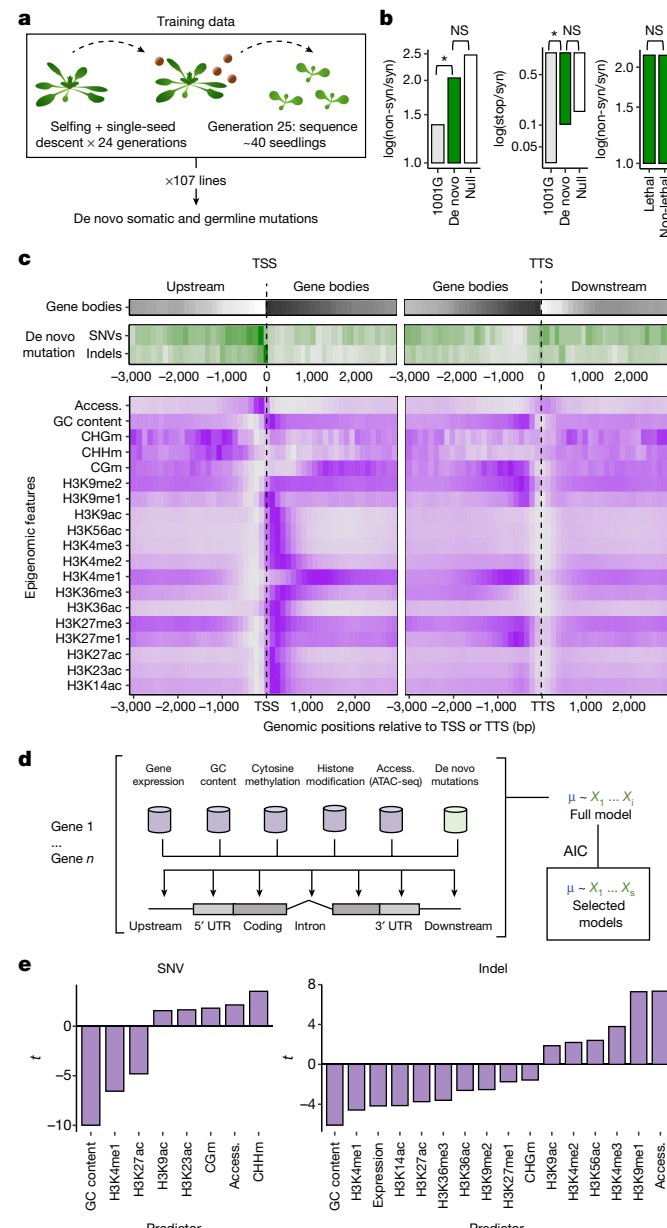

**Fig. 1 | Identifying epigenomic and other features associated with mutations in *Arabidopsis*. a**, Experimental design for identifying germline and somatic mutations in the main dataset[12]. **b**, Relaxed purifying selection in de novo mutation calls: rates of non-synonymous (non-syn) and stop codon variants (stop) as compared with polymorphisms detected in 1,135 natural accessions from the 1001 Genomes (1001G) project[35] and to a null model based on mutation spectra and nucleotide composition of coding sequences. Comparison of de novo mutations between genes predicted to have or not have lethal effects when mutated is also shown[37]. *P* values from $\chi^2$ test; *$P$ < 0.05. NS, not significant. **c**, Genome-wide distributions in gene body density, observed mutation rates and candidate predictive features in relation to transcription start sites (TSS) and transcription termination sites (TTS). Darker shading represents greater density. SNV, single-nucleotide variant; CHGm, CHHm, CGm, methylation in the CHG, CHH and CG contexts, respectively. **d**, Modelling approach to predict mutation probability from a range of features. ATAC-seq, assay for transposase-accessible chromatin using sequencing; AIC, Akaike information criterion. **e**, Predictive models and *t*-values of predictor variables from the generalized linear model.

in randomly chosen single-seed descendants, so very few mutations, only those causing inviability or sterility, should have been removed by selection[12]. Somatic mutations experience even less selection[21,22].

Therefore, as expected, non-synonymous changes and premature stop codons accounted for a greater share of variants than in natural populations, and their frequencies were indistinguishable from a null model of random mutation. We also confirmed that there was no bias in detecting non-synonymous mutations when comparing genes predicted to be sensitive or insensitive to mutation (Fig. 1b).

## Epigenome-mediated mutation bias

We tested whether the location of mutations in our dataset was associated with epigenomic features, focusing on biochemical properties previously linked to mutation: gene expression, GC content, cytosine methylation, histone modifications and chromatin accessibility (Fig. 1c). We built linear models of mutation frequencies in genic regions as a function of these features across the genome (Fig. 1d; Methods).

These models revealed features positively and negatively associated with mutations, with several having been already linked to mutagenesis or DNA repair (Fig. 1e). For example, the negative relationship between GC content and mutation[23] is consistent with GC-biased gene conversion[24] and reduced DNA denaturation in GC-rich regions[25]. Likewise, previous work has linked H3K4me1 to genome stability, DNA repair and lower mutation rates[26–31]. By contrast, methylated cytosines correlate with elevated mutation rate, consistent with the effects of cytosine deamination[12,32], while highly accessible chromatin regions (for example, transcription factor-binding sites) can impair nucleotide excision repair[33]. In conclusion, we uncovered associations between mutation frequencies and biochemical features known to affect DNA repair and vulnerability to damage.

We note in advance here that all downstream analyses led to the same conclusions for single-nucleotide variants (SNVs) and insertions and deletions (indels), or for germline and somatic mutations. All were less frequent in gene bodies and essential genes, and we therefore report combined results. Our conclusions also did not change when we repeated the analyses after training our initial epigenome prediction model on non-coding regions only. Finally, we confirmed that observed mutation biases could not be explained by variation in read depth, mappability, the distribution of false positives or selection on mutations (Extended Data Fig. 3).

## Lower mutation rate in gene bodies

We calculated predicted mutation probabilities (predicted mutations per base pair) as a function of epigenomic features around genes and found that mutation rates were lower within gene bodies (Fig. 2a). These predictions were confirmed by observed mutations in multiple independent datasets (Fig. 2b, Supplementary Data 1). We called mutations in new *Arabidopsis* mutation accumulation populations, the largest reported to date: germline and somatic mutations in 400 lines established from eight genetically diverse founder genotypes, four each from the extreme North and South of Europe. Observed distributions of germline and somatic mutations were very similar to epigenome-predicted mutation rates. These data also provided evidence for genetic variation in mutation bias, raising the possibility of mutation bias evolvability (Extended Data Fig. 4). Somatic variants identified from 10 rosettes and from reanalysing deep sequencing data of 64 leaves in two *Arabidopsis* plants[21] further confirmed predicted patterns, as did previously discovered germline mutations in a bottlenecked *Arabidopsis* lineage[32] (Extended Data Figs. 3, 4).

By combining mutation datasets, we found that the frequency of mutation was 58% lower in gene bodies than in nearby intergenic space. Epigenome-predicted mutation probabilities explained over 90% of the variance in the pattern of observed mutations around gene bodies (Fig. 2a, b, Extended Data Fig. 5). Since only 20–30% of gene body sites are estimated to be subject to selection, mutation bias in genic regions could affect sequence evolution around genes more than selection[34].

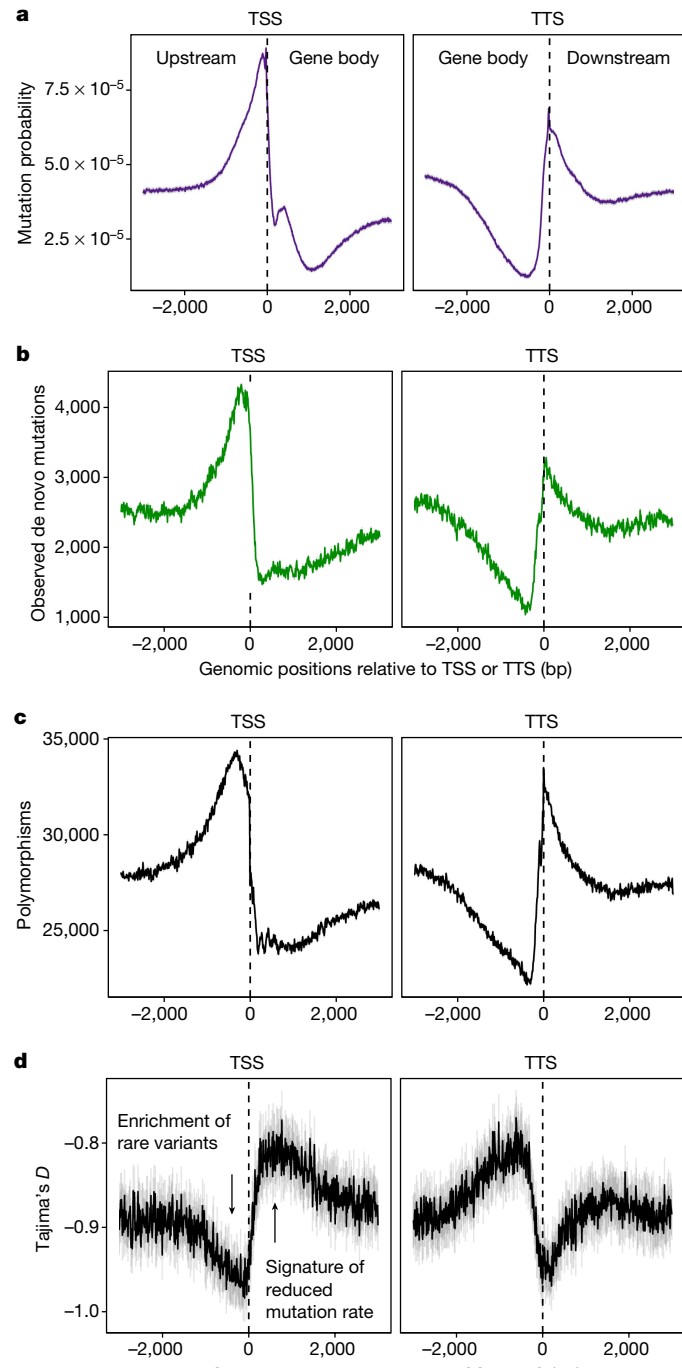

**Fig. 2 | Lower mutation rate in gene bodies. a**, Mutation probability score (predicted SNVs plus indels per base pair from models in Fig. 1e; mean ± 2 s.e.m. in grey) based on epigenomic states and mutations observed in original mutation accumulation lines. **b**, Observed de novo mutations from all independent mutation accumulation datasets (mean ± 2 s.e.m. in grey, bootstrapped). **c**, Segregating polymorphisms (SNVs plus indels, $S$, mean ± 2 s.e.m. in grey, bootstrapped) in 1,135 *Arabidopsis* accessions[35]. **d**, Tajima's $D$ calculated from polymorphisms in *Arabidopsis* accessions[35] around TSS and TTS (mean ± 2 s.e.m. in grey). Note that these TSS and TTS plots do not consider gene length or intergenic distances and that, for example, not all sequences downstream of TSSs are genic sequences, and not all sequences upstream of TSSs are intergenic sequences. Specifically, we did not distinguish between intergenic regions (or genes) longer or shorter than 3,000 bp.

Genetic diversity in a global set of *Arabidopsis* accessions[35] supported these results (Fig. 2c, d). Over 90% of the variance in polymorphism levels found around gene bodies could be explained by our

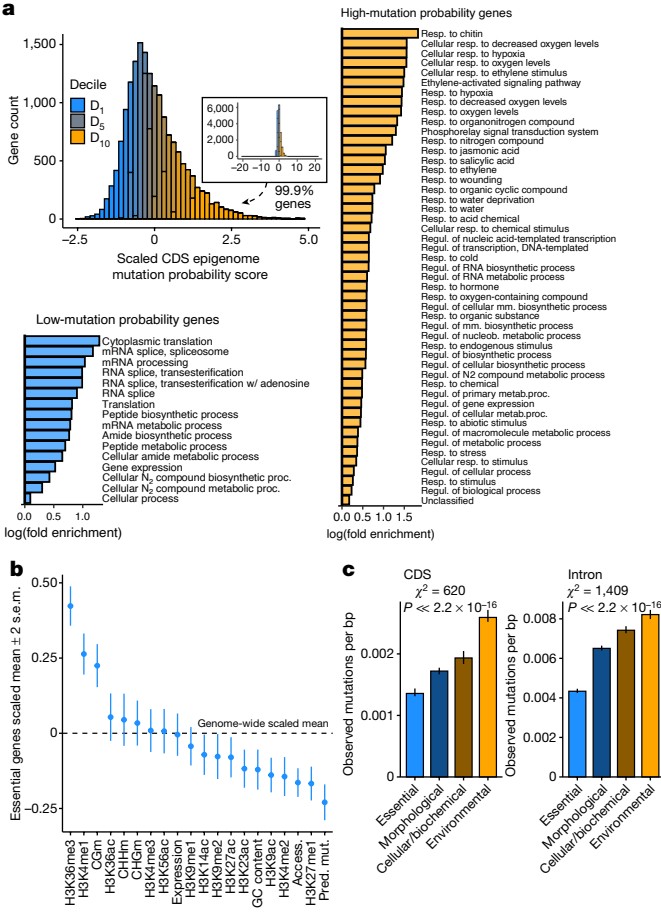

**Fig. 3 | Lower mutation probability in essential genes. a**, Variation in epigenome-derived mutation probability scores in coding sequence (CDS) among genes and gene ontology terms enriched in genes in the top ('high-mutation probability genes') and bottom ('low-mutation probability genes') deciles. Mm., macromolecular; $N_2$, nitrogen; nucleob., nucleobase-containing compound; reg., regulation; resp., response. **b**, Enrichment of epigenomic and other features in coding sequences of 719 genes known to be essential from mutant analyses (mean ± 2 s.e.m.). **c**, Total observed mutation rate (±2 s.e.m., bootstrapped) in genes ($n = 2,339$) with experimentally determined functions[38]. The bars are coloured according to relative differences in mutation rates among genes classified by function (that is, orange refers to high mutation rate and blue represents low mutation rate). $P ≈ 0$ for both CDS and intron mutations.

experimentally observed mutation rates (Extended Data Fig. 5). To determine whether low levels of polymorphism in gene bodies were indeed caused by reduced mutation rather than purifying selection, we analysed the site frequency spectrum. Theory shows that purifying selection causes an enrichment of rare alleles (reduced frequency of deleterious variants), whereas site frequency spectrum scales with mutation rate such that lower mutation rate causes a depletion of rare alleles (fewer young alleles)[36]. Our analysis of the site frequency spectrum statistic Tajima's $D$ around genes confirmed a depletion of rare alleles in gene bodies (less negative $D$), consistent with a reduced mutation rate. We validated this inference with extensive forward population genetic simulations (Extended Data Fig. 6). In conclusion, evolution around genes in *Arabidopsis* appears to be explained by mutation bias to a greater extent than by selection.

## Gene structure and mutation

We further discovered emergent relationships between gene structure and mutation rate (Extended Data Fig. 7). Owing to the distribution of

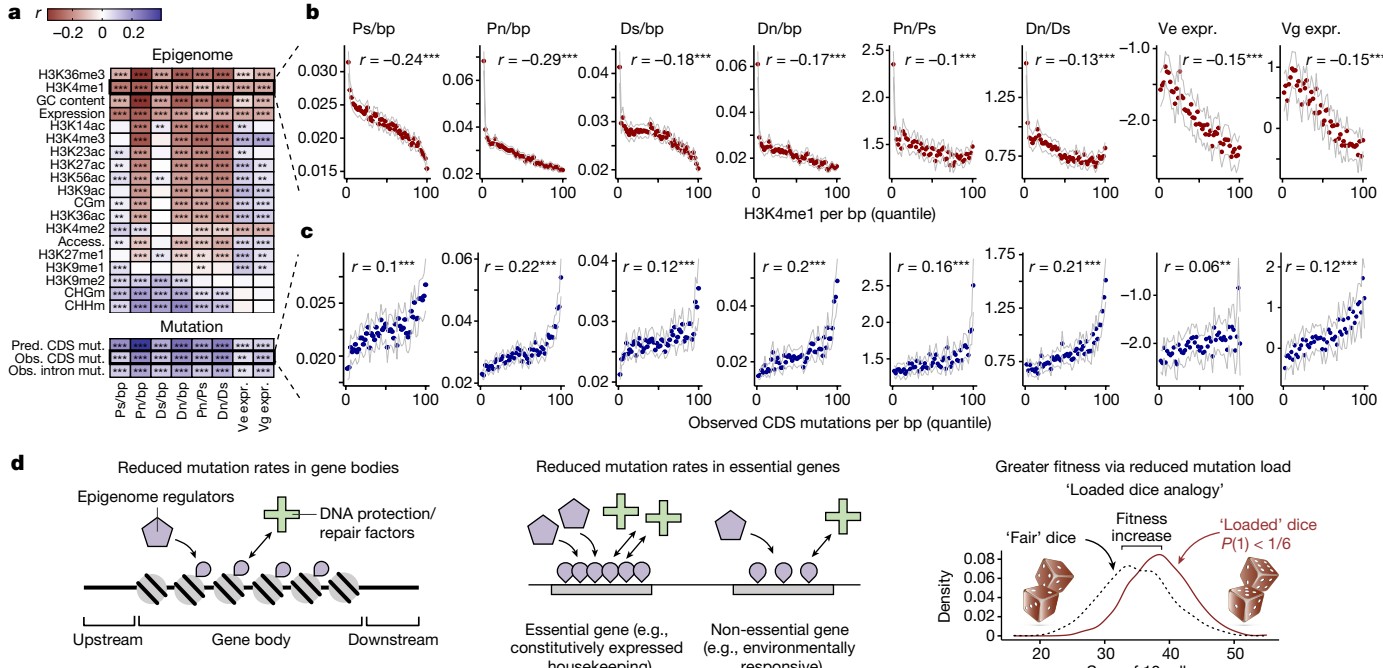

**Fig. 4 | Adaptive reduction in deleterious mutations. a**, Correlations between epigenomic and other features, predicted and observed mutation rates, and measures of evolutionary constraint and rates of sequence evolution. Synonymous (Ps) and non-synonymous polymorphism (Pn) in natural populations, synonymous (Ds) and non-synonymous divergence (Dn) from *Arabidopsis lyrata*, environmental variance of gene expression (Ve expr.) and genetic variance of gene expression (Vg expr.). 'Pred. mut.' is the predicted mutation rate as a function of epigenomic and other features. 'Obs. mut.' is the observed mutation rate in genes based on de novo mutations called across all mutation accumulation datasets. ***$P < 2 \times 10^{16}$, **$P < 0.05$. **b, c**, Relationship between H3K4me1 (**b**) and estimates of evolutionary constraint and rate of sequence evolution (**c**) across quantiles of observed mutation rates per gene. Pearson correlation reflects raw correlation across genes. Data are visualized by mean values ± 2 s.e.m. in 50 quantiles (each quantile = 2% of genes). **d**, Conceptual diagrams summarizing our findings.

epigenomic features along gene bodies, mutation probabilities are highest in extreme 5′ and 3′ coding exons. Natural polymorphisms in *Arabidopsis* and *Populus trichocarpa* showed a similar pattern. Consistent with the effects of mutation bias, *D* was more negative in peripheral exons. The predicted mutation rate of coding regions was 28% and 39% higher in genes annotated as lacking 5′ untranslated regions (UTRs) and 3′ UTRs, respectively. The inferred effect size of 5′ UTRs and 3′ UTRs on coding-exon mutation probabilities and polymorphism was greatest in extreme 5′ and 3′ coding exons. UTR lengths were negatively correlated with mutation probabilities and polymorphisms in peripheral coding exons. Mutation probabilities were also 90% greater in genes lacking introns and lower in genes with more ($r = -0.34$) and longer ($r = -0.24$) introns. These patterns were mirrored by patterns of polymorphism and Tajima's *D*. In conclusion, an unexpected emergent effect of UTRs and introns in *Arabidopsis* appears to be lower mutation rates in coding regions.

## Fewer mutations in essential genes

We next investigated mutation rates in relation to gene functions, discovering that genes with the lowest epigenome-predicted mutation rates were enriched for conserved biological functions (for example, translation). By contrast, genes with the highest predicted mutation rates had specialized functions (for example, environmental response) (Fig. 3a). Comparing genes whose effects have been measured with knockout experiments[37] confirmed that essential genes are enriched for epigenomic features associated with low mutation, and, as predicted, observed mutation rates were significantly lower in the coding regions of essential genes. By contrast, genes with environmentally conditional functions had the highest mutation rates. Intron mutations showed the same pattern, confirming that these results

are not due to selection on coding sequences biasing our mutation datasets (Fig. 3c). We found no evidence that reduced mutation rate in essential genes could be explained by the potential intrinsic mutational properties of CG methylation, expression level or GC content. Instead, the observed 37% reduction in mutation rates in essential genes is consistent with a reduction in mutation, plausibly explained by their enrichment for low-mutation-associated epigenomic features (for example, H3K4me1).

These results were further supported by our discovery of reduced mutation rate in genes with lethal knockout effects[38] and broadly expressed genes[39]. Again, these results were consistent with epigenomic profiles (Extended Data Fig. 8). In conclusion, we find that genes with the most important functions experience reduced mutation rate, as predicted by their epigenomic features.

## Reduction in mutation load

Comparing predicted mutation rates with signatures of evolutionary constraint revealed that genes subject to purifying selection are enriched for epigenomic features associated with low mutation rate (Fig. 4a, b). We confirmed these predictions with our dataset of empirical mutations—mutation rate was significantly correlated with measures of evolutionary constraint on coding and regulatory function (Fig. 4a, c). These patterns were replicated in analyses of mutations in introns, where selection is weaker than in exons, further indicating that results are not due to selection biasing our mutation datasets. These findings demonstrate that genes subject to stronger purifying selection are maintained in epigenomic states that underlie a significant reduction in their mutation rate (Extended Data Fig. 9). In conclusion, mutation bias acts to reduce levels of deleterious variation in *Arabidopsis* by decreasing mutation rate in constrained genes.

## Evolution of mutation bias

Our findings reveal adaptive mutation bias that is mediated by a link between mutation rate and the epigenome. This is mechanistically plausible in light of evidence that DNA repair factors can be recruited by specific features of the epigenome[8]. Hypomutation targeted to features enriched in functionally constrained loci throughout the genome would reduce the relative frequency of deleterious mutations. The adaptive value of this bias can be conceptualized by the analogy of loaded dice with a reduced probability of rolling low numbers (that is, deleterious mutations), and thus a greater probability of rolling high numbers (that is, beneficial mutations) (Fig. 4d).

This intuitive model fits established theory showing that adaptive mutation bias could evolve despite drift when the length of sequence affected ($L_{segment}$) is large[2,3,5,40]. While this criterion can rarely be satisfied for single-gene modifiers, it can be if the mutation is suppressed in many constrained loci. For example, the total sequence length of the coding regions of essential genes enriched for H3K4me1 is three times the estimated minimum $L_{segment}$ required for targeted hypomutation to evolve in *Arabidopsis*, assuming a 30% reduction in mutation rate (Extended Data Fig. 10). Thus, while perhaps initially surprising, our synthesis between epigenomics and population genetic theory predicts that the observed biases could readily arise via natural selection[2].

## Conclusions

While it will be important to test the degree and extent of mutation bias beyond *Arabidopsis*, the adaptive mutation bias described here provides an alternative explanation for many previous observations in eukaryotes, including reduced genetic variation in constrained loci[41] and the genomic distributions of widely used population genetic statistics[42]. Since mutational biases are a product of evolution, they could differ between organisms, potentially explaining differences in the distribution of fitness effects of new mutations among species[43,44]. Finally, because epigenomic features are plastic, epigenome-associated mutation bias could even contribute to environmental effects on mutation[45]. Our discovery yields a new account of the forces driving patterns of natural variation, challenging a long-standing paradigm regarding the randomness of mutation and inspiring future directions for theoretical and practical research on mutation in biology and evolution.

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

## Methods

### Identification of de novo mutations in *A. thaliana*
**Col-0 mutation accumulation lines.** Our training set of mutations was identified from 107 mutation accumulation lines of the *A. thaliana* Col-0 accession, which is the basis of the *A. thaliana* TAIR10 reference genome sequence[12]. The lines had been previously grown for 24 generations of single-seed descent before sequencing with 150-bp paired-end reads on the Illumina HiSeq 3000 platform, of pools of approximately 40 seedlings of each line from the 25th generation (Fig. 1a). Seedlings were sampled at the four-leaf stage, at 2 weeks of age. Variants were identified with GATK HaplotypeCaller[12]. In many organisms, germline mutations are primarily influenced by processes specific to reproductive organs[10]. Because plants may lack a completely segregated germline[46], we hypothesized that mechanisms that influence local mutation rates in the germline may be reflected in the distribution of somatic mutations as well, or at least that the processes governing mutation rate variability across the genome may be similar in germline and somatic tissue. Therefore, in addition to the original variants called[12], we implemented a custom filtering pipeline to identify a high-confidence set of additional de novo mutations (Extended Data Fig. 1). This set included, in addition to somatic variants, germline variants that had not been called in the original analyses[12]. Somatic mutations were previously excluded because they appear as heterozygous calls[12]. Germline mutations were previously excluded if at least 1 out of the 107 lines also included a putative somatic mutation at the same position[12]. On the basis of previously reported germline mutation rates (1–2 per genome and generation) and with the knowledge that these lines were self-fertilized each generation, we expected the seedlings that were sequenced to be segregating for 2–4 additional heterozygous germline variants, which would have been called as somatic mutations by our pipeline (approximately 2–5% of putatively somatic mutations). Because we combined putative somatic and germline mutations to characterize the mutational landscape of the *A. thaliana* genome, this did not have an obvious effect on our results.

**Testing for mutation calling artefacts by resequencing ten siblings of a single-mutation accumulation line.** To test for the possibility that our results were in part artefacts of the pooled-seedling sequencing approach[12], we resequenced entire rosettes of individual plants that were sibling from the same mutation accumulation line (#73) and asked whether the distribution of called variants (that is, putative somatic mutations around TSS and TTS) was similar to the patterns seen with the seedling pools of the 107 individual lines described in the preceding section (Extended Data Fig. 6). Specifically, we grew 10 siblings of line #73 and extracted DNA from 3-week-old whole rosettes. Barcoded PCR-free libraries for the 10 siblings were sequenced, with 150-bp paired-end reads, at approximately 60× depth each on a single lane of the Illumina HiSeq 3000 platform. Additionally, for one sibling, the same library was sequenced in an independent lane at approximately 600× depth. After adapter and quality trimming with cutadapt (version 2.3) and removing duplicates with samtools markdup (version 1.10), reads were aligned to the TAIR10 reference genome with bwa-mem (version 0.7.17) and variants were called independently for each sample with GATK HaplotypeCaller version 4.1.0.

**Measuring the effects of mappability of reads.** We wanted to ensure that variation in mappability could not explain the observed distribution of de novo variants. To evaluate the possibility that results were an artefact of bias in mappability across gene regions, we calculated mappability for $k = 100$, $e = 1$, across the *A. thaliana* reference genome using GenMap[47]. We then plotted and visualized mappability around TSSs and TTSs to confirm that differences in mappability were not the same as the signals of mutation bias detected in our numerous datasets of de novo mutation. While we did not see any evidence that mappability

bias covaried with patterns of mutation bias, for building our predictive model of mutation rate as a function of epigenomic and other features, we still chose to filter out variants called in regions of poor mappability (±100 bp of mappability < 1), as our analysis of resequenced siblings suggested that variants called in low-mappability regions are more likely to be false positives (since variants called in many independent lines had lower mappability).

**Simulating reads and identifying true false positives.** To further rule out artefacts, we calculated the expected distribution of false positives using simulated short reads. We simulated Illumina reads based on the TAIR10 reference genome using ART[48] with the following parameters: -l 150 -f 30 -m 500 -s 30. Reads were mapped to the TAIR10 genome with NextGenMap, the same caller as used in the original calling of mutation accumulation lines[49], and variants were called with GATK HaplotypeCaller. This was repeated for a total of 1,000 simulated genomes. Because these are simulated reads, all variants that are called must be false positives. To test the possibility that the main results found in this study, such as elevated mutation and polymorphism upstream of TSSs, are artefacts of bias resulting from Illumina sequencing (which is included in simulations) or from mapping error (which is captured by mapping the simulated reads), we plotted the distributions of false positives around these regions to confirm that the distribution of false positives was more similar to likely false positives (for example, called in many lines) and unlike the higher confidence variants called in real sequencing data.

### Identification of de novo mutations in a new *A. thaliana* mutation accumulation experiment
To validate our predictive model of the mutation probability score, we used a second *A. thaliana* mutation accumulation experiment descended from eight founders collected in natural environments[50]. The lines were grown for seven to ten generations of single-seed descent before 150-bp paired-end read Illumina sequencing of pools of 40 seedlings. The specifics of the populations were as follows: founder CN1A18: 56 lines for 10 generations; founder CN2A16: 51 lines for 10 generations; founder SJV12: 48 lines for 7 generations; founder SJV 15: 36 lines for 7 generations; founder RÖD4: 50 lines for 8 generations; founder RÖD6: 50 lines for 8 generations; founder SB4: 53 lines for 8 generations; and founder SB5: 56 lines for 8 generations. Mutations were identified as described in ref. [11]. Briefly, raw reads were mapped to the TAIR10 reference genome, variants were called using GATK HaplotypeCaller, merged with the GenotypeGVCFs tool and filtered by variant quality (QD > 30) and read depth (DP > 3). A germline mutation was called if a single mutation accumulation line per founder population had a homozygous alternative allele. Somatic mutations were called as heterozygous variants found in only one of the mutation accumulation lines derived from a single founder genotype. This should remove any true heterozygous calls, variants between cryptic duplications in the founder, and low confidence calls, as suggested by our preceding analyses by resequencing siblings from the original mutation accumulation experiment.

### Identification of de novo somatic mutations in a resequencing dataset of *A. thaliana* leaves
To further test our power to predict the distribution of de novo mutations in an independent experiment, we used published data generated from Illumina sequencing of 64 samples of leaf tissue (rosettes and cauline leaves) of two Col-0 plants[21]. Raw fastq files were downloaded from NCBI and forward reads were mapped twice to the TAIR10 reference genome using bwa-mem (bwa mem ${sample}_**R1**.fastq.gz ${sample}_**R1**.fastq.gz), and duplicate reads (that is, PCR duplicates) were filtered using samtools markdup. Variants for every sample were called with GATK HaplotypeCaller. Variants were filtered to include only those found in a single sample (as our previous work had already shown that putative somatic variants called in many independent samples tend

to be enriched for regions of low mappability and exhibit distributions more similar to the expected distribution of false positives).

**De novo mutations in a natural mutation accumulation lineage.** We analysed mutations that had accumulated in a single *A. thaliana* lineage that recently colonized North America[32]. The 100 samples came both from modern populations as well as historical herbarium specimens and contained 8,891 new variants with at least 50% genotyping rate in the population. Phylogenetic coalescent analyses indicated that these 100 samples shared a common ancestor around 1519–1660, presumably the ancestor that colonized North America, and thus that these lines have recent mutations that accumulated after a population bottleneck (small $N_e$) and therefore under weak selection[32]. We used these to study the level of polymorphisms around TSSs and TTSs in a wild population with a simple demographic history.

## Constructing a model to predict mutation probability

**Sequence and epigenomic features.** We were interested in studying epigenomic features plausibly linked to mutation rate[16–19,28,51–55]. To build a high-resolution predictive model of mutation rate variation, we extracted or generated data describing genome-wide sequence and epigenomic features. First, we calculated GC content (% of sequence), which can affect DNA denaturation[5,25,56–58], across regions[9,23,59–64]. From the Plant Chromatin State Database, we also downloaded 62 BigWig formatted datasets characterizing the distribution of histone modifications[14] H3K4me2, H3K4me1, H3K4me3, H3K27ac, H3K14ac, H3K27me1, H3K36ac, H3K36me3, H3K56ac, H3K9ac, H3K9me1, H3K9me2 and H3K23ac, many of which have been linked to mutational processes[8,9,11,12,19,33,65–70]. For each specific histone modification, depths were scaled (0 to 1) and averaged across each region for downstream analyses.

**Col-0 cytosine methylation.** Because cytosine methylation is known to affect mutation rates via deamination of methylated cytosines[9,11,12,33,66], we wanted to include cytosine methylation as a predictor variable in our model. Methylated cytosine positions for Col-0 (6909) wild-type leaves were obtained from the 1001 Epigenomes dataset GSM1085222 (ref.[71]) under the file GSM1085222_mC_calls_Col_0.tsv.gz. Because the context of cytosines can vary and influence the functional effect of methylation, cytosines were further classified into three categories (CG/CHG/CHH) for all downstream analyses. For each region, we calculated the number of methylated cytosines in each category per bp.

**Chromatin accessibility.** ATAC-seq can measure chromatin accessibility, which also affects mutation rates[9,11,12,33,66,72]. Col-0 seeds were stratified on MS-agar (with sucrose) plates at 4 °C for 4 days in the dark. Plates were transferred to 23 °C long-days and kept vertically for easier harvesting of seedlings. On the eleventh day of light exposure, 10–20 seedlings each from three MS-agar plates were fixed with formaldehyde by vacuum infiltration and stored at −80 °C.

Fixed tissue was chopped finely with 500 μl of general purpose buffer (GPB; 0.5 mM spermine•4HCl, 30 mM sodium citrate, 20 mM MOPS, 80 mM KCl, 20 mM NaCl, pH 7.0, sterile filtered with a 0.2-μm filter, followed by the addition of 0.5% of Triton-X-100 before usage). The slurry was filtered through one-layered Miracloth (pore size: 22–25 μm), followed by filtration through a cell strainer (pore size: 40 μm) to collect nuclei. Approximately 50,000 DAPI-stained nuclei were sorted using fluorescence-activated cell sorting (FACS) as two technical replicates. Sorted nuclei were heated to 60 °C for 5 min, followed by centrifugation at 4 °C (1,000g for 5 min). Supernatant was removed, and the nuclei were resuspended with a transposition mix (homemade Tn5 transposase, a TAPS-DMF buffer and water) followed by a 37 °C treatment for 30 min. 200 μl SDS buffer and 8 μl 5 M NaCl were added to the reaction mixture, followed by 65 °C treatment overnight. Nuclear fragments were then cleaned up with Zymo DNA Clean & Concentrator

columns. 2 μl of eluted DNA was subjected to 13 PCR cycles, incorporating Illumina barcodes, followed by a 1.8:1 ratio clean-up using SPRI beads. Genomic DNA libraries were prepared using the same library preparation protocol from the Tn5 enzymatic digestion step onwards.

Each technical replicate (derived from nuclei sorting) was sequenced with 3.5 million 150-bp paired-end reads on an Illumina HiSeq 3000 instrument. The reads were aligned as two single-end reads to the TAIR10 reference genome using bowtie2 (default options), filtered for the SAM flags 0 and 16 (only reads mapped uniquely to the forward and reverse strands), and converted separately to .bam files. The .bam files were merged, sorted, and PCR duplicates were removed using picardtools. The sorted .bam files were merged with the corresponding sorted bam file of a second technical replicate (samtools merge --default options) to obtain a final depth of approximately 6 million reads for each replicate.

Peaks were called for each biological replicate using MACS2 using the following parameters:

macs2 callpeak -t [ATACseqlibrary].bam -c [Control_library].bam -f BAM --nomodel --shift −50 --extsize 100 --keep-dup=1 -g 1.35e8 -n [Output_Peaks] -B -q 0.05

Peak files and .bam alignment files from three biological replicates were processed with the R package DiffBind to identify consensus peaks that overlapped in at least two replicates (FDR < 0.01). Library quality was estimated by measuring the frequency of reads in peak (FRIP) scores for all three replicates, which were 0.36, 0.36 and 0.39, above the standard quality threshold of 0.3.

**Gene expression.** Gene expression was calculated as the mean across 1,203 accessions[71], from which we also extracted the genetic variance (Vg) and environmental variance (Ve) as well as the coefficient of variation (variance/mean) in expression for each gene. This dataset provided information for 17,247 genes with complete data.

**Predictive model of mutation rates.** We wanted to ask whether intragenomic mutation variability in the genome could be predicted by features of the genome that previous work had shown to have potential or demonstrated relationships with mutations. To model mutation rate genome-wide at the level of individual genes, we created a generalized linear model. The response variable was the untransformed (that is, assuming normality, to avoid risk of increased false positives caused by transformation[73,74]) observed mutation rate across every genic feature (upstream, UTR, coding, intron and downstream). The predictor variables were GC content, classes of cytosine methylation, histone modifications, chromatin accessibility and expression of each gene. From this full model, a limited predictive model was selected on the basis of forward and backward selection with the lowest AIC value by the stepAIC function in R. These models were created separately for indels (adjusted $R$-squared: 0.001791; $F$-statistic: 34.6 on 16 and 299635 d.f.; $P < 2.2 \times 10^{-16}$) and SNVs (adjusted $R$-squared: 0.0009687; $F$-statistic: 37.32 on 8 and 299643 d.f.; $P < 2.2 \times 10^{-16}$). For downstream analyses, we used the predicted mutation probability (the mutation probability score) based on these models (predicted SNVs + indels) for genes, exons and other regions of interest from the TAIR10 genome annotation. While the linear regression approach used here enables hypothesis testing to some extent (one can generate confidence intervals and $P$ values describing the level of significance of individual effects), our primary goal was to create a predictive model of mutation bias as a function solely from genomic and epigenomic features; the causality of the associations uncovered in these analyses for individual predictors must be confirmed with future functional work.

**Variance inflation factor.** To test whether our results were skewed by overly correlated predictor variables (included in the model even after model reduction by minimizing AIC), we explored models where predictor variables were manually removed on the basis of their variance

inflation factor score. Specifically, we used the vif function from the R package car to calculate variance inflation factor scores for each variable in our best AIC models for SNVs and indels. We then removed all variables with scores below 3. We recalculated mutation probability scores for every genomic feature. Because the resulting predicted mutation probability scores were very similar, with Pearson correlation $r = 0.95$ between gene-level mutation probability scores from the full model and the reduced model, we report only results based on the full model.

## Analysis of natural polymorphism rates

**Rates of polymorphism among genic exons.** We calculated rates of natural polymorphism across exons in TAIR10 gene models from sequence variation among 1,135 natural *A. thaliana* accessions[35]. These analyses revealed elevated polymorphism rates in peripheral (first and last) exons. To test whether this is an artefact unique to *A. thaliana*, we calculated rates of natural polymorphism across exons from sequence variation among 544 *P. trichocarpa* accessions[75]. Specifically, we downloaded VCF and annotation data from Phytozome (v3.0) and calculated rates of variation across exons grouped by order (from 5′ to 3′) and total exon number.

**Signatures of selection and constraint from natural populations.** We calculated gene-level summary statistics for signatures of selection and constraint in the following way. Synonymous and non-synonymous polymorphism among natural *A. thaliana* accessions and divergence from *A. lyrata* (Pn, Ps, Dn and Ds, respectively) were calculated using mkTest.rb (https://github.com/kr-colab). The alpha test statistic for evidence of selection, which is a derivative of the McDonald-Kreitman test[76–78], was calculated from these values for each gene where data were available (not all genes have orthologues assigned in *A. lyrata*) as 1 − (Ds × Pn)/(Dn × Ps). Positive values of alpha are conventionally interpreted as evidence of positive selection because non-synonymous variants in genes with such values tend to become fixed. For each decile of genes classified according to mutation probability, we calculated the proportion for which alpha is positive. Enrichment of non-synonymous variants compared to genome-wide average were confirmed by independent calculation of Waterson's diversity estimate (θ) of non-synonymous variation. The frequency of loss-of-function mutations was calculated as before[79,80], where loss of function was defined as premature stop codons and frameshifts disrupting at least 10% of the coding region of the canonical gene model. Genes experiencing purifying selection should exhibit lower levels of natural polymorphism than what would be predicted by mutation rate alone. To test this, we built a linear model of coding region polymorphisms as a function of predicted mutation rates. We calculated scaled residuals for each gene and tested whether they are more negative in genes expected to be under purifying selection. To estimate constraints on gene regulatory function, we looked at average expression across diverse genotypes. We also tested for relationships between predicted mutation rates and the coefficient of variation in gene expression, additive genetic variance for gene expression across diverse genotypes, and environmental variance in gene expression[71].

**Relationships between epigenomic and other features, mutation rates and gene function.** The preceding analyses revealed significant associations between epigenomic and other features and signatures under selection indicating that genes that experience purifying selection are enriched for features associated with low mutation rate. To further dissect the mechanistic basis of this pattern, we wanted to directly test for relationships between epigenomic states, mutation rates and gene function. We analysed gene ontology categories for genes in the top and bottom deciles ranked by predicted mutation rate[81], reporting gene ontologies that were significantly enriched in these groups after Bonferroni adjustment of raw *P* values.

We also analysed a manually curated dataset of mutation-induced lethality obtained from phenotyping lines with loss-of-function mutations[37]. Genes annotated as lethal effect when mutated (that is, required for viability) were compared with genes showing non-lethal phenotypic effects to assess differences in epigenomic and other features.

We analysed a dataset of phenotypes from 2,400 *A. thaliana* knockout lines[38]. Genes had been classified as being essential (such as an RNA processing gene where loss of function results in lethality[82]), causing morphological defects (for example, altered stomata and trichome size), cellular biochemical defects (for example, intracellular transport of small molecules) and conditional defects (for example, effects depending on the environment). We then compared epigenomic and other features in essential genes to other classes of genes. These analyses showed that genes with essential functions were enriched for features associated with reduced mutation, whereas genes annotated as having non-essential functions were depleted for these features.

## Estimating selection on different types of de novo mutations

Synonymous, non-synonymous and stop-gained variants are expected to have different effects on gene function, although they are of the same mutational class (SNVs). They are all from coding regions, which have an overall mutation probability that is distinct from other regions of the genomes, such as introns, in our model of de novo mutations. For comparison, we calculated the rates of synonymous, non-synonymous and stop-gained SNVs in natural populations of *A. thaliana*, which have been subject to long-term natural selection. We also derived an expected null ratio of non-synonymous to synonymous mutations using knowledge on the relative base composition of all coding regions in the reference genome, the relative proportion of coding region mutations (for example, CG to TA mutations are most common), and the proportion of all possible codon transitions that lead to synonymous versus non-synonymous mutations. Ratios of non-synonymous to synonymous and stop-gained to synonymous mutations were compared between observed de novo mutations and those observed in natural populations or the null expectation by chi-squared tests.

**Expected non-synonymous-to-synonymous substitution ratios in the absence of selection.** To further validate that the observed de novo mutations we used to train our mutation probability model were not subject to appreciable selection, we simulated 10,000 de novo mutations across the *Arabidopsis* genome with custom scripts in R. Mutations in coding regions were randomly assigned to non-synonymous or synonymous changes based on codon use and observed mutational spectra of coding regions. We then calculated the observed ratio of non-synonymous to synonymous mutations in the simulated data. We repeated this simulation 10,000 times to produce a distribution of expected non-synonymous-to-synonymous ratios. We then compared the non-synonymous-to-synonymous ratio in our observed de novo mutations to this distribution. Finally, we tested whether our observation fell within the 95% bootstrapped interval.

**Expected number of synonymous mutations under random variation.** Because we had found that observed mutations were less frequent in coding regions, we wanted to determine whether this difference was significantly higher than expected by chance. We therefore asked how the number of synonymous mutations observed compared with that expected under a random process, starting with a simulated set of random mutations across the genome. We calculated the number of these mutations in coding regions that are expected to lead to a synonymous nucleotide substitution based on codon use and observed mutational spectra of coding regions. We repeated this simulation 1,000 times to generate a distribution of expected synonymous mutations. Comparing our observed de novo synonymous mutations to the mean of this distribution, we calculated the reduction in the observed synonymous mutation rate.

**Non-synonymous-to-synonymous ratios and mutation probabilities in more deleterious ('lethal effect versus non-lethal effect') genes.** We wanted to test whether the rates of non-synonymous-to-synonymous variation were lower in genes that are predicted to experience stronger negative selection. We split genes with a high-essentiality and low-essentiality prediction score (see above) or empirically determined lethal versus non-lethal effects of loss-of-function alleles (see above)[37]. We then calculated the differences in the observed mutation rate between these groups of genes and compared them with a *t*-test. We also calculated the number of observed non-synonymous and synonymous SNVs in these groups of genes and compared their ratios by a chi-squared test.

**Non-synonymous-to-synonymous ratios in mutation probability deciles.** We wanted to test whether mutation probability deciles predicted by our model differed in their rates of non-synonymous to synonymous mutations in our observed de novo mutations. If there was a strong gradient (for example, if genes predicted to have low mutation rate had lower rates of non-synonymous variation than genes predicted to have high mutation rate), this could suggest an effect of purifying selection acting directly on the detected mutations. To improve the power to detect differences among genes differing by mutation probability scores, we also assigned mean expression values to genes for which expression could not be called in our expression dataset[71] and calculated mutation probability score. We binned genes into mutation probability deciles and compared mutation deciles and their corresponding non-synonymous-to-synonymous ratio to confirm that there was no relationship suggestive of selection.

**Minor allele frequencies in natural populations.** Our results had indicated that mutation rates were high upstream and downstream of genes relative to the gene bodies, not only in observed and predicted de novo mutations but also in natural polymorphisms. If this pattern was driven by mutation bias, we would expect to see lower minor allele frequencies upstream and downstream of genes, because this would indicate the presence of newly derived alleles from recent mutation rather than lower minor allele frequency caused by greater negative selection since we expect a priori that gene bodies (particularly coding regions whose code makes them sensitive to mutation) are subject to greater constraint. Conversely, lower minor allele frequencies in gene bodies would be consistent with the action of purifying selection in gene bodies, because lower allele frequencies are expected when negative selection had an opportunity to reduce allele frequencies. We therefore calculated the minor allele frequency (vcftools --freq) and their mean for every polymorphic position in the genome of 1,135 natural *A. thaliana* accessions[35] in relation to TSSs and TTSs across the entire genome.

**Tajima's *D* around gene bodies.** Tajima showed that reduced mutation and purifying selection, while having the same effect to reduce the number of polymorphisms, have opposite effects on his statistic, *D*[36]. That is, mutation rate has a scaling effect on *D* such that reduced mutation rates lead to less negative *D*, whereas purifying selection leads to more negative *D*. Therefore, analysis of *D* can be used to quantify the relative importance of these alternative, but not mutually exclusive, forces shaping rates of sequence evolution. *D* is, on average, negative across the *A. thaliana* genome, and *D* also scales with mutation rate. Thus, if *D* is more negative in regions with lower polymorphism, this could indicate that purifying selection is the dominant force underlying lower rates of variation. By contrast, if *D* is less negative in regions of low polymorphism, this would indicate that lower mutation rate is the primary force responsible for lower rates of variation. Therefore, to further investigate whether the observed rates of polymorphism around gene bodies in 1,135 natural *A. thaliana* accessions were driven at least

in part by mutation biases or only by selection, we calculated Tajima's *D* (vcftools --TajimaD) in 100-bp windows across the entire genome and averaged these values in relation to TSSs and TTSs for every gene. We used bootstrapping (*n* = 100) to calculate the confidence interval (±2 s.e.m.) around this mean value.

**Tajima's *D* in exons.** We used Tajima's *D* to estimate the extent to which mutation bias rather than selection after random mutation could explain differences in rates of natural polymorphism in exons (elevated polymorphism in peripheral exons). We calculated Tajiima's *D* in every exon and grouped genes according to their total number of exons and plotted the average Tajiima's *D* in relation to exons ordered from 5' to 3' ends. Tajima's *D* was consistently more negative in peripheral exons, reflecting the effects of increased population mutation rate in these loci, so we further investigated the underlying causes by testing whether genes with and without (and longer or shorter) UTRs have differences in Tajima's *D* in peripheral exons. Finally, we asked whether genes with more and longer introns have less negative Tajima's *D* values, to test whether the lower rates of polymorphism observed in these genes was caused at least in part by reduced mutation rate, rather than selection after random mutation.

## Simulations of mutation bias and selection using SLiM
Our observation that Tajima's *D* is less negative in regions of low polymorphism, such as gene bodies, suggested that the reduced polymorphism therein is caused by a lower mutation rate, consistent with the mutation biases that we discovered in the analysed mutation datasets. To verify this interpretation, we conducted simulations using the software SLiM (v3)[83]. These simulations modelled genic and intergenic space, based explicitly on the first 100 genes on chromosome 1. For each simulation, we modelled a population of 1,000 individuals for 10,000 generations. The selfing rate was assigned to 0.98, a low estimate based on field observations[84,85]. The baseline mutation rate (per base and per generation) was derived from the empirically measured population mutation rate[13] (from $N_e = $ -300,000, $u = -1 \times 10^{-9}$ and adjusted for $N_e = 1,000$). Recombination rate (probability per genome per generation) was $1 \times 10^{-4}$. To investigate the effects of mutation bias and selection, we assigned a scaled mutation rate in gene bodies of 0.2, 0.5 or 1, reflecting an 80%, 50% or 0% reduction relative to the baseline mutation rate in intergenic spaces. We also assigned proportions of deleterious mutations to be 0, 0.1 and 0.3, reflecting a 0%, 10% and 30% frequency of deleterious mutations independently in gene bodies and intergenic regions. All possible combinations of the three parameters were then simulated 200 times. Tajima's *D* was calculated across the entirety of each genome in 100-bp windows using VCFtools. The position of each window was calculated in relation to the TSSs and TTSs of each gene. Counts of polymorphisms and Tajima's *D* were averaged across all genomes in 10-bp windows for regions 3 kb upstream and downstream of the TSS and TTS of each gene. The variation in polymorphism level and Tajima's *D* values were compared with the empirical observations of natural polymorphisms in 1,135 natural *A. thaliana* accessions[66] using Pearson correlation.

## Relationship between mutation probability, epigenomic and other features, and breadth of expression across tissues
Because we found that essential genes have higher levels of epigenomic and other features that lower predicted mutation rates, we wanted to further test the hypothesis that essential housekeeping genes were also enriched for such features and therefore experience a subsequently lower probability of mutation and lower de novo mutation calls. We used gene expression data from 54 tissues[39]. We calculated the correlation between the number of tissues with expression of more than 0 and either the predicted mutation probability score or the observed mutations for each gene. Because these results confirmed that genes expressed in more tissues have lower predicted mutation probability

scores, we examined epigenetic features H3K4me1, H3K36me3 and CG methylation, which are enriched in essential genes, finding that genes expressed in all tissues were also enriched for these features.

**Determining the effect of strong purifying selection on coding sequences.** Our results had revealed significant biases in mutation probability in relation to gene bodies. Because we had found that mutations were significantly higher upstream of genes and significantly lower within gene bodies in five independent datasets, we considered the possibility that this overwhelming bias was the result of extremely strong purifying selection on de novo mutations (that is, removal of lethal mutations before they could be detected by us). We therefore simulated 10,000 random mutations across the TAIR10 genome. If mutations fell within coding regions, we randomly assigned them to be removed by selection (that is, dominant lethal). For this, we explored three levels of selection: $s = 0.01$ where 1% of mutations were removed (that is, had lethal effects), $s = 0.1$ where 10% of mutations were removed, $s = 0.2$ where 20% of mutations were removed, or $s = 0.3$ where 30% of mutations were removed. While $s = 0.3$ represents an exceptionally and unexpectedly high level of selection, especially in soma, evidenced by empirical estimates of the extent of gene essentiality in *A. thaliana*, this served as a positive control for observing the effects of extraordinarily strong selection on the expected distribution of mutations in a random mutation model.

**Comparing expected and observed levels of synonymous mutation.** Because we had observed a significant reduction in mutation rate in coding regions, we wanted to test whether this was driven only by functionally impactful mutation (for example, amino acid substitutions). To do so, we simulated 6,182 random SNVs. For each variant, we asked whether it was found within the coding region of any gene. We counted the total number of coding region variants and multiplied this number with the expected fraction, 0.28, of synonymous variants based on *A. thaliana* codon usage and mutation spectrum. We iterated this simulation 100 times to produce a confidence interval of expected synonymous variants in our training set of de novo mutations.

### Reporting summary

Further information on research design is available in the Nature Research Reporting Summary linked to this paper.

## Data availability

A complete table of called mutations is available in Supplementary Data 1. Genic feature (that is, upstream, UTR, intron, CDS, and so on) level data (mutation and epigenomic features) are available in Supplementary Data 2. Gene-level data (for example, mutation, epigenomic and other features, function, expression and selection) are available in Supplementary Data 3. Derived data objects used to create figures can be found as Source Data for individual figures, and additional intermediate data files are available on GitHub (https://github.com/greymonroe/mutation_bias_analysis). Raw mutation data used as our training set were deposited in Figshare (https://doi.org/10.25386/genetics.6456065). Previous raw Illumina sequencing reads from 64 *A. thaliana* leaves are available under NCBI SRA BioProject PRJNA497989. Raw Illumina sequencing reads from additional mutation accumulation experiments (European lines) are available under NCBI SRA BioProject PRJNA770533. Raw reads from the ATAC-seq experiments are available under ENA Project PRJEB48038. Raw reads from resequencing MA73 are available under ENA Project PRJEB48100. Variant data of natural *A. thaliana* accessions are available at http://1001genomes.org/data/GMI-MPI/releases/v3.1/. The TAIR10 reference genome and annotation are available at www.arabidopsis.org. The *P. trichocarpa* reference genome, annotation and variant data are available at https://phytozome-next.jgi.doe.gov/info/Ptrichocarpa_v3_1. Chromatin state data are available through the Plant Chromatin State Database (http://systemsbiology.cau.edu.cn/chromstates). Tissue-specific expression data are available at https://www.ebi.ac.uk/arrayexpress/experiments/E-MTAB-7978/. There are no restrictions on the availability of data used in this study.

## Code availability

Functions to characterize Tajima's $D$ and polymorphisms in relation to TSSs and TTSs are available on GitHub (https://github.com/greymonroe/polymorphology). The annotated code for models and statistical analyses is available on GitHub (https://github.com/greymonroe/mutation_bias_analysis).

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

**Acknowledgements** We thank members of the Weigel laboratory and the broader community for comments on earlier versions of this manuscript, especially A. Britt, P. Flood, K. Krasileva, M. Lynch, D. Petrov, J. Ross-Ibarra, D. Runcie, B. Schmitz and D. Sloan. This work was supported by NSF grants DEB 0844820 and DEB 1257902 (to C.B.F.), NSF DEB 0845413 and DEB 1258053 (to M.T.R.), the UC Davis Department of Plant Sciences (to J.G.M.), and by DFG grant ERA-CAPS 1001G+ and the Max Planck Society (to D.W.).

**Author contributions** All authors contributed to the work presented in this paper. J.G.M. and D.W. conceived the project. M.-L.W., M.T.R. and C.B.F. conducted the mutation accumulation experiments with material from C.B., J.A. and E.I. These lines were sequenced by J.H., M.N. and C.B. T.S. performed ATAC-seq. M.K. and P.C.-B. generated deep resequencing data for siblings. Data analyses were led by J.G.M. with M.-L.W., T.S. and P.C.-B., with major additional contributions from M.L., M.E.-A., M.K., D.K., D.W., C.B.F. and D.W. J.G.M. and D.W. wrote the manuscript with major contributions from T.S., P.C.-B., C.B., M.L., M.E.-A., M.K., D.K., M.-L.W., J.A., M.T.R. and C.B.F. All authors read and provided feedback on the manuscript. Interpretation of data and results were led by J.G.M. and D.W., with major contributions from T.S., P.C.-B., C.B., M.L., M.E.-A., M.K., D.K., M.-L.W., J.A., M.T.R. and C.B.F.

**Funding** Open access funding provided by Max Planck Society.

**Competing interests** The authors declare no competing interests.

**Additional information**
**Correspondence and requests for materials** should be addressed to J. Grey Monroe or Detlef Weigel.

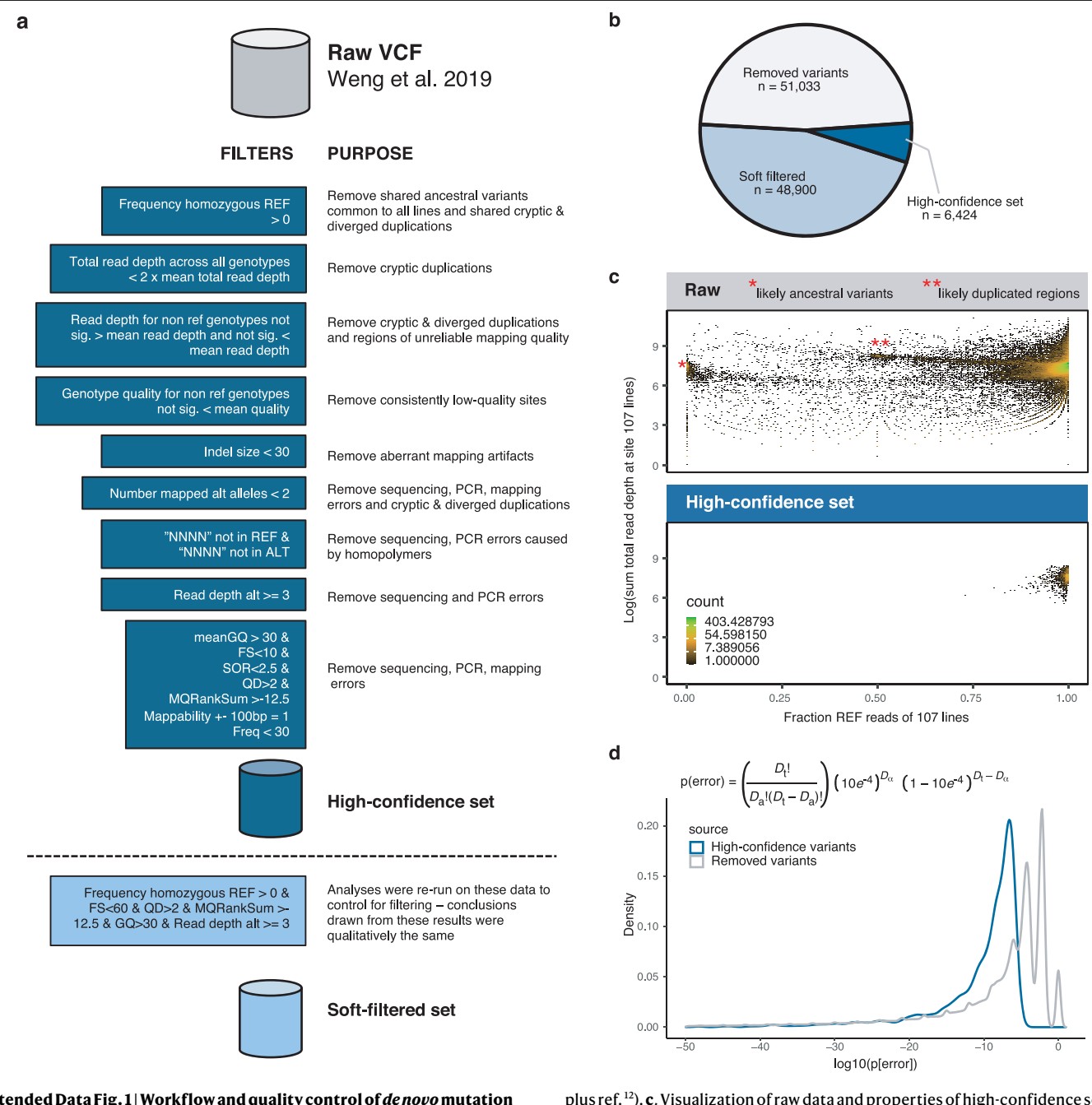

**Extended Data Fig. 1 | Workflow and quality control of *de novo* mutation identification. a**, Filtering pipeline. **b**, High-quality *de novo* mutations called in this study on the original mutation accumulation experiment data (107 replicate lineages of Col-0, total number = mutations from this study plus ref.[12]). **c**, Visualization of raw data and properties of high-confidence set. **d**, Estimated probability of mutation calls being erroneous based on alternative and total read depths in the high-confidence set and variants removed by filtering.

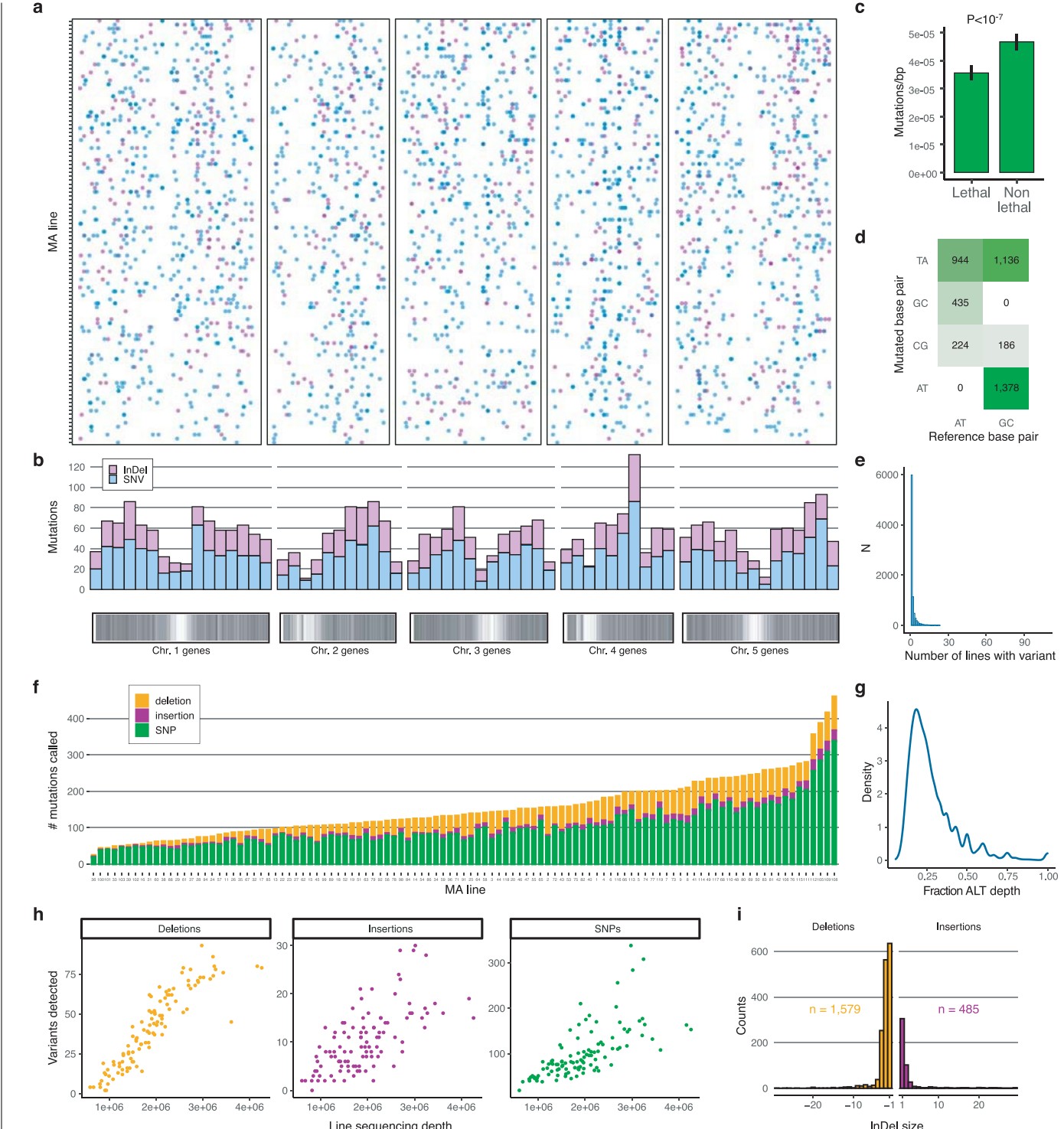

**Extended Data Fig. 2 | Summary of observed *de novo* mutations and distribution across original Col-0 mutation accumulation (MA) lines. a,** *De novo* mutations detected in genic regions (genes ±1,000 bp) in individual MA lines. SNVs in light blue, InDels in magenta. Our investigation was focused on mutations in and around genes, so for clarity mutations elsewhere (i.e., near centromeres) are not shown. **b,** Distribution of mutations across genic regions per 2 Mb windows. Vertical black lines in the lower plot mark the location of genes. **c,** Mutation rates in lethal- and non-lethal-effect genes (n = 27,206 genes, mean ± 2 s.e.m., two-sided t-test) **d,** Frequencies of single nucleotide transitions and transversions. **e,** Distribution of frequency of specific mutations across lines. **f,** Number of germline and somatic mutations detected in each MA line. **g,** Distribution of alternative allele read depth for putative somatic mutations. **h,** Relationship between number of detected mutations and total sequencing depth (total number of informative reads in variant sites) in MA lines. **i,** Size distribution of insertions and deletions.

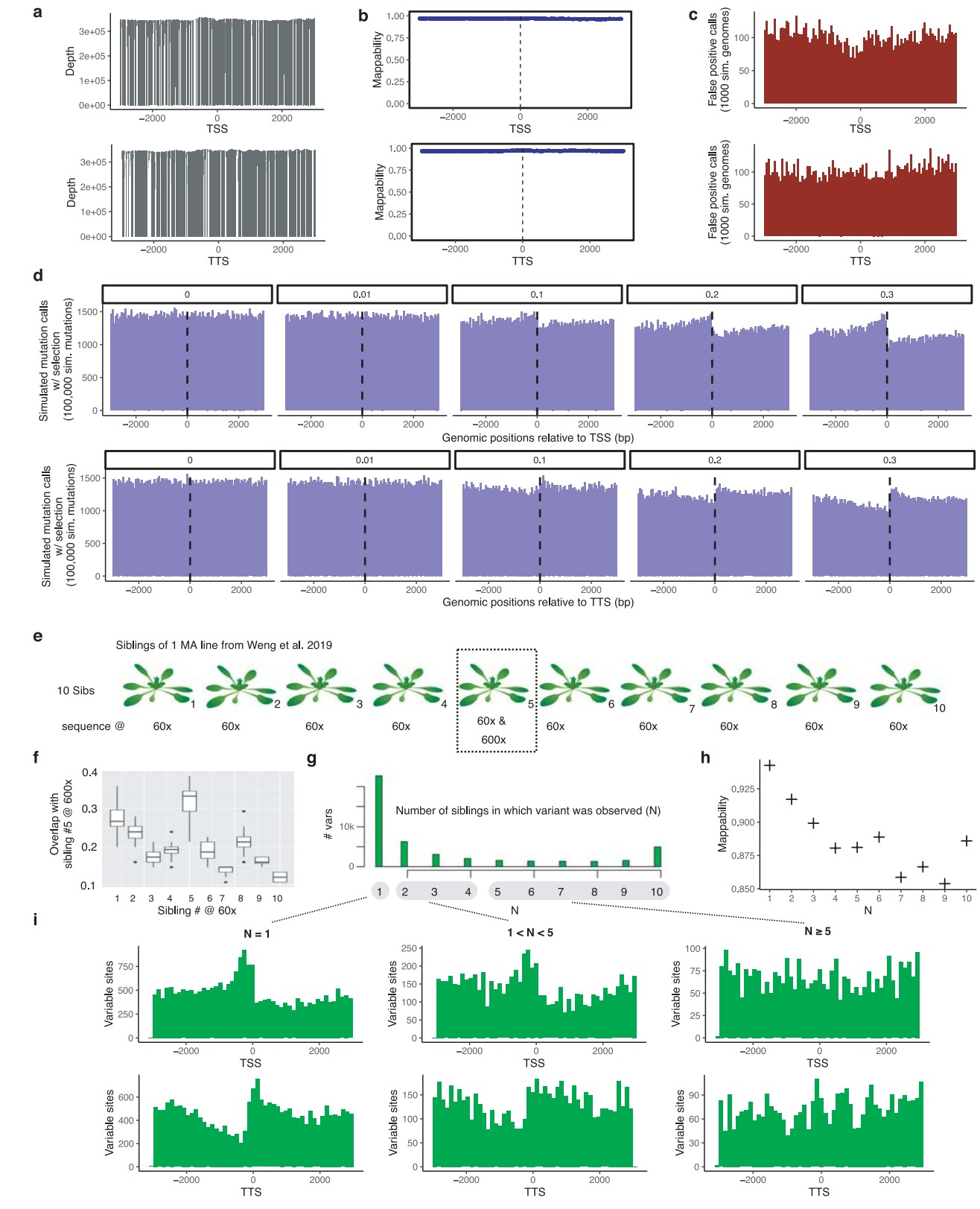

**Extended Data Fig. 3** | See next page for caption.

**Extended Data Fig. 3 | Sequencing depth, mappability, and false positives do not explain observed biases in distributions of natural polymorphisms or observed mutations used to predict mutation probabilities.**
**a**, Sequencing depth around transcription start (TSS) and termination (TTS) sites in one randomly chosen mutation accumulation line. **b**, Mappability around TSS TTS site calculated with GenMap[47]. **c**, Rates of false positive SNP and InDel calls around TSS and TTS determined from 1,000 iterations of simulated Illumina reads. **d**, Simulation of effect of selection on gene bodies. Selection could take the form of mutations being dominant lethal or through somatic competition of mutations with small selection coefficients. 0 = 0%, 0.01 = 1%, 0.1 = 10%, 0.2 = 20%, 0.3 = 30% of gene body mutations removed by purifying selection. 30% is estimated to be the approximate upper bound of constrained sites in gene bodies[34]. **e**–**i**, Resequencing of 10 siblings of one MA line from ref. [12]. **e**, Overview of experimental design for testing the effect of sequencing depth on calling somatic mutations. **f**, Filtered heterozygous variants (SNVs and InDels) called in sibling #5 sequenced at ~600x depth overlap more with variants called from sibling #5 at ~60x sequencing depth than with other siblings at ~60 sequencing depth. The boxplots show the distribution of 20 iterations of sampling equal numbers of heterozygous variants (to account for differences in total number of variants called in different siblings) for each sibling sequenced at ~60x and compared to sibling 5 sequenced at ~600x. Boxplots show median with maxima and minima reflecting interquartile range (IQR), whiskers = 1.5*IQR (n = 20 iterations). **g**, Frequency distribution of unfiltered heterozygous variants called in 10 siblings sequenced at ~60x depth each. Note that because these siblings are descendants of 25 generations of self-fertilization, the number of true heterozygous (inherited segregating) calls is expected to be very small compared to heterozygous variants that are chimeric somatic mutations. **h**, Average mappability of variants detected in different numbers of siblings out of the 10 sequenced siblings. **i**, Variants called independently in one sibling or, less so, in two to four siblings show signatures of mutation bias. In contrast, variants called in five or more siblings (which should more likely be false positives due to cryptic duplications or regions with poor mappability) do not show a biased distribution around TSS and TSS, with overall distribution similar to known false positives.

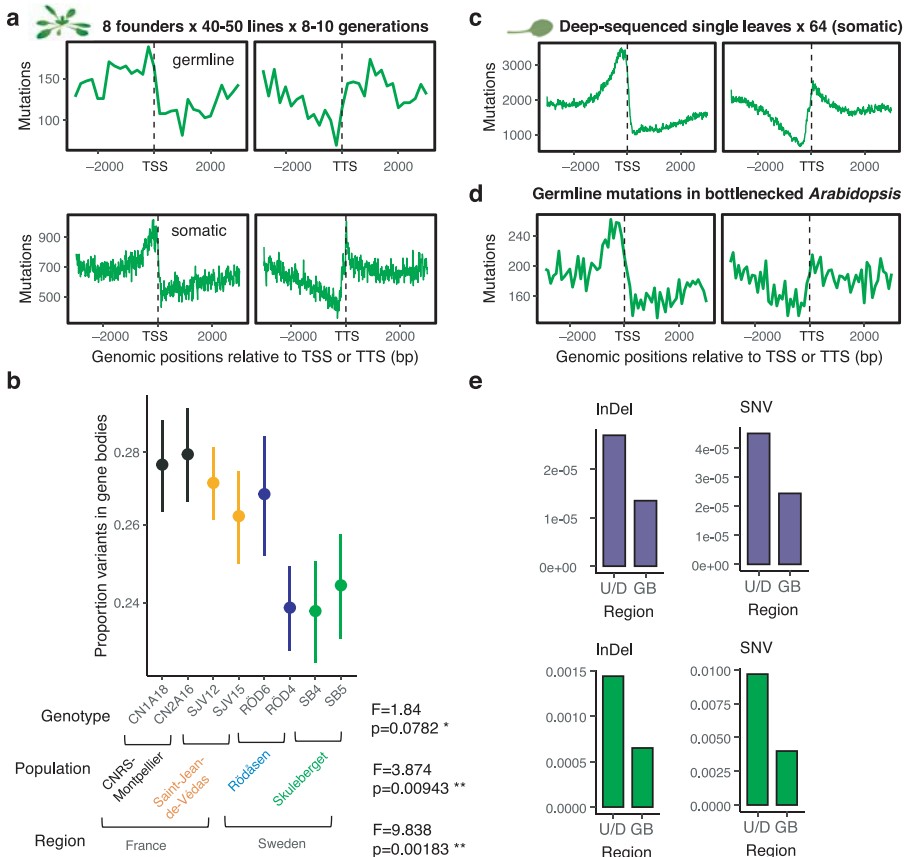

**Extended Data Fig. 4 | Variants called in additional mutation accumulation datasets. a**, Germline and somatic mutations around gene regions in mutation accumulation lines derived from eight founder genotypes. For each founder, 35-60 lines were propagated for 8-10 generations. **b**, The proportion of somatic variants detected in gene bodies (gene body/(gene body + upstream + downstream)) among descendants of the same founder. F- and p-values from one-way ANOVA. (n = 400 unique mutation accumulation lines, mean ± 2 s.e.m.) **c**, Somatic variants detected from reanalysis of 64 individual leaves from two Col-0 plants[21]. **d**, Germline variants detected in a bottlenecked *A. thaliana* lineage following colonization of North America since -1600 (ref. [32]). **e**, Epigenome-predicted and observed mutation rates across all datasets for InDels and SNVs, comparing gene bodies (GB) with upstream/downstream (U/D) regions. P-values from chi-squared tests.

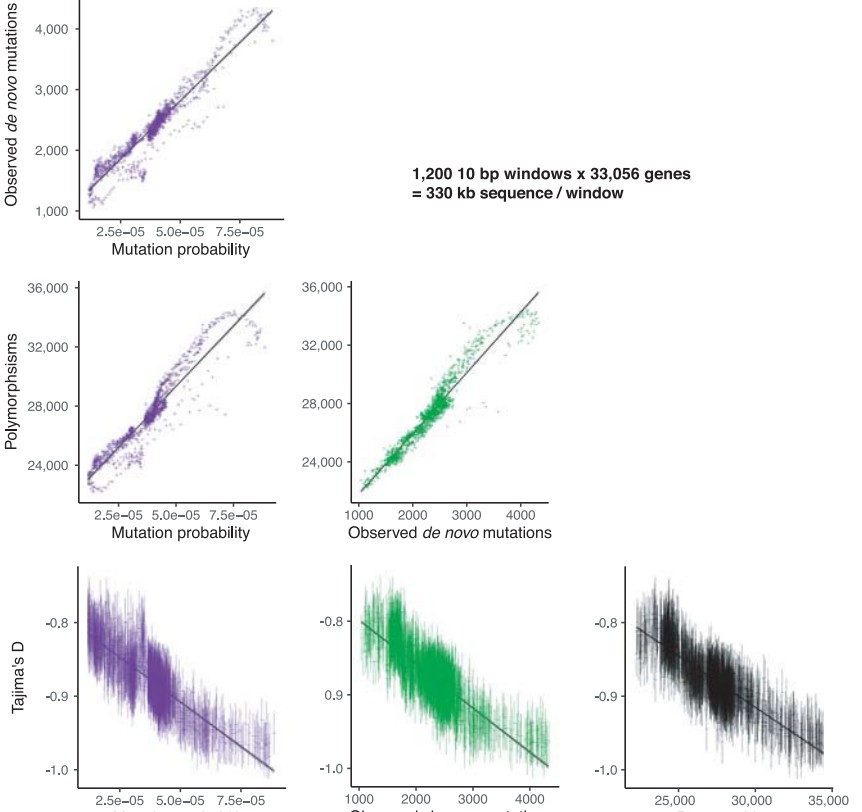

1,200 10 bp windows x 33,056 genes
= 330 kb sequence / window

**Extended Data Fig. 5 | Relationships between epigenome-predicted mutation probability, observed *de novo* mutations, polymorphisms in natural populations, and Tajima's D in natural populations.** These data show the quantitative relationships apparent in Fig. 2 of the main text. Each point reflects the value in one window of 1,200 calculated windows across all 33,056 genes, in relation to genome-wide transcription start and termination sites (TSS, TTS). Error bars indicate ±2 s.e.m. confidence intervals. For epigenome-predicted mutation probability scores, each point reflects the mean ±2 s.e.m. across all genes. For observed *de novo* mutations, each point reflects the total number of mutations ±2 s.e.m. (bootstrapped). For polymorphisms, each point reflects the total number of variants ±2 s.e.m. (bootstrapped). For Tajima's D, each point reflects the mean ±2 s.e.m. Tajima's D is already predicted by existing theory to be negatively correlated with mutation rate, as regions with higher mutation rate will be enriched for newer and therefore rarer variants.

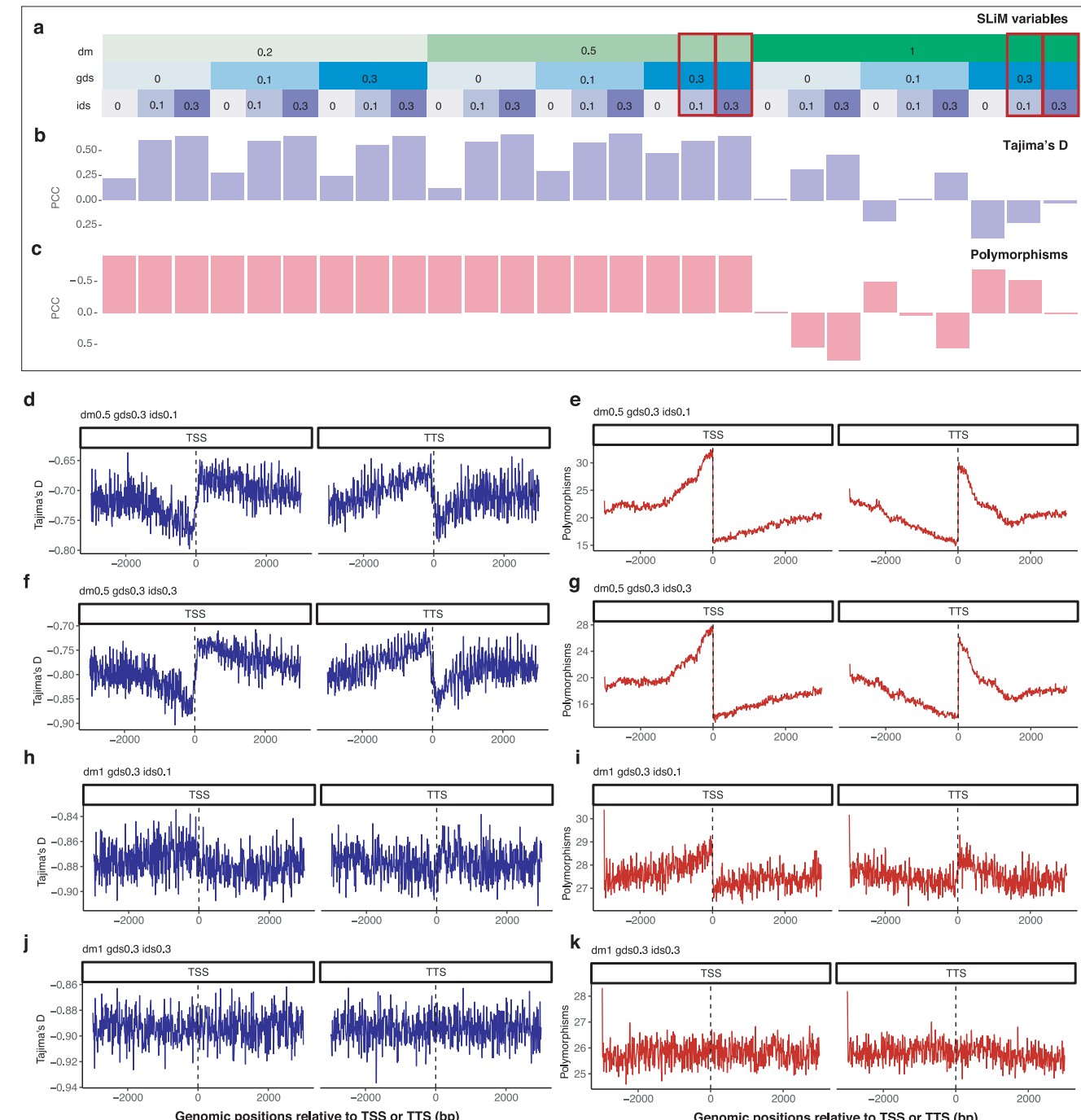

**Extended Data Fig. 6 | Effects of mutation rate and selection heterogeneity on polymorphisms and Tajima's D along genes.** Simulation results from SLiM[83] for the first 100 genes of chromosome 1 using a population of 1,000 individuals and 10,000 generations. **a**–**c**, Average correlation between 200 permutations of simulated scenarios and observed patterns of variation in natural *A. thaliana* accessions. **a**, Parameter choice: Difference between mutation rate (dm) in gene bodies and intergenic space (e.g., 0.5 = 50% reduction in mutation rate) and proportion of mutations that are deleterious in the genic (gds) and intergenic (ids) regions. The parameter combinations shown in **d**–**h** are highlighted with red outlines. **b**, Pearson correlation coefficients (ppc) comparing Tajima's D values from each simulation to that of observed data in natural *A. thaliana* accession. **c**, Pearson correlation coefficients (pcc) comparing number of polymorphisms accumulated in each simulation to that of observed data in wild *Arabidopsis* accessions. **d**–**k**, Examples of polymorphism (red) and Tajima's D (blue) in relation to gene bodies (TSS, TTS) averaged from 200 permutations of a scenario approximating empirical estimates of mutation rate heterogeneity and selection heterogeneity between gene bodies and intergenic space. Parameters (see **a**) given for each scenario. Strong purifying selection in gene regions alone (with equal mutation rates between gene bodies and intergenic space), which also reduces levels of polymorphism in gene bodies, causes more negative Tajima's D values in gene bodies, which is inconsistent with observed data in natural *A. thaliana* accessions.

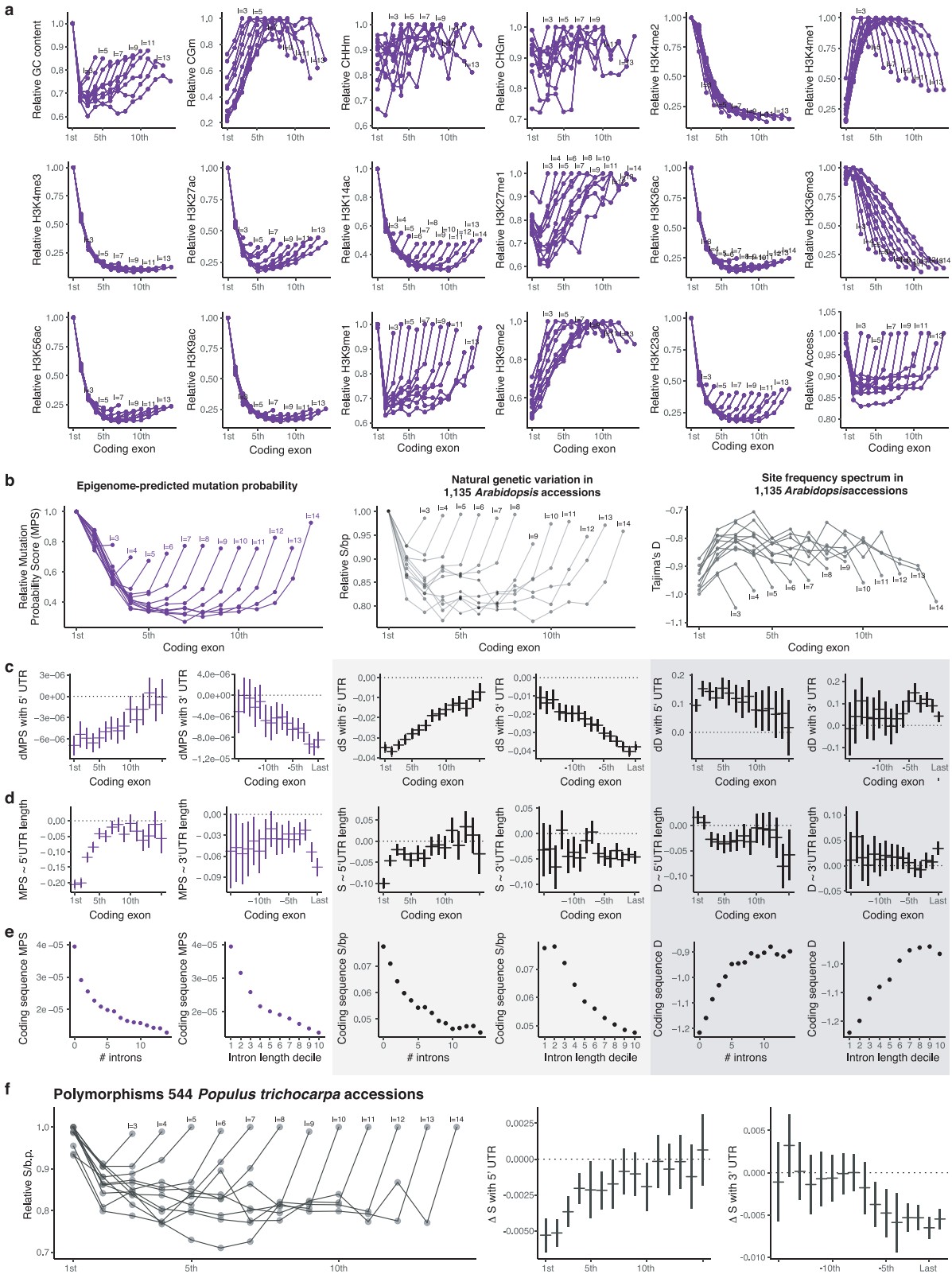

**Extended Data Fig. 7** | See next page for caption.

**Extended Data Fig. 7 | Relationships between untranslated regions or introns and mutation rates. a**, Distribution of epigenomic features in genes with different numbers of exons. **b**, Epigenome-predicted Mutation Probability Score (MPS), rates of natural polymorphism, and Tajima's D in genes with different numbers of exons. **c**, Left: comparison of Mutation Probability Score (MPS) between genes with UTRs and those lacking 5' or 3' UTRs. Horizontal lines mark the mean difference between genes with and without UTRs. Vertical lines mark the mean +- confidence intervals of two-sided t-tests. Center: rates of natural polymorphism in natural *A. thaliana* accessions. Right: Tajima's D in natural accessions. (n = 35,526 gene models). **d**, Left: Pearson's correlation coefficients for relationship between predicted mutation probabilities and the absolute length of 5' and 3' UTRs. Horizontal lines mark the means, and vertical lines mark the mean +- confidence intervals. Center: same for rates of natural polymorphism in natural accessions. Right: same for Tajima's D in natural accessions. (n = 35,526 gene models). **e**, Left: Relationships between intron number and total intron length with predicted mutation probabilities. Points indicate mean values. Center: same for rates of natural polymorphism in natural accessions. Right: same for Tajima's D in natural accessions. **f**, Results for 544 *Populus trichocarpa* accessions[75]. Horizontal lines mark the means, and vertical lines mark the mean ± confidence intervals of two-sided t-tests (n = 73,013 gene models).

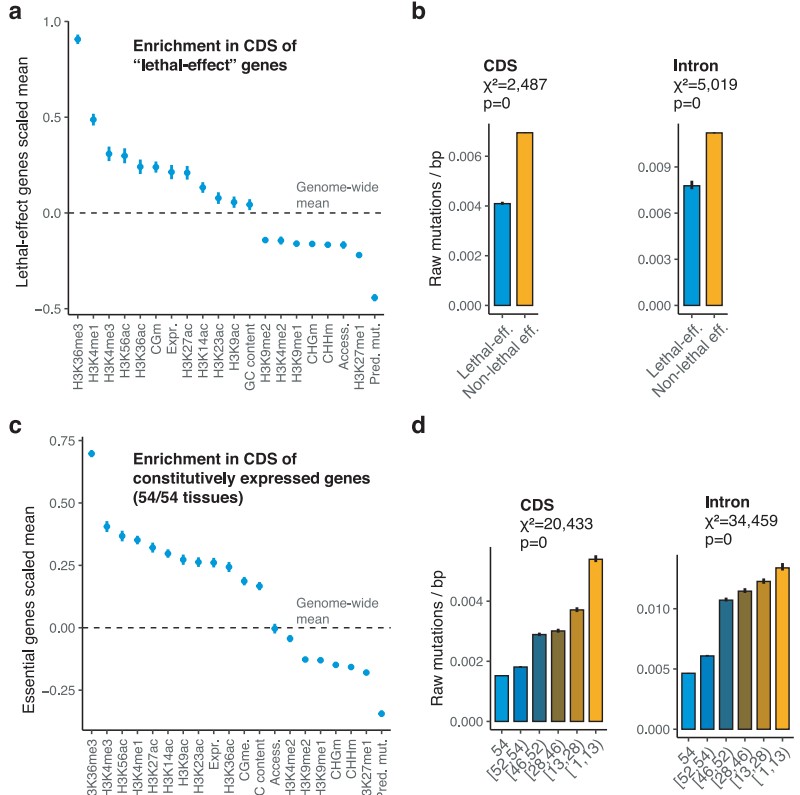

**Extended Data Fig. 8 | Epigenomic and other features and mutation rates of lethal-effect and constitutively expressed genes. a**, Enrichment of features in coding sequences of "lethal-effect" genes (n = 2,720 lethal-effect genes, mean ±2 s.e.m.). **b**, Total mutation rate (±2 s.e.m., bootstrapped) in lethal- and non-lethal-effect genes (n = 27,206 genes). **c**, Enrichment of features in coding sequences of constitutively (across all tissues) expressed genes (n = 9,957 genes, mean ± 2 s.e.m.). **d**, Total mutation rate (±2 s.e.m., bootstrapped) in genes binned according to the number of tissues in which they are expressed (n = 25,987 genes with tissue-specific expression data)[39].

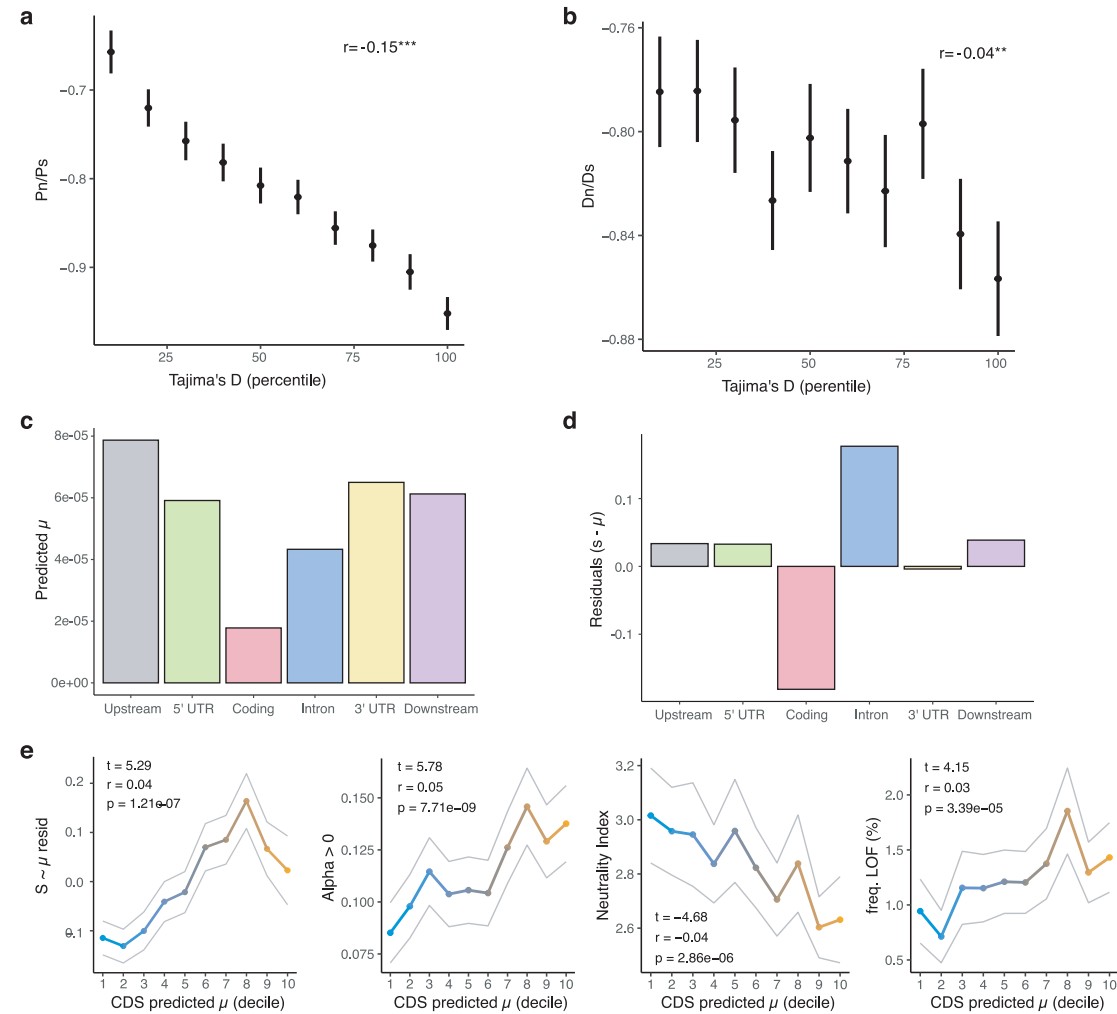

**Extended Data Fig. 9 | Predicted mutation rates and evidence of selection on natural polymorphisms versus *de novo* mutations across gene regions.** **a**, Relationship between Tajima's D of gene bodies and coding region selection estimated by $P_n/P_s$ (n = 21,407 genes) and **b**, $D_n/D_s$ (n = 21,407 genes). **c**, Epigenome-predicted mutation probability in different gene features. **d**, Scaled residuals ((Obs-Pred)/Pred) from S ~ u. Significantly negative residuals in coding regions are consistent with purifying selection in natural populations acting on new mutations. **e**, Relationships between epigenome-predicted mutation probability and other estimates of constraint. Residuals between predicted mutation rate and observed mutations are positively correlated with predicted mutation rate indicating that genes subject to purifying selection are predicted to mutate less. Genes with low predicted mutation rates are also less likely to have *alpha* > 0, a measure of variants under positive selection. Genes with low predicted mutation rate are depleted in natural populations for non-synonymous variants that reach fixation, as measured by the Neutrality Index, and for loss-of-function variants.

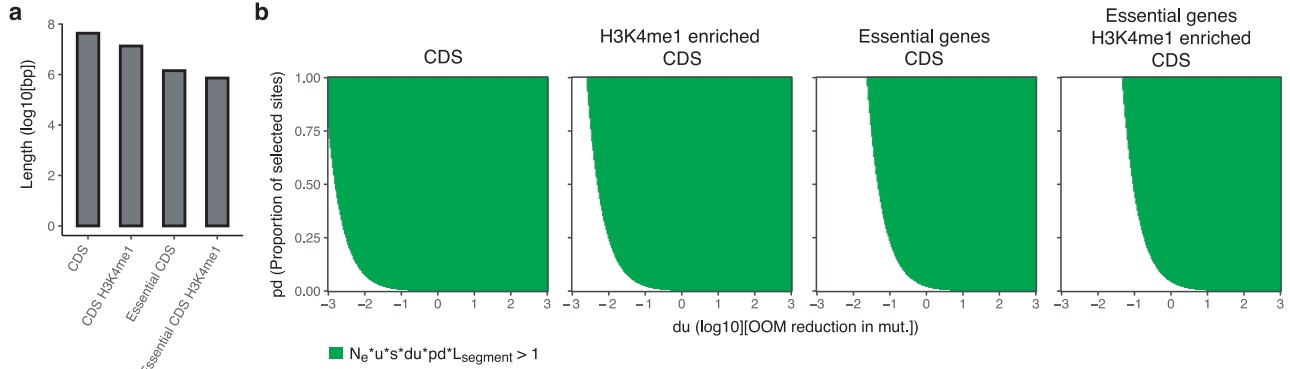

**Extended Data Fig. 10 | Estimates of L_segment for different regions. a**, Length of sequence space (L_segment) reflecting different types of regions. **b**, Test of parameter space that satisfies population genetic theoretical predictions for the possibility for targeted hypomutation to evolve. OOM = orders of magnitude. Selection on intragenomic mutation rate variation will be effective[5,40] when $N_e*u*s*du*pd*L_{segment} > 1$ where $N_e$ is the effective population size, $u$ is the mutation rate, $s$ is the average selection coefficient on deleterious mutations, $du$ is the degree of change in mutation rate, $pd$ is the proportion of sites subject to purifying selection, and $L_{segment}$ is the region of the genome affected. Assuming an effective population size of ~300,000 (ref. [86–88]), a mutation rate of ~10⁻⁸ (ref. [13]), an average selection coefficient of 0.01 (ref. [5]), an order-of-magnitude reduction in mutation rate[5], and functionally constrained regions where 20% of sites are under selection[5], the total length of the sequence affected, $L_{segment}$, would have to be at least ~200 kb, which (accounting for differences in effective population size) is similar to previous estimates in humans[5]. For perspective, this minimum $L_{segment}$ is considerably shorter (~1.5%) than the sum of coding regions with elevated levels of H3K4me1 (top quartile is ~13 Mb, or 15% of the genome), a feature enriched in gene bodies and essential genes and associated with lower mutation rate. Thus, selection is expected to act with high efficiency on variants that cause DNA repair and protection mechanisms to preferentially target such regions.

# Reporting Summary

Nature Research wishes to improve the reproducibility of the work that we publish. This form provides structure for consistency and transparency in reporting. For further information on Nature Research policies, see our Editorial Policies and the Editorial Policy Checklist.

## Statistics

For all statistical analyses, confirm that the following items are present in the figure legend, table legend, main text, or Methods section.

| n/a | Confirmed | |
|---|---|---|
| ☐ | ☒ | The exact sample size (*n*) for each experimental group/condition, given as a discrete number and unit of measurement |
| ☐ | ☒ | A statement on whether measurements were taken from distinct samples or whether the same sample was measured repeatedly |
| ☐ | ☒ | The statistical test(s) used AND whether they are one- or two-sided<br>*Only common tests should be described solely by name; describe more complex techniques in the Methods section.* |
| ☐ | ☒ | A description of all covariates tested |
| ☐ | ☒ | A description of any assumptions or corrections, such as tests of normality and adjustment for multiple comparisons |
| ☐ | ☒ | A full description of the statistical parameters including central tendency (e.g. means) or other basic estimates (e.g. regression coefficient) AND variation (e.g. standard deviation) or associated estimates of uncertainty (e.g. confidence intervals) |
| ☐ | ☒ | For null hypothesis testing, the test statistic (e.g. *F*, *t*, *r*) with confidence intervals, effect sizes, degrees of freedom and *P* value noted<br>*Give P values as exact values whenever suitable.* |
| ☒ | ☐ | For Bayesian analysis, information on the choice of priors and Markov chain Monte Carlo settings |
| ☒ | ☐ | For hierarchical and complex designs, identification of the appropriate level for tests and full reporting of outcomes |
| ☐ | ☒ | Estimates of effect sizes (e.g. Cohen's *d*, Pearson's *r*), indicating how they were calculated |

*Our web collection on statistics for biologists contains articles on many of the points above.*

## Software and code

Policy information about availability of computer code

| Data collection | SLiM v3 |
|---|---|
| Data analysis | R v4.0.2;  cutadapt v2.3; samtools markdup v1.10; GATK HaplotypeCaller v4.1.0, ART-MountRainier-2016-06-05, bwa mem v2,  vcftools 0.1.13, SLiM v3, MACS2, mkTest.rb v1, |

For manuscripts utilizing custom algorithms or software that are central to the research but not yet described in published literature, software must be made available to editors and reviewers. We strongly encourage code deposition in a community repository (e.g. GitHub). See the Nature Research guidelines for submitting code & software for further information.

## Data

Policy information about availability of data

All manuscripts must include a data availability statement. This statement should provide the following information, where applicable:
- Accession codes, unique identifiers, or web links for publicly available datasets
- A list of figures that have associated raw data
- A description of any restrictions on data availability

Data Availability
Data are available in the Supplementary Tables and Supplementary Data. Mutation data used as our training set are deposited on Figshare (https://doi.org/10.25386/genetics.6456065). Raw Illumina sequencing reads from additional mutation accumulation experiments are available via NCBI SRA. Genomes of natural Arabidopsis thaliana accessions are available at http://1001genomes.org/data/GMI-MPI/releases/v3.1/ . Chromatin state data are available through the Plant Chromatin State Database (http://systemsbiology.cau.edu.cn/chromstates). Raw reads from the ATAC-seq experiments are available via NCBI SRA.  Tissue-specific expression data are available at https://www.ebi.ac.uk/arrayexpress/experiments/E-MTAB-7978/. There are no restrictions on the availability of data used

in this study.

Code availability
Annotated scripts of code used in this study will be made available on Github (https://github.com/greymonroe).

# Field-specific reporting

Please select the one below that is the best fit for your research. If you are not sure, read the appropriate sections before making your selection.

[✗] Life sciences          [ ] Behavioural & social sciences          [ ] Ecological, evolutionary & environmental sciences

For a reference copy of the document with all sections, see nature.com/documents/nr-reporting-summary-flat.pdf

# Life sciences study design

All studies must disclose on these points even when the disclosure is negative.

| | |
|---|---|
| Sample size | Sample sizes of 107 (original mutation accumulation experiment), 50+ (second mutation accumulation experiment, reflects number of lines per founder genotype), 64 number of individual sequenced leaves to detect somatic mutation) , and 10 (number of siblings of one original mutation accumulation line) Individuals for calling de novo mutations are limited primarily be sequencing costs and the number measured was deemed sufficient as shared variants are expected to be readily identifiable. |
| Data exclusions | Sequence variants were excluded that failed filtering thresholds based on quantitative estimates of quality or confidence. |
| Replication | We reproduced our finding that mutation rates are elevated upstream and downstream of transcribed regions with additional sequencing of mutation accumulation line siblings (repeated once). We further replicated our results with analyses of somatic and germline mutation distributions in additional datasets (repeated twice). The first of these was the largest mutation accumulation dataset conducted to date. The second of these was a reanalysis of somatic variants detected in single Arabidopsis leaves. We reproduced the observation of elevated polymorphism rates in 3` and 5` coding regions with analysis of an outgroup species, Populus trichocarpa (repeated once). |
| Randomization | Mutation accumulation lines were randomized for growing conditions (i.e. position on tray). |
| Blinding | Investigators were blinded to the location of mutations during variant filtering. |

# Reporting for specific materials, systems and methods

We require information from authors about some types of materials, experimental systems and methods used in many studies. Here, indicate whether each material, system or method listed is relevant to your study. If you are not sure if a list item applies to your research, read the appropriate section before selecting a response.

## Materials & experimental systems

| n/a | Involved in the study |
|---|---|
| [✗] | Antibodies |
| [✗] | Eukaryotic cell lines |
| [✗] | Palaeontology and archaeology |
| [✗] | Animals and other organisms |
| [✗] | Human research participants |
| [✗] | Clinical data |
| [✗] | Dual use research of concern |

## Methods

| n/a | Involved in the study |
|---|---|
| [✗] | ChIP-seq |
| [✗] | Flow cytometry |
| [✗] | MRI-based neuroimaging |

