## [Peer Review File · Nature]

Manuscript Title: Mutation bias reflects natural selection in *Arabidopsis thaliana*

Reviewer Comments & Author Rebuttals

Reviewer Reports on the Initial Version:

Referee #1 (Remarks to the Author):

This paper describes a thorough and systematic analysis of mutation rates in a model system *Arabidopsis thaliana* with the goal of examining what genomic and organismal features correlate with mutation rate. Using mutation accumulation lines, the authors are able to show that mutation rates are dependent on several factors that may result in bias.

These results are very significant as they could have profound evolutionary implications. The fact that there is mutation bias is not surprising. What is important in their results is that the bias in mutation rates can have adaptive consequences – for example, if certain types of exons are more subject to mutation. Of even more significant interest is that mutation rates appear to be biased based on histone modifications and chromatin accessibility. Since these two features are connected to gene function (in a general way), these would lead to a bias in mutation rates that are correlated with a functional feature – which in turn would lead to evolutionary bias. This is in contradiction to the Darwinian paradigm of random mutation independent of function (although this does not invalidate Darwinian evolution, but adds a wrinkle to the process).

The only issue I would like to raise is the mutation calling based on their sequencing. In a sense everything rests on their ability to identify mutations above the inevitable noise level that arises in sequencing. Although I do not think this is an issue given that such noise would be expected to result in random distributions of mutations (and their results emphatically say they do not) it would still be nice to have a little more detail in the main text on how they filter out possible errors. In the current version they just say in the main text “Strict filtering was used to eliminate false positives – fewer than 10% of all called variants were included in the final high confidence set of de novo mutations.” It would I think improve the paper for readers if they expand a little more on this in the main text.

Referee #2 (Remarks to the Author):

Monroe et al. study gene-level variation of de novo mutations in advanced generation A. *thaliana* MA lines. They find that local mutation rates vary within genes (+/- some upstream region) and that this variation can be predicted from local sequence and epigenomic features. They further show that genic mutation patterns in the MA lines correlate with polymorphism patterns in natural populations, suggesting that patterns of natural genetic variation are to some extent the result of “mutational variation” rather than of selection. Finally, the authors show that evolutionary constrained genes display lower mutation rates. They use this latter observation to suggest that these lower rates are a way to avoid putative deleterious mutations in those constraint genes and therefore constitute what they call “adaptive mutation bias”.

I think that the model predictions are very interesting and do advance our understanding of the molecular determinants of local mutation rate, although many of the associations between predictors and local rates confirm what has been found in previous studies (as they also point out). I also think their work provide new insights into the mechanisms that generate natural genetic variation over evolutionary time-scales.

However, I do not think that the study makes a very strong case for what the authors call “adaptive mutation bias”, which – in my view - undermines the key message and novelty of their paper (see below points). Only because a rate reduction in evolutionary constrained genes may lead to decreased probabilities of deleterious mutations, does not immediately imply that mutations in hypermutable regions (e.g. at 5’ and 3’ ends of genes) are adaptive in anyway. There is no direct demonstration of this. That mutation rates (or substitution rates) are locus-specific is already routinely used in phylogenetic inference methods; and does not in itself present something substantially novel.

Below I am listing several specific concerns:

1. The authors re-analyze de novo mutations (SNV and indels) in 107 advanced generation Mutation Accumulation (MA) lines. Using a new filtering approach, which also includes the detection of somatic mutations, they obtain about 4 times more mutations than previously reported. It is unclear how/if this boost in detected de novo mutations translates to estimates of genome-wide mutation rates. Fig. 2D (y-axis) seems to indicate that the predicted rate in genes is in the range of 10^{-5} which would be four orders of magnitude higher than previous estimates of the global rate in *A. thaliana*. Similarly predicted rates in TEs in their re-analysis of the epiRIL panel seem to be in the order of 10^{-5} . Although the authors claim that false positive calls had been controlled, it is useful to compare the rates obtained with their new calling approach with previous reports (even if the primary interest of the paper is on relative rates along the genome).
2. To predict gene-level mutation rates the authors employ a multiple regression approach where the detected de novo mutation rate per base pair is used as the response variable and local sequence and epigenomic features as predictors. Very little information is given in the main text and methods section as to how this model is actually set up (e.g. what is the distribution of the response variable, what is the link-function of the generalized linear model). It is for example also not unclear how the response variable (i.e. the rate) is actually calculated. From my understanding the authors do the following: For a given target sequence, say a 5’ UTR in a given gene, they count up the detected de novo mutations among the 107 lines at the last sampled generation and divide that number by the total number of bases in that UTR. If I understand that correctly, we would be looking for mutation events in a space of $107 \times \text{length of UTR} \times \text{generation}$ to get a rate estimate per base-pair/per generation in that UTR. My intuition tells me that with the a mutation rate of 10^{-9} (if this rate is still accurate in that study, see comment above) it is very unlikely to see any event in a space given by $107 \times \text{length of UTR} \times \text{generation}$. This in turn would mean that the response variable probably look like a zero inflated random variable with a long right tail. The distribution of that variable will determine what link function to use, or what transformation, if any. More information should be provided.
3. The authors implement a stepwise model selection procedure to arrive at a sparse predictor set. Models are selected based on their AIC values, which includes a penalty proportional to the number of parameters. The authors claim that “to avoid over fitting caused by correlated predictor variables, we selected models (separately for SNVs and InDels) with the lowest Akaike information criterion (AIC) value.” However, this statement is not quite correct. The stepwise approach guards neither against overfitting nor against multicollinearity. In fact, both can be a major issue in that procedure, and there is extensive literature on this. Since epigenomic modifications and sequence features often correlate strongly across the genome, because of their combinatorial occurrence, multicollinearity is most likely an issue in model fitting. One approach is to assess multicollinearity by inspecting the Variance Inflation Factors (VIF) of the predictors and removing those with $VIF > 3$. A complimentary option is to perform model selection on a subset of the data and to assess the predictive ability of the model in another set of the data. I am raising these points because the primary aim of their study is to not only obtain a predictive model but also to obtain accurate estimates of the contributions of the individual predictors, as they reveal important biological insights about mutation rate control. With multicollinearity it is known that predictor estimates can be highly biased. In fact, predictors can be selected into the model without having any significant

marginal effect on the response variable. These points should be carefully explored.

4. One of their genetic mutation rate predictors is local cytosine methylation. One mechanism through which this can occur is through spontaneous cytosine deamination. To support this point the authors re-analyzed sequencing data from *ddm1*-derived *A. thaliana* epiRILs, which segregate methylated (wt-derived) and unmethylated (*ddm1*-derived) genomic segments. They show that transposable elements (TEs) in wt-inherited genomic segments have higher mutation rates than TEs in *ddm1*-derived genomic segments. However, extended Figure 5 seems to suggest that all rates are generally higher in wt-inherited regions, not just C→T mutation rates. In other words, the deamination argument does not seem to be supported. Additionally, the conclusions from their re-analysis conflict the results of Lauss et al. (2018). That latter study also estimated mutation rates in wt versus *ddm1* regions (not just in TEs but anywhere) and found a trend toward increased rates in *ddm1*-derived regions which they attributed to a loss of heterochromatin (and thus an increase in accessible regions).

This discrepancy may be related to one important technical consideration in estimating rates in those epiRILs: The wt segments that segregate in the epiRILs are derived from a different wt plant than the one that was used to generate the *ddm1* mutant. The *ddm1* mutation (in the Col-0 background) has been maintained by repeated backcrossing for many generations. The wt parent that was used to initiate the epiRILs was not immediately related to the *ddm1* parental plant. This means the Col-0 sequence backgrounds of these two parents are very different with respect to the Col-0 reference sequences. Mutations called with respect to the Col-0 reference may thus depend on the on very different laboratory histories of the Col-0 sequence of the wt plant and the *ddm1* parental plant (i.e. the wt parental plant could differ at many more sites from the Col-0 ref sequence, which could explain the increased rate for the wt segments). One way to bypass this is to focus on non-shared mutations, which are those that arose during inbreeding (in fact this is what Lauss et al. had done).

Finally, it is also not clear why the authors test their hypothesis about the effect of their predictor in TE sequences on not genetic sequences (*ddm1*-loss also effects genic sequences). Their model predictors were trained on genes. It is unclear how the authors expect that the relationship between cytosine methylation and mutation rates to also hold for TEs. In fact, patterns of C→T demethylation are much different in genes compared to TEs pointing toward a different predictor effect in these two very different genomic elements.

5. In Fig. 2E, the authors show that the distribution of natural polymorphisms are positively correlated with predicted mutation rates, which is consistent with natural variation in genic regions being substantially due to mutation bias rather than primarily due to the selection after random mutation. However, later they find that “in genes where mutation rate is predicted to be low, the observed number of polymorphisms tends to be even lower than that predicted by mutation rate alone, which is consistent with the action of purifying selection on natural polymorphisms rather than on de novo mutations. Similarly, coding regions have significantly negative residuals compared to other genic regions, supporting the action of purifying selection after mutation.” This means that – after all – the genic patterns are shaped by a mixture of mutation bias and selection, but it remains unclear how strong the role of selection is. The authors offer no way to quantify the contribution of selection on the polymorphisms patterns. Perhaps this could be achieved through a SFS analysis of different genic sectors. The question here would be if the selection coefficients also vary across the sectors in the manner that is consistent with the polymorphism patterns.

6. Fig. 3g., decreased rates in genes with higher number of introns, seems to partly conflict the prediction that lower rates are detected in evolutionary conserved genes, insofar that evolutionary conserved genes (at least those marked by gbM) are higher in exon content compared to other genes. The authors should explore this/comment on this.

7. The authors’ main support for their “adaptive mutation biased” hypothesis is summarized in Fig.

4. There they show some evidence that “Functionally constrained genes have lower predicted mutation rates”. They do this by correlating predicted mutation rates in coding regions (CDS) with various intra- and inter-specific measures of genetic variation (PS, PN, DS, DN, neutrality index, delta), and intra-specific variation in gene expression. These correlations are generally low, ranging from weak ($r=0.03$) to moderate ($r=0.26$); thus not lending much support to the “adaptive mutation bias” hypothesis. In addition, the way the data is presented is very misleading, in my view. The authors show plots where the mean of the above intra- and inter-specific measures of genetic variation is plotted for different deciles of predicted CDS mutation rates. These plots give the impression of a strong relationship. A fairer presentation would be to show a scatterplot of the two quantitative variables with the fitted regression line. Another option is to plot the 95% CI around the means (y-axis variable) for each deciles (x-axis variable). Either way will provide a visual presentation that is more consistent with the low/moderate correlation values.

8. In that same figure, the relationship between predicted CDS rate and the coefficients of variation in gene expression indicates that predicted hypermutable CDS (thus, more polymorphic CDS in natural populations) have a higher expression range. This relationship could simply be the consequences that these polymorphisms lead to expression variation via cis-acting eQTL, rather than hypermutability being the result of gene expression states. The authors should comment on this.

9. In that same figure, correlations are very similar between CDS and PS and PN (or DS and DN) suggesting that mutations of functionally neutral and putatively non-neutral mutations are nearly equally effected. This is consistent with previous observation that constraint loci tend to evolved more slowly in general. In fact the DN/DS ratio correlation is much lower suggesting again that correlation with measures of constrained evolution are weak.

Referee #3 (Remarks to the Author):

In this provocative manuscript, Monroe et al. argue that in *Arabidopsis thaliana*, local mutation rates have been selectively optimized such that they are lower in genes where mutations are more likely to be deleterious. If true, this finding would have very broad implications for evolutionary biology. Their observations are intriguing, but the analyses are somewhat convoluted, making it difficult to assess the strength of their conclusion. Their proposed mechanism of selection through genome-wide epigenetic marks is tantalizing but has some problems. It is difficult to prove that their observation results from selective optimization of mutation rates. My detailed comments follow.

Major comments

1. My main concern of the data analysis part is the use of predicted instead of actually observed mutation rates. They provide no evidence how good the prediction is. In fact, all of their analysis can use actual mutation rates, and it is unclear to me why they did not do that. For instance, a supergene analysis should be able to generate Fig. 2d and Fig. 3. Fig. 4 can be generated by considering 10 supergenes for the 10 deciles. I understand the purpose of showing the correlation between epigenetic marks and mutation rates, but this is about the potential selection mechanism, which is not needed for comparing trends of mutations and trends of polymorphisms/evolution. Their comparison between mutation and polymorphism/evolution patterns would be much more straightforward and convincing if they had used the actual mutations.
2. They interpreted their observation of correlations between predicted mutation rates and several gene expression or evolutionary constraint measures by selective optimization of mutation rates. I am not fully convinced by this interpretation and believe there are other possibilities.
(1) The formula in line 223 works only in the absence of recombination (e.g., see Kimura 1967 *Genet. Res.* or Lynch 2011 *Genome Biol. Evol.*). Because *A. thaliana* is selfing, it effectively has no recombination. So, one could argue that the formula works here. But this unusual reproductive mode evolved within the last million years in *A. thaliana*. For the above argument to work, one

must assume that all the optimizations of mutation rates occurred within the last million years in *A. thaliana*. If so, how do we explain the *Populus* results, as *Populus* is outcrossing and should not have the type of selective optimization of mutation rates?

(2) Another point about the formula in line 223 is that, under the assumption that there is a global mutation rate modifier, the drift barrier is determined by this formula when L_{segment} is the length of the whole genome (or functional genome) (Lynch 2011 *Genome Biol. Res.*). In other words, ΔU and U are tiny at the drift barrier. So, when L_{segment} is say 1% of the functional genome for a particular epigenetic feature, ΔU must be much bigger for selection to work. Given that U is already very low, it is impossible for ΔU to be large (you cannot reduce mutation rate to negative values). Thus, the authors' model is actually incompatible with the drift barrier hypothesis. I am not saying that their model is wrong because it is incompatible with the drift barrier hypothesis, as the drift barrier hypothesis could be wrong. But my reading is that the authors seem to be saying that their model is compatible with the drift barrier hypothesis. This needs to be clarified.

(3) If the observations in Fig. 4 are genuine, I wonder if they could be produced by non-adaptive processes. One possibility is that the correlations are intrinsic. For instance, methylated Cs have elevated mutation rates for chemical reasons. If somehow there is a correlation between the abundance of methylated Cs and gene expression purely for mechanistic reasons of gene expression, one may see a correlation between gene expression level and mutation rate that is not a result of selective optimization. Because dn/ds is strongly correlated with gene expression level in a wide variety of species including *A. thaliana* (Zhang and Yang 2015 *Nat. Rev. Genet.*), one would also see a correlation between dn/ds and mutation rate.

(4) Related to the above point, for the authors' model (i.e., selection acting on epigenetic marks) to work, there must be a significant correlation between epigenetic marks and gene expression/selective constraint, which the authors have not shown. If they can confirm this correlation, the non-adaptive process outlined above would easily work because all one need is an intrinsic relationship between epigenetic marks and mutation rate, which has been reported and cited in the manuscript. Thus, it seems that the condition necessary for the selective optimization hypothesis to work also makes the intrinsic property hypothesis work. In other words, the conclusion of adaptive mutation rate optimization cannot be proven.

(5) A related comment is on the interpretation of the negative correlation between mutation rate and UTR/intron length. Again, this negative correlation could be intrinsic instead of a result of selection.

3. Fig. 4, I did not find the GO analysis informative. Which function can be said to be unimportant? All functions are important, and designation of important/unimportant function is quite arbitrary. One less arbitrary index that the authors are encouraged to use is gene essentiality based on knock-down/out phenotypes, and there is such data for *A. thaliana* at the genomic scale. A key advantage of using gene essentiality is that it is an objective measure of functional importance yet it is not likely to be strongly correlated with gene expression level. This potentially allows one to distinguish the intrinsic property hypothesis from the adaptive optimization hypothesis.

Minor comments

1. Fig. 1b is large but contains little discernable information, so should be removed.
2. Fig. 1c. If my reading is correct, it shows that gene-poor regions (white parts in chromosomes) have lower mutation rates than gene-rich regions, directly contrasting the main conclusion of the paper. Please clarify.
3. Fig. 1d, it is difficult to sense the ratio after the log-transformation. While it is OK to use the log-scale, I suggest that they present untransformed values on the Y-axis for easy reading. My back-of-the-envelope calculation based on the numbers provided in the manuscript shows that the difference between "de novo" and "null" bars of non-syn/syn is significant instead of NS. Please check the statistics or provide the raw numbers used in the test.
4. Fig. 4a, how different is the mutation rate (in absolute terms) among the 10 deciles? Please provide the information in the figure. Because the theoretical model involves ΔU , it is critical to know how large the ΔU is.
5. Extended Data Fig. 6b. Negative values are interpreted as purifying selection. How about

positive values? Do they mean balancing selection? How likely is it that introns are subject to strong balancing selection?

Referee #4 (Remarks to the Author):

In their manuscript entitled "Adaptive mutation bias in *Arabidopsis thaliana*" by Grey Monroe et al have studied the determinant of mutation rate in a mutation accumulation experiment from *Arabidopsis thaliana*. They found some fluctuations of mutation rate along the genome and suggest that genes with higher level of constraints have lower mutation rate. This proposed reduced selection for mutation rate is supposed to be driven by genome wide factors that may generate a large enough benefit to select for a lower mutation rate.

My major concern with the present data is the number of cellular generation involved in the experimental design. One complexity of mutation accumulation is to ensure that no selection is occurring in the process. In bacterial experimental evolution, one simple tool is Ka/Ks ratio that can be used to investigate the lack of selection. We can push things even further with looking at predicted mutation effects using Direct Coupling Analysis as we did in Couce et al PNAS 2017 (<https://doi.org/10.1073/pnas.1705887114>). In these experiments a Ka/Ks equal to one is a simple proof that there is no selection detectable at the genome scale. Here the data are going in the good direction with a reduced selection, but without any more details on the null model used and the formula used to compute these ratios it is hard to be sure.

Indeed, a large fraction of the genome may show no Ka/Ks deviation as it is unconstrained, but the rest of the genome may lead to the absence of signal at that level despite a true selection that may later on affect the detection. In that respect the different formulation of the null model taking into account mutation biases has to be explicitly given as it may have a strong impact.

In that respect it would be worth doing simulations to see how much selection in selectively important genes in the MA process would be needed to lead to A) lead to a significant deviation in the overall KA/Ks, B) to generate the signature of selection detected.

An alternative could also be to take the genes ranked by their mutation rate as done in the last figure and to plot this time their Ka/Ks ratio for each of the deciles to see if there is a consistent association between low mutation rate and a signature of selection in the MA.

I am not sure about the meaning of the scaled values presented in figure 2. Indeed, throughout the text, the amplitude of the signals should be more precisely delineated. P values are not a proof of large effect. Are we talking about 10 fold differences? In Couce et al we use log-odds ratio for discrete traits and interquartile odds ratios for continuous ones. This is interesting but giving more explicitly the range of the observed signal as in figure 3 would be nice.

If there are enough synonymous mutations, analysis could be done using just these, or adding synonymous state to the model.

Experiments with bacteria revealed that a major driver of mutation rate is the bases before and after the focal base as well as the specificity of the base (<https://doi.org/10.1093/molbev/msv055>, <https://doi.org/10.1073/pnas.1705887114>). Here it seems that all these effects have been ignored. But how much of the signal could be due to these local sequence composition effects. As sequence specificities may vary from coding to non coding this can also explain some of the variability observed.

The p values between the changes in mutation and the mean per decile are meaningless. P values of any level can be obtained when performed on mean binned data. The rough p values and correlation should be given, even if the plot is made with the deciles for clarity and communication purposes. Again, how much signature of selection in the MA would be needed to explain some of these signals.

Besides these technical comments, the analysis performed are really interesting.
Is there any hypothesis on why mutation rate may be larger at TSS and TTSs sites precisely?
Secondary structure have been shown to affect mutation rate (Barbara Wright work
(<https://doi.org/10.1099/mic.0.2007/005470-0>) on bacteria that we followed in Hoede et al Plos
Genetics 2006). Could that be involved at these sites?
For all the histone methylation marks, is there a synthetic way to present their various
contribution: the ones linked to expression, the ones linked to deletion to acessibility... Apart
from one or two mentioned in the text, the others are just left to specialists for interpretation. So a
synthetic figure giving some insights would be great.

Olivier Tenaillon (I always sign my reviews)

Synopsis

We thank the reviewers for their very helpful feedback and recommendations - addressing these has led to new discoveries and substantially improved our work. We apologize for the length of the rebuttal, but given the extensive, thoughtful and detailed comments of the reviewers, we felt that an equally detailed and well considered set of responses is called for. We hope the reviewers agree.

To reviewer #1 and others, we are particularly grateful for the enthusiasm expressed for the transformative value of this set of discoveries. To further strengthen our manuscript, we have added new results from an as of yet unpublished dataset comprising the largest mutation accumulation experiment conducted to date in *Arabidopsis*. We use this new data to validate predictions made in our original submission and confirm that our model has excellent power to explain the distribution of new mutations in multiple independent datasets.

To reviewer #2, we are particularly grateful for the suggestion to investigate site frequency spectrum statistics in natural populations to tease apart the role of selection and mutation bias in driving the patterns we observe in levels of natural variation in wild populations. The revised manuscript now prominently includes multiple results coming directly from this recommendation - most strikingly, we found that Tajima's D is significantly higher in gene bodies and in internal exons, supporting the interpretation that reduced mutation bias, rather than negative selection alone, is largely responsible for observed lower levels of natural polymorphism in gene bodies.

To reviewer #3, we are particularly grateful for the suggested analyses to examine functionally explicit groups of genes, determined empirically through knockout lines, rather than GO analysis alone. This new line of investigation has shed light on the mechanistic link between gene function, mutation rate, and signatures of selection. Importantly, these new analyses confirm that epigenomic states are distributed non-randomly among genes (eg., lethal-effects vs non-lethal-effects of mutations in them) in a way that points to selection acting on interactions with DNA protection/repair, rather than these biases being an intrinsic property of transcription or cytosine methylation. With the new mutation datasets we also confirm that the predicted patterns can be directly observed in raw data.

To reviewer #4, we are particularly grateful for the suggested analyses to demonstrate that unexpected selection has no major effect on the distribution of detected *de novo* mutations. We have added new analyses and several simulations supporting the expectation that the observed mutations with which we train our model did not experience detectable selection. For example, we compared K_a/K_s in predicted mutation-sensitive vs -insensitive genes and show that they are the same despite sensitive genes having significantly lower raw mutation rates. We have also added simulations showing that the mutation biases we observe cannot reasonably be explained by purifying selection on coding region variants.

We hope that the reviewers will find these and other major changes in the revised manuscript as exciting as we do and that they satisfactorily address all major concerns.

Summary of major revisions (primary pertinent reviews indicated as R1, R2, R3, R4)

- (R1,R2,R3,R4) Validation of model for mutation probabilities based on epigenetic features by
 - adding unpublished data from the largest mutation accumulation experiment conducted to date in *Arabidopsis*; used these also to show that results are robust to genetic backgrounds (this resulted in five additional authors)
 - adding analysis of data from a somatic mutation experiment; used these to validate model predictions
 - adding reanalysis of data from germline mutations detected in a bottlenecked *Arabidopsis* population and used it to validate our model predictions
- (R2) Analyzed SFS statistics, Tajima's D and minor allele frequency in natural populations and now show that these support mutation bias as the likely explanation for patterns of gene evolution:
 - Tajima's D is more negative in upstream and downstream regions compared to gene bodies, mirroring differences in Mutation Probability Score
 - Tajima's D is more negative in peripheral exons, also mirroring differences in Mutation Probability Score
- (R3) Compiled functional categories of genes from published knockout experiments and added analyses of their epigenomic states emergent mutation probabilities:
 - Essential/lethal genes are enriched for features associated with low mutation probability, including H3K36me3 and H3K4me1
 - Essential genes are enriched for GC methylation, which indicates our results are not an intrinsic property related to high mutability of methylated cytosines in non-essential genes
 - Essential genes do not have higher mean expression, indicating that our results are not an intrinsic property related exclusively to transcription-coupled repair
 - Genes expressed in many tissues are enriched for features associated with low mutation probability, consistent with DNA repair being more likely to be targeted to epigenomic states associated with housekeeping genes
- (R4) Confirmed that results are not caused by selection having skewed detected mutations in the training set of the original data:
 - Comparison of nonsynonymous/synonymous ratios among different classes of genes indicates that potential selection on *de novo* mutations did not skew our results
 - Simulations show that our observed *de novo* mutation training dataset has 70% fewer synonymous mutations compared to random expectations
 - Simulations of dominant lethal mutations in coding regions show that our observed bias cannot be reasonably explained by unexpected selection on the *de novo* mutation dataset
- (R1,R2, R3,R4) Updated the manuscript to clarify the units of the predicted mutation variation around the genome, introducing a "Mutation Probability Score", to highlight that this is a measure of

epigenetic features that predict mutation probability and that what we present is different from “mutation (per generation) rate” as it integrates information from both germline and somatic mutations

- (R2) Explored models dropping predictors based on variance inflation factor (VIF) and found that results were effectively unchanged, indicating our conclusions can not be explained by covariance between predictors
- (R3) Reframed our discussion of the relationship between our results and existing theory, emphasizing the importance of our findings on L_{segment} , rather than dU , as the most important parameter of the drift-barrier hypothesis for our results
- (R4) Added a conceptual figure that summarizes the model of mutation bias revealed by our results
- Finally, changed the title to provide a less ambiguous synopsis of our discovery

Referee #1 (Remarks to the Author):

This paper describes a thorough and systematic analysis of mutation rates in a model system *Arabidopsis thaliana* with the goal of examining what genomic and organismal features correlate with mutation rate. Using mutation accumulation lines, the authors are able to show that mutation rates are dependent on several factors that may result in bias.

These results are very significant as they could have profound evolutionary implications. The fact that there is mutation bias is not surprising. What is important in their results is that the bias in mutation rates can have adaptive consequences – for example, if certain types of exons are more subject to mutation. Of even more significant interest is that mutation rates appear to be biased based on histone modifications and chromatin accessibility. Since these two features are connected to gene function (in a general way), these would lead to a bias in mutation rates that are correlated with a functional feature – which in turn would lead to evolutionary bias. This is in contradiction to the Darwinian paradigm of random mutation independent of function (although this does not invalidate Darwinian evolution, but adds a wrinkle to the process).

The only issue I would like to raise is the mutation calling based on their sequencing. In a sense everything rests on their ability to identify mutations above the inevitable noise level that arises in sequencing. Although I do not think this is an issue given that such noise would be expected to result in random distributions of mutations (and their results emphatically say they do not) it would still be nice to have a little more detail in the main text on how they filter out possible errors. In the current version they just say in the main text “Strict filtering was used to eliminate false positives – fewer than 10% of all called variants were included in the final high confidence set of de novo mutations.” It would I think improve the paper for readers if they expand a little more on this in the main text.

Thank you for the helpful recommendation - we agree the variant calling is quite important here. We have added more details to the main text to help readers understand what we mean by filtering without necessitating a detailed look at the methods and extended data. We hope that this helps the flow and clarity of the revised manuscript.

Referee #2 (Remarks to the Author):

Monroe et al. study gene-level variation of de novo mutations in advanced generation A. *thaliana* MA lines. They find that local mutation rates vary within genes (+/- some upstream region) and that this variation can be predicted from local sequence and epigenomic features. They further show that genic mutation patterns in the MA lines correlate with polymorphism patterns in natural populations, suggesting that patterns of natural genetic variation are to some extent the result of “mutational variation” rather than of selection. Finally, the authors show that evolutionary constrained genes display lower mutation rates. They use this latter observation to suggest that these lower rates are a way to avoid putative deleterious mutations in those constraint genes and therefore constitute what they call “adaptive mutation bias”.

I think that the model predictions are very interesting and do advance our understanding of the molecular determinants of local mutation rate, although many of the associations between predictors and local rates confirm what has been found in previous studies (as they also point out). I also think their work

provide new insights into the mechanisms that generate natural genetic variation over evolutionary time-scales.

However, I do not think that the study makes a very strong case for what the authors call “adaptive mutation bias”, which – in my view - undermines the key message and novelty of their paper (see below points). Only because a rate reduction in evolutionarily constrained genes may lead to decreased probabilities of deleterious mutations, does not immediately imply that mutations in hypermutable regions (e.g. at 5' and 3' ends of genes) are adaptive in anyway. There is no direct demonstration of this. That mutation rates (or substitution rates) are locus-specific is already routinely used in phylogenetic inference methods; and does not in itself present something substantially novel.

Thank you for raising this issue. We believe that our results do demonstrate that the bias in mutation we observe has the potential to be adaptive in that this bias would increase fitness because it reduces the frequency of deleterious mutation, as in adaptive *Mutation Bias*. This is a different phenomenon than *Adaptive Mutation* bias. The latter reading could be interpreted as a claim that bias exists to target mutations to regions where variation is particularly advantageous (for example, immunity genes), or targeted in a way that is responsive to the environment. We have not made such an assertion for our system, but rather use a definition of adaptive mutation bias as the adaptive (evolutionarily advantageous) reduction in deleterious mutations through genome-wide mechanisms that reduce mutation probability at evolutionarily constrained loci.

We have realized that this subtle but critical distinction could be confusing to readers and have therefore decided to change the title of the paper accordingly. While the novelty of our discovery is the same - in contrast to traditional assumptions, mutation bias appears to be adaptive in that it reduces the frequency of deleterious variants - we hope this new title provides a less ambiguous description of our discovery. In summary, while we do propose that the bias we observe is adaptive, we can see how placing the words “adaptive” and “mutation” next to each other in the title might be unclear as to the precise meaning. Therefore, we have changed it to “Mutation bias reflects natural selection in *Arabidopsis thaliana*”.

We are open to suggestions for alternative titles. As you might imagine, the title has been a topic of much discussion among the authors. We know this paper may be received as provocative so we seek a title that best captures the weight of the results in a way consistent with the interpretation. Again: Suggestions are more than welcome!

Below I am listing several specific concerns:

1. The authors re-analyze de novo mutations (SNV and indels) in 107 advanced generation Mutation Accumulation (MA) lines. Using a new filtering approach, which also includes the detection of somatic mutations, they obtain about 4 times more mutations than previously reported. It is unclear how/if this boost in detected de novo mutations translates to estimates of genome-wide mutation rates. Fig. 2D (y-axis) seems to indicate that the predicted rate in genes is in the range of 10^{-5} which would be four orders of magnitude higher than previous estimates of the global rate in *A. thaliana*. Similarly predicted rates in TEs in their re-analysis of the epiRIL panel seem to be in the order of 10^{-5} . Although the authors claim that false positive calls had been controlled, it is useful to compare the rates obtained with their new

calling approach with previous reports (even if the primary interest of the paper is on relative rates along the genome).

Thank you for the thoughtful comments. We have updated the revised manuscript to more clearly describe the units in our results. These numbers indicate the predicted mutation probability (rather than rate with generation in the denominator). This is why the values are different than typical per-generation mutation rates. These higher values reflect that we used somatic mutations for higher power in our dataset to study the distribution of mutation across the genome. Importantly, as you point out, our aim is to study the relative, not the absolute differences in mutation probability throughout the genome. In the revised manuscript we have added new datasets to show that the relative differences in mutation probability in functionally distinct regions of the genome predict observed differences in additional *de novo* mutation datasets, including ones that include only germline or somatic mutations.

2. To predict gene-level mutation rates the authors employ a multiple regression approach where the detected *de novo* mutation rate per base pair is used as the response variable and local sequence and epigenomic features as predictors. Very little information is given in the main text and methods section as to how this model is actually set up (e.g. what is the distribution of the response variable, what is the link-function of the generalized linear model). It is for example also not unclear how the response variable (i.e. the rate) is actually calculated. From my understanding the authors do the following: For a given target sequence, say a 5' UTR in a given gene, they count up the detected *de novo* mutations among the 107 lines at the last sampled generation and divide that number by the total number of bases in that UTR. If I understand that correctly, we would be looking for mutation events in a space of $107 \times \text{length of UTR} \times \# \text{generation}$ to get a rate estimate per base-pair/per

generation in that UTR. My intuition tells me that with the a mutation rate of 10^{-9} (if this rate is still accurate in that study, see comment above) it is very unlikely to see any event in a space given by $107 \times \text{length of UTR} \times \# \text{generation}$. This in turn would mean that the response variable probably look like a zero inflated random variable with a long right tail. The distribution of that variable will determine what link function to use, or what transformation, if any. More information should be provided.

Thank you for the comments. We agree and have updated the revised manuscript to more thoroughly describe our model, which as you accurately point out, is a simple multiple linear regression. We also present results from untransformed data. As we show with multiple cross validation efforts using new, independent mutation accumulation datasets and polymorphism in natural populations, the resulting model has remarkable power to predict the distribution of new mutations in functionally distinct regions (i.e., gene bodies vs. promoters).

3. The authors implement a stepwise model selection procedure to arrive at a sparse predictor set. Models are selected based on their AIC values, which includes a penalty proportional to the number of parameters. The authors claim that “to avoid over fitting caused by correlated predictor variables, we selected models (separately for SNVs and InDels) with the lowest Akaike information criterion (AIC) value.” However, this statement is not quite correct. The stepwise approach guards neither against overfitting nor against multicollinearity. In fact, both can be a major issue in that procedure, and there is

extensive literature on this. Since epigenomic modifications and sequence features often correlate strongly across the genome, because of their combinatorial occurrence, multicollinearity is most likely an issue in model fitting. One approach is to assess multicollinearity by inspecting the Variance Inflation Factors (VIF) of the predictors and removing those with $VIF > 3$. A complimentary option is to perform model selection on a subset of the data and to assess the predictive ability of the model in another set of the data. I am raising these points because the primary aim of their study is to not only obtain a predictive model but also to obtain accurate estimates of the contributions of the individual predictors, as they reveal important biological insights about mutation rate control. With multicollinearity it is known that predictor estimates can be highly biased. In fact, predictors can be selected into the model without having any significant marginal effect on the response variable. These points should be carefully explored.

Thank you for the helpful comments and recommendations. To more convincingly demonstrate the effectiveness of our model, we have validated it by comparing its predictions with the functional distribution of mutations in multiple new datasets. These include an as of yet unpublished dataset that comprises the largest mutation accumulation experiment in *A. thaliana* carried out to date:

We further show that the predictions from our model extends to somatic mutations in a new analysis of a published dataset composed of deeply sequenced leaf tissue (Wang et al 2019):

Finally, we confirm that the bias we infer is also observed in a natural mutation accumulation experiment, in a recently bottlenecked lineage of *A. thaliana* found in North America (Exposito-Alonso et al 2018):

These new results complement our other analyses inspired by your suggestions. We updated the revised manuscript to reflect the points you have raised. For example, we no longer state that we used AIC to reduce correlated predictor variables. We also explored models where we manually removed predictors with $VIF > 3$; we found that our results were essentially unchanged (few variables changed and genic predicted mutation rates were correlated with the original analysis at $r=0.95$).

4. One of their genetic mutation rate predictors is local cytosine methylation. One mechanism through which this can occur is through spontaneous cytosine deamination. To support this point the authors re-analyzed sequencing data from *ddm1*-derived *A. thaliana* epiRILs, which segregate methylated (wt-derived) and unmethylated (*ddm1*-derived) genomic segments.. They show that transposable elements (TEs) in wt-inherited genomic segments have higher mutation rates than TEs in *ddm1*-derived genomic segments. However, extended Figure 5 seems to suggest that all rates are generally higher in wt-inherited regions, not just C→T mutation rates. In other words, the deamination argument does not seem to be supported. Additionally, the conclusions from their re-analysis conflict the results of Lauss et al. (2018). That latter study also estimated mutation rates in wt versus *ddm1* regions (not just in TEs but anywhere) and found a trend toward increased rates in *ddm1*-derived regions which they attributed to a loss of heterochromatin (and thus an increase in accessible regions).

This discrepancy may be related to one important technical consideration in estimating rates in those epiRILs: The wt segments that segregate in the epiRILs are derived from a different wt plant than the one that was used to generate the *ddm1* mutant. The *ddm1* mutation (in the Col-0 background) has been maintained by repeated backcrossing for many generations. The wt parent that was used to initiate the epiRILs was not immediately related to the *ddm1* parental plant. This means the Col-0 sequence backgrounds of these two parents are very different with respect to the Col-0 reference sequences. Mutations called with respect to the Col-0 reference may thus depend on the on very different laboratory histories of the Col-0 sequence of the wt plant and the *ddm1* parental plant (i.e. the wt parental plant could differ at many more sites from the Col-0 ref sequence, which could explain the increased rate for the wt segments). One way to bypass this is to focus on non-shared mutations, which are those that arose during inbreeding (in fact this is what Lauss et al. had done).

Finally, it is also not clear why the authors test their hypothesis about the effect of their predictor in TE sequences on not genetic sequences (*ddm1*-loss also effects genic sequences). Their model predictors were trained on genes. It is unclear how the authors expect that the relationship between cytosine methylation and mutation rates to also hold for TEs. In fact, patterns of C→T demethylation are much different in genes compared to TEs pointing toward a different predictor effect in these two very different genomic elements.

Thank you for raising these points. Because this result is not directly pertinent to the main conclusions of our paper, as the effects of cytosine methylation on mutation via deamination have been previously described, we have removed this extended data figure in order to make room for the new results we have added.

5. In Fig. 2E, the authors show that the distribution of natural polymorphisms are positively correlated with predicted mutation rates, which is consistent with natural variation in genic regions being

substantially due to mutation bias rather than primarily due to the selection after random mutation. However, later they find that “in genes where mutation rate is predicted to be low, the observed number of polymorphisms tends to be even lower than that predicted by mutation rate alone, which is consistent with the action of purifying selection on natural polymorphisms rather than on de novo mutations. Similarly, coding regions have significantly negative residuals compared to other genic regions, supporting the action of purifying selection after mutation.” This means that – after all – the genic patterns are shaped by a mixture of mutation bias and selection, but it remains unclear how strong the role of selection is. The authors offer no way to quantify the contribution of selection on the polymorphisms patterns. Perhaps this could be achieved through a SFS analysis of different genic sectors. The question here would be if the selection coefficients also vary across the sectors in the manner that is consistent with the polymorphism patterns.

Thank you for this suggestion. We agree that it will be exciting to more fully quantify the proportion of variation that is explained by mutation bias versus that explained by selection after mutation, especially given the covariance we uncovered between these two phenomena.

We have taken your advice and performed multiple SFS analyses. This has revealed striking patterns suggesting that the bias we see in rates of polymorphism is driven largely by mutation bias. We find that polymorphisms upstream of transcription start sites and downstream of transcription termination sites have significantly lower minor allele frequencies and more negative Tajima’s D compared to ones within gene bodies (panels on right). We should expect the effect of selection within gene bodies to reduce allele frequencies on average, suggesting that the variance (lower polymorphism in gene bodies) we have discovered in these regions is indeed predominated by lower mutation probability in gene bodies.

We followed this analysis by measuring Tajima’s D across coding exons to assess if the increased polymorphism could be explained by relaxed selection rather than mutation bias. This analysis revealed that peripheral exons have significantly lower Tajima’s D, which is consistent with the elevated polymorphism in these regions being due to increased mutation rate as predicted, rather than a lack of negative selection after unbiased (random) mutation (panels on the right and next page).

6. Fig. 3g., decreased rates in genes with higher number of introns, seems to partly conflict the prediction that lower rates are detected in evolutionary conserved genes, insofar that evolutionary conserved genes (at least those marked by gbM) are higher in exon content compared to other genes. The authors should explore this/comment on this.

We may not have presented the data in the clearest manner. We believe our results are actually consistent with your interpretation because they confirm that genes with many exons have lower mutation probability. We presented the results in terms of the number of introns, which equals number of exons minus one. Thus, the relationship we discovered between intron number and mutation probability is consistent with genes with many exons having lower mutation probability.

7. The authors' main support for their "adaptive mutation biased" hypothesis is summarized in Fig. 4. There they show some evidence that "Functionally constrained genes have lower predicted mutation rates". They do this by correlating predicted mutation rates in coding regions (CDS) with various intra- and inter-specific measures of genetic variation (PS, PN, DS, DN, neutrality index, delta), and intra-specific variation in gene expression. These correlations are generally low, ranging from weak ($r=0.03$) to moderate ($r=0.26$); thus not lending much support to the "adaptive mutation bias" hypothesis. In addition, the way the data is presented is very misleading, in my view. The authors show plots where the mean of the above intra- and inter-specific measures of genetic variation is plotted for different deciles of predicted CDS mutation rates. These plots give the impression of a strong relationship. A fairer presentation would be to show a scatterplot of the two quantitative variables with the fitted regression line. Another option is to plot the 95% CI around the means (y-axis variable) for each deciles (x-axis variable). Either way will provide a visual presentation that is more consistent with the low/moderate correlation values.

Thank you for the suggestion. We have added 2 * standard error intervals around the means for each decile in every plot. We agree that understanding the effect sizes is an exciting area for future work. Importantly, we would like to emphasize that the significance of our discovery does not derive primarily from the effect size of the observed mutation bias (though some of the new analyses suggest that it might be at least comparable to other non-random evolutionary processes), but rather because it challenges prevailing assumptions about mutations occurring randomly with respect to fitness consequences. We agree that future research is warranted to more fully understand the limits and magnitude of the reported mutation bias.

We have also added a number of new analyses that support the finding that mutation probabilities are lower in constrained genes, measured by empirically determined phenotypic effects. Specifically, in the revised manuscript we now compare genes with experimentally validated lethal vs. non-lethal effects of knockout mutations, as well as being annotated with essential, morphological, cellular/biochemical, and environmentally conditional effects. These analyses confirm our results regarding the relationship between mutation probability and evolutionarily informed estimates of functional constraint - essential genes are enriched for epigenomic features associated with low mutation probability.

8. In that same figure, the relationship between predicted CDS rate and the coefficients of variation in gene expression indicates that predicted hypermutable CDS (thus, more polymorphic CDS in natural populations) have a higher expression range. This relationship could simply be the consequences that these polymorphisms lead to expression variation via cis-acting eQTL, rather than hypermutability being the result of gene expression states. The authors should comment on this.

Thank you, we agree that this is a reasonable and interesting hypothesis, reflecting the circularity that emerges from covariance between selective constraint and mutation probability. We hope the revised manuscript more clearly articulates our assertion that an increase in gene expression variation reflects expression variation in such genes being often tolerated by evolution.

9. In that same figure, correlations are very similar between CDS and PS and PN (or DS and DN) suggesting that mutations of functionally neutral and putatively non-neutral mutations are nearly equally effected. This is consistent with previous observation that constraint loci tend to evolved more slowly in general. In fact the DN/DS ratio correlation is much lower suggesting again that correlation with measures of constrained evolution are weak.

Thank you for raising this important point. As you point out, the effect of the mutation probabilities we have found is confirmed in the relationship with Pn, Ps, Dn, and Ds when comparing predicted mutation probability and patterns of natural variation in genes. However, as you also thoughtfully point out, the signal is stronger with nonsynonymous variants, which cannot be easily explained by mutation bias preferentially reducing nonsynonymous changes in genes under purifying selection (there is no plausible mechanism for this). Rather, the most likely explanation is differential levels of selection among genes in natural populations, with genes having the lowest mutation probabilities being subject to stronger negative selection. Given the significance of this finding as a major deviation from the traditional assumptions that mutation probabilities are independent of fitness consequences, we are hesitant to include in the scope of this particular paper a deep investigation into the strength of this effect (i.e., strong

vs. weak correlations). We anticipate considerable work to fully quantify this in *Arabidopsis* and other species into the future.

Referee #3 (Remarks to the Author):

In this provocative manuscript, Monroe et al. argue that in *Arabidopsis thaliana*, local mutation rates have been selectively optimized such that they are lower in genes where mutations are more likely to be deleterious. If true, this finding would have very broad implications for evolutionary biology. Their observations are intriguing, but the analyses are somewhat convoluted, making it difficult to assess the strength of their conclusion. Their proposed mechanism of selection through genome-wide epigenetic marks is tantalizing but has some problems. It is difficult to prove that their observation results from selective optimization of mutation rates. My detailed comments follow.

Major comments

1. My main concern of the data analysis part is the use of predicted instead of actually observed mutation rates. They provide no evidence how good the prediction is. In fact, all of their analysis can use actual mutation rates, and it is unclear to me why they did not do that. For instance, a supergene analysis should be able to generate Fig. 2d and Fig. 3. Fig. 4 can be generated by considering 10 supergenes for the 10 deciles. I understand the purpose of showing the correlation between epigenetic marks and mutation rates, but this is about the potential selection mechanism, which is not needed for comparing trends of mutations and trends of polymorphisms/evolution. Their comparison between mutation and polymorphism/evolution patterns would be much more straightforward and convincing if they had used the actual mutations.

Thank you for an excellent suggestion! We have added several analyses of actual observed mutation data to the revised manuscript. For example, simulations reveal a 70% reduction in the expected number of synonymous mutations in coding regions in our actual data, which is consistent with reduced mutation probability in gene bodies. We also show that genes in which mutations are predicted to have deleterious effects (high lethality index predicted from phenotypes in knockout lines) are significantly depleted in the set of observed *de novo* mutations (first panel on the right). That there is no difference in nonsynonymous/synonymous ratios (second panel on the right) confirms that this difference is unlikely to be due to selection having affected our *de novo* mutation dataset.

For the majority of results, we used predicted values, what we call mutation probabilities, which can be thought of as an integrated index of epigenetic features associated with mutations in our empirical data set. We did this for several reasons: 1) We could experiment with removing coding regions from the training set when building the predictive model. This allowed us to show that the results cannot be readily explained by negative selection on coding sequences in our *de novo* mutation data set. 2) Relatively few mutations can be directly observed in gene bodies ; after all, there is a ~70% reduction in synonymous variants compared to the genome-wide background. There is thus relatively limited power to study differences between classes genes. 3) Most importantly, by using a predictive model, we are able to test

the power of that model on new datasets: We have added three new datasets to the revised manuscript to demonstrate this. First, we have added unpublished results from the largest mutation accumulation experiment performed to date in *A. thaliana*, which involved 400 lines from 8 founder genotypes propagated by single seed descent for 8-10 generations. By sequencing these lines we could describe mutational patterns across the genome in a similar fashion to our original mutation accumulation experiment. We show that our predictive model has remarkable power to explain the functional distribution of these new mutations, a result which is now prominent in the revised manuscript. The figure below shows the distribution of predicted mutation probabilities and observed *de novo* mutations around transcription start sites and transcription termination sites:

In addition, we called *de novo* somatic mutations in a previous data set from 64 individual leaves from two *A. thaliana* plants that had been deeply sequenced (Wang et al 2019). These data confirm again that our predictive model has outstanding power to predict the distribution of *de novo* variants:

Finally, we validated the predictive power of our model with an analysis of mutations in an isogenic North American *A. thaliana* lineage that experienced a recent severe genetic bottleneck. Again we see that the distribution of new mutations is remarkably similar to our predictive model trained on the original empirical set of *de novo* mutations:

We found additional evidence that the observed distribution of new mutations is driven to a large extent by mutation bias, rather than by selection alone, by quantifying the average minor allele frequency around transcription start and termination sites in 1,135 *A. thaliana* accessions. These data show that minor allele

frequencies are higher in gene bodies (which is not expected if the observed reduction in the number of polymorphisms in gene bodies is driven entirely by strong negative selection). We also calculated Tajima's D values in these same windows, and they further support the interpretation that the pattern observed in natural populations is driven to a large extent by mutation bias:

We then ran a series of simulations to test whether these results could be plausibly explained by extremely strong purifying selection. We were unable to find support for this hypothesis. Even if we assume that 30% of coding sequence variants are dominant lethal (which is unrealistically high), we find the distribution of variants only weakly related to the patterns we observe independently in our datasets. These results can be found in the extended data, along with previous results showing that the distribution of mutations we observe cannot be explained by mapping depth, mappability, or the distribution of false positive variant calls:

2. They interpreted their observation of correlations between predicted mutation rates and several gene expression or evolutionary constraint measures by selective optimization of mutation rates. I am not fully convinced by this interpretation and believe there are other possibilities.

(1) The formula in line 223 works only in the absence of recombination (e.g., see Kimura 1967 Genet. Res. or Lynch 2011 Genome Biol. Evol.). Because *A. thaliana* is selfing, it effectively has no recombination. So, one could argue that the formula works here. But this unusual reproductive mode evolved within the last million years in *A. thaliana*. For the above argument to work, one must assume that all the optimizations of mutation rates occurred within the last million years in *A. thaliana*. If so, how do we explain the *Populus* results, as *Populus* is outcrossing and should not have the type of selective optimization of mutation rates?

(2) Another point about the formula in line 223 is that, under the assumption that there is a global mutation rate modifier, the drift barrier is determined by this formula when L_{segment} is the length of the whole genome (or functional genome) (Lynch 2011 Genome Biol. Res.). In other words, ΔU and U are tiny at the drift barrier. So, when L_{segment} is say 1% of the functional genome for a particular epigenetic

feature, ΔU must be much bigger for selection to work. Given that U is already very low, it is impossible for ΔU to be large (you cannot reduce mutation rate to negative values). Thus, the authors' model is actually incompatible with the drift barrier hypothesis. I am not saying that their model is wrong because it is incompatible with the drift barrier hypothesis, as the drift barrier hypothesis could be wrong. But my reading is that the authors seem to be saying that their model is compatible with the drift barrier hypothesis. This needs to be clarified.

Thank you for the thoughtful comments. We agree that this section previously did not clearly articulate our interpretation of the results and we changed this in the revised manuscript. We have removed the statement that our data suggest a high ΔU , which, as you accurately point out, is confusing, as the existing mutation rate sets a clear limit to how much this can change. We have therefore simplified our discussion around these equations, focusing on how the mechanistic model presented indicates that a large L_{segment} is being affected, due to the genome-wide distribution of the features that are plausibly linked to mutation probability through DNA protection/repair mechanisms. We hope that this more concise explanation of the relationship between our results and previous theory (i.e., drift-barrier hypothesis) is now clearer.

(3) If the observations in Fig. 4 are genuine, I wonder if they could be produced by non-adaptive processes. One possibility is that the correlations are intrinsic. For instance, methylated Cs have elevated mutation rates for chemical reasons. If somehow there is a correlation between the abundance of methylated Cs and gene expression purely for mechanistic reasons of gene expression, one may see a correlation between gene expression level and mutation rate that is not a result of selective optimization. Because dn/ds is strongly correlated with gene expression level in a wide variety of species including *A. thaliana* (Zhang and Yang 2015 Nat. Rev. Genet.), one would also see a correlation between dn/ds and mutation rate.

(4) Related to the above point, for the authors' model (i.e., selection acting on epigenetic marks) to work, there must be a significant correlation between epigenetic marks and gene expression/selective constraint, which the authors have not shown. If they can confirm this correlation, the non-adaptive process outlined above would easily work because all one need is an intrinsic relationship between epigenetic marks and mutation rate, which has been reported and cited in the manuscript. Thus, it seems that the condition necessary for the selective optimization hypothesis to work also makes the intrinsic property hypothesis work. In other words, the conclusion of adaptive mutation rate optimization cannot be proven.

These points are spot on. The revised manuscript confirms the difference in epigenomic states between essential and non-essential genes (panels on the right). These differences include significant differences in histone modifications, which are associated with lower mutation probability. Importantly, we also show that lethal-effect genes have significantly higher cytosine methylation, which is inconsistent with mutation bias being a coincidence due to intrinsic

properties of the epigenetic machinery. Although methylated cytosines are intrinsically more mutation prone because of spontaneous deamination frequently leading to C>T substitutions, this effect is apparently overcome by the mechanisms underlying the mutation bias we have discovered. Similarly, we find no difference in mean expression between lethal-effect and non-lethal-effect genes (t.test, $p=0.57$; result, but not a separate figure, included in revised manuscript because of space constraints), indicating that intrinsic effects of transcription-coupled repair cannot explain our results.

Because the observed patterns look so much like the outcome of natural selection, we are somewhat hesitant to discuss non-evolutionary forces (while hypothetically plausible) without some clear supporting evidence to the relationship we observe between mutation probability and adaptive constraint on genes and functionally distinct regions. Specifically, we are concerned that an interpretation of apparently adaptive bias in mutation - specifically reducing deleterious mutations - as an intrinsic property of the genome would be received as even more controversial than the interpretation that we put forth (and even be potentially abused by non-scientists with ulterior motives to undermine evolutionary biology generally). Thus, while we expect these results still likely to be viewed as provocative, we believe our interpretation of the results in a manner that is intentionally consistent with the action of evolutionary processes rather than a purely emergent phenomenon is more conservative and parsimonious in light of standard Darwinian theory.

(5) A related comment is on the interpretation of the negative correlation between mutation rate and UTR/intron length. Again, this negative correlation could be intrinsic instead of a result of selection.

This is an interesting idea as well. We agree, as with mutation bias in general, the relationship could potentially be explained as an emergent phenomenon. In either case, the pattern does seem to provide an adaptive benefit that mirrors selection: greater intron number in *A. thaliana* facilitates lower mutation probability in coding sequences, which are more susceptible to deleterious mutation, by distancing these from elevated mutation probability at the periphery of transcribed regions.

3. Fig. 4, I did not find the GO analysis informative. Which function can be said to be unimportant? All functions are important, and designation of important/unimportant function is quite arbitrary. One less arbitrary index that the authors are encouraged to use is gene essentiality based on knock-down/out phenotypes, and there is such data for *A. thaliana* at the genomic scale. A key advantage of using gene essentiality is that it is an objective measure of functional importance yet it is not likely to be strongly correlated with gene expression level. This potentially allows one to distinguish the intrinsic property hypothesis from the adaptive optimization hypothesis.

Thank you for this excellent suggestion. We are very excited by the results that have emerged from performing these recommended analyses. First, we analyzed genes classified as lethal-effect and non-lethal-effect based on phenotypes observed in knockout lines. We discovered that lethal-effect genes are enriched for features predicted to reduce mutation probability such as H3K4me1. This is important, because it provides a plausible mechanistic explanation for selection to act on interactions between DNA repair/protection processes to become preferentially targeted to specific epigenomic features. We then show that this difference in epigenomic states is manifest as differences in mutation probability between lethal-effect and non-lethal-effect genes.

Second we analyzed genes classified as essential or having morphological defects, cellular/biochemical defects, or defects in specific environmental conditions when knocked out. This revealed similar results, demonstrating that essential genes are enriched for features associated with low mutation probability. In contrast, genes with other phenotypic consequences when mutated (especially ones whose effects are environmentally conditional) have lower levels of these features and thus higher mutation probability. We finally added another analysis showing that genes that are consistently expressed in all tissues are enriched for features associated with low mutation probability.

Combined with results from the GO analysis (which we agree do not provide a complete picture on their own), these new findings yield compelling evidence for a simple model for the evolution of beneficial mutation bias: constitutively expressed essential genes are enriched for epigenomic features that have become associated with a reduction of mutation probability in these genes, resulting in reduced deleterious mutation load. One can think of two alternative scenarios: (1) These genes happened to be enriched for specific epigenomic features and the DNA repair machinery subsequently evolved to be preferentially targeted to these features. (2) The DNA repair machinery was more likely to be targeted to specific epigenomic features to begin with, and these genes subsequently evolved to be decorated with such features. While this will be ultimately difficult to prove conclusively, the first scenario appears to be the much more likely scenario:

Reduced mutation rates in gene bodies

Reduced mutation rates in essential genes

Greater fitness via reduced mutation load

Minor comments

1. Fig. 1b is large but contains little discernable information, so should be removed.

Thank you for the suggestion. We agree - and to make room for the results we have added to the revision, we have moved this to the extended data.

2. Fig. 1c. If my reading is correct, it shows that gene-poor regions (white parts in chromosomes) have lower mutation rates than gene-rich regions, directly contrasting the main conclusion of the paper. Please clarify.

This figure shows the mutations used to inform our predictive model, which is focused on genic regions. We have clarified this in the figure legend.

3. Fig. 1d, it is difficult to sense the ratio after the log-transformation. While it is OK to use the log-scale, I suggest that they present untransformed values on the Y-axis for easy reading. My back-of-the-envelope calculation based on the numbers provided in the manuscript shows that the difference between “de novo” and “null” bars of non-syn/syn is significant instead of NS. Please check the statistics or provide the raw numbers used in the test.

Thank you for the suggestion. We have added untransformed values to the y axis for clarity.

4. Fig. 4a, how different is the mutation rate (in absolute terms) among the 10 deciles? Please provide the information in the figure. Because the theoretical model involves ΔU , it is critical to know how large the ΔU is.

In light of the previous comments, we have removed our discussion of ΔU .

5. Extended Data Fig. 6b. Negative values are interpreted as purifying selection. How about positive values? Do they mean balancing selection? How likely is it that introns are subject to strong balancing selection?

We are hesitant to interpret the positive residuals as evidence of balancing selection in introns. We speculate that this is simply a function of introns being the gene features with the greatest disconnect between strength of selection and mutation rate: introns are predicted to experience fewer mutations but weak selection. In other words, introns enjoy a low mutation probability due to their position within gene bodies, but mutations are also more often maintained via drift, so the number of observed polymorphisms is higher than predicted from mutations alone (when considered in relation to UTRs which may also have lower constraint but a correspondingly high mutation probability, thus being closer to the fitted model).

Referee #4 (Remarks to the Author):

In their manuscript entitled "Adaptive mutation bias in *Arabidopsis thaliana*" by Grey Monroe et al have studied the determinant of mutation rate in a mutation accumulation experiment from *Arabidopsis thaliana*. They found some fluctuations of mutation rate along the genome and suggest that genes with higher level of constraints have lower mutation rate. This proposed reduced selection for mutation rate is supposed to be driven by genome wide factors that may generate a large enough benefit to select for a lower mutation rate.

My major concern with the present data is the number of cellular generation involved in the experimental design. One complexity of mutation accumulation is to ensure that no selection is occurring in the process. In bacterial experimental evolution, one simple tool is K_a/K_s ratio that can be used to investigate the lack of selection. We can push things even further with looking at predicted mutation

effects using Direct Coupling Analysis as we did in Couce et al PNAS 2017 (<https://doi.org/10.1073/pnas.1705887114>). In these experiments a Ka/Ks equal to one is a simple proof that there is no selection detectable at the genome scale. Here the data are going in the good direction with a reduced selection, but without any more details on the null model used and the formula used to compute these ratios it is hard to be sure.

Indeed, a large fraction of the genome may show no Ka/Ks deviation as it is unconstrained, but the rest of the genome may leading to the absence of signal at that level despite a true selection that may later on affect the detection. In that respect the different formulation of the null model taking into account mutation biases has to be explicitly given as it may have a strong impact.

In that respect it would be worth doing simulations to see how much selection in selectively important genes in the MA process would be needed to lead to A) lead to a significant deviation in the overall KA/Ks, B) to generate the signature of selection detected.

An alternative could also to take the genes ranked by their mutation rate as done in the last figure and to plot this time their Ka/Ks ratio for each of the deciles to see if there is a constant association between low mutation rate and a signature of selection in the MA.

Thank you for the excellent suggestions. Given the nature of our findings that mutations occur less frequently in constrained loci, the possibility that this was due to unexpected selection in the experiment has been a constant concern. With the experimental design (single seed descent, where a single seedling is arbitrarily chosen for propagation, regardless of its phenotype), we expect selection to be at an absolute minimum in the germline, as long as the mutations do not kill the seedling or extraordinarily delay its germination. In the soma, where mutations remain heterozygous, selection has anyway been shown to be minimal to the point of being undetectable (Wang et al 2019). Nevertheless we have sought to ensure that our results cannot be explained by selection on the *de novo* mutations used in our training data. For example, by building a predictive model of mutation probability rather than relying on raw mutations for downstream analysis, we were able to construct a model in which coding regions are excluded from the training set. The predictions for coding regions were nevertheless very similar to the full model that used raw mutations from inside and outside coding regions. We also found that low mutation probability genes have lower than expected (fitted values) levels of natural polymorphism, which suggests that they are subject to purifying selection in the wild rather than in our data. As you point out, we also observed that our set of *de novo* mutations was statistically similar to a null expectation in terms of rates of nonsynonymous and synonymous variants, which we have explained in further detail in the manuscript. Specifically, we clarify that the null expectation was built from the known codon use of *A. thaliana* and the molecular spectrum of coding sequence mutations.

We have added an analysis comparing nonsynonymous to synonymous variants in genes distinguished by their estimated degree of lethal effect determined from empirical measurements of knockout lines (panel on the right). We found that the rate of non-synonymous mutation relative to synonymous mutation is the same for genes that are highly mutation sensitive (“lethal effect”) vs. less sensitive (“non-lethal effect”), despite these groups of genes showing significantly different rates of *de novo* mutation in our data. This result supports the

interpretation that lower mutation rates observed in our data are indeed the result of mutation bias rather than strong purifying selection removing functionally impactful mutations from coding regions of essential genes in the set of *de novo* mutations.

We also calculated nonsynonymous and synonymous ratios in genes grouped by their mutation probabilities, as you suggested. This confirmed that genes with low vs high mutation rates do not vary in terms of nonsynonymous and synonymous ratio in a predictable manner:

We have also added several simulations demonstrating that selection cannot reasonably explain our data. First, we simulated random mutations across coding regions of the genome (1,000 iterations) and calculated the expected nonsynonymous/synonymous ratios. The distribution is shown below with our data marked with the vertical red line in the panel on the right. This result confirmed that our data are consistent with neutral rates of variation.

We also simulated selection in coding regions across the genome at varying strengths of purifying selection. In these simulations we created 10,000 random mutations across the genome and selected against coding sequence mutations at $s = 0.01$ (1% of coding region mutations dominant lethal), $s = 0.1$ (10% of coding region mutations dominant lethal), and $s = 0.3$ (30% of coding region mutations dominant lethal). Even at this extremely high degree of simulated purifying selection (0.3) we did not observe the amount of bias that we find in our *de novo* mutation calls dataset (although we do see a slight reduction of variation in gene bodies).

In summary, with these new results generated based on your suggestions along with our previous results, we were unable to detect evidence that unexpected selection on our *de novo* mutation dataset is a likely explanation for our results.

I am not sure about the meaning of the scaled values presented in figure 2. Indeed, throughout the text, the amplitude of the signals should be more precisely delineated. P values are not a proof of large effect. Are we talking about 10 fold differences? In Couce et al we use log-odd ratio for discrete traits and interquartile odds ratios for continuous ones. This is interesting but giving more explicitly the range of the observed signal as in figure 3 would be nice.

Apologies for the confusion - we have updated the figures to more precisely denote the relevant units. Specifically, we have defined a simple Mutation Probability Score, which is the indel + SNP mutation probability predicted by the model built from our set of observed *de novo* mutations.

If there are enough synonymous mutations, analysis could be done using just these, or adding synonymous state to the model.

Thanks for the suggestion. We have added an additional simulation of the expected number of synonymous mutations we would observe if mutations occurred randomly across the genome. From this simulation we found that there is a 70% reduction in synonymous variants among the observed set of *de novo* mutations relative to expectations under random mutation, and have added this result to the paper. This result provides additional support that the pattern we see is not due to selection but rather to mutation bias.

Experiments with bacteria revealed that a major driver of mutation rate is the bases before and after the focal base as well as the specificity of the base (<https://doi.org/10.1093/molbev/msv055>,<https://doi.org/10.1073/pnas.1705887114>). Here it seems that all these effects have been ignored. But how much of the signal could be due to these local sequence composition effects. As sequence specificities may varie from coding to non coding this can also explain some of the variability observed.

Thank you, this is also a very interesting point. Local sequence context might well play a role, and it might indeed contribute to differences in epigenetic marks and DNA repair. Such a scenario would, however, not invalidate any of our assertions about mutation bias along genic sequences.

The p values between the changes in mutation and the mean per decile are meaningless. P values of any level can be obtained when performed on mean binned data. The rough p values and correlation should be given, even if the plot is made with the deciles for clarity and communication purposes. Again, how much siganture of selection in the MA would be needed to explain some of these signals.

Thank you for raising this. We have clarified in the revised manuscript that these p-values do not reflect the binned values but rather rough correlations and p-values. The binned data are used only for visualization purposes to capture non-linearities and deviations from the relationship. We have also added error bars.

Besides these technical comments, the analysis performed are really interesting.

Is there any hypothesis on why mutation rate may be larger at TSS and TTS sites precisely? Secondary structure have been shown to affect mutation rate (Barbara Wright work (<https://doi.org/10.1099/mic.0.2007/005470-0>) on bacteria that we followed in Hoede et al Plos Genetics 2006). Could that be involved at these sites?

Thank you! Our model suggests that mutation probabilities are higher upstream of TSS and downstream of TTS because the chromatin is more accessible, to provide access for regulatory transcription factors and the transcription machinery. In addition, recombination, which is known to be mutagenic (Halldorsson et al 2019), might contribute to these patterns, because they are similar to patterns of SPO11-generated double strand breaks (DSBs) (Choi et al 2018). Also, these regions are depleted for features that mark internal gene bodies, and thus may simply lack the signals that recruit DNA repair.

For all the histone methylation marks, is there a synthetic way to present their various contribution: the ones linked to expression, the ones linked to deletion to accessibility... Apart from one or two mentioned in the text, the others are just left to specialists for interpretation. So a synthetic figure giving some insights would be great.

If the reviewer feels strongly about this, we could add a table regarding the effects of the different epigenomic features. We have shied away from doing this, because they are not always unambiguous. For example, cytosine methylation in the context of so-called gene body methylation is associated with higher levels of gene expression, but in the context of so-called TE-type methylation, it is associated with reduced levels of gene expression.

We have nevertheless added a summary figure to the manuscript that partially goes in this direction. While perhaps different than what was suggested, this figure provides a conceptualization of the general model of mutation bias that we observe - mutation bias is linked to epigenomic features which themselves are linked to differences in functional constraint across the genome. Though we do detect strong associations between specific genomic and epigenomic features and mutation probability, we were hesitant to include individual associations in our conceptual model as future functional work will likely be needed to fully dissect the causal relationships with specific features, potential interactions, and more precise molecular mechanisms underlying the associations that we have uncovered.

Olivier Tenailon (I always sign my reviews)

Reviewer Reports on the First Revision:

Referee #1 (Remarks to the Author):

The authors have dealt with the one general issue I raised in the first review. I remain enthusiastic about this study given its possible impact on evolutionary thinking.

Referee #2 (Remarks to the Author):

Grey et al present a revised version of their now entitled paper "Mutation bias reflects natural selection in *Arabidopsis thaliana*". The authors made substantial efforts trying to address the many technical and conceptual concerns raised by the four reviewers. I continue to think that it is an interesting paper, but I just don't think it is quite enough to be published in Nature. Building on my previous comments, my major concern is the lack of sufficient quantification of their predictive model. In my view this is the heart of the paper, and drives all downstream conclusions/interpretations. Most of the results/conclusions currently rely too much on visual comparisons between predicted and observed patterns, rather than on quantitative measures such as "variance explained by the model". Such model performance statistics should be provided. Without them, it remains impossible for me to judge the relevance/impact of their paper. Here are my major concerns. These lean on some of my previous points, but also include additional questions in light of the revised version, the comments of the other reviewers, and me having had a bit more time to think about this.

1.

The section "Predicting mutation probability across the genome" introduces their predictive model. The variance in observed mutation frequency by their final selected model is nowhere reported. Only the final predictor set along with the associated t-values of the coefficients are shown in Fig. 1. The t values are relatively large and should be highly significant. But as with anything in genomics, the input sample sizes are also large (all the genes), which renders standard errors of the coefficients small. A more meaningful quantity to report is something like the R² value, which tells us how much of the variance in the mutation frequency patterns is explained by the model.

Also, standard statistical output such as values of the coefficients, t values and associated degrees of freedom, a plot of the distribution of the response variable(s) and perhaps residual diagnostic would be good to see (e.g. residuals versus fits plots). Again, it is still not clear to me how they set up their "generalized linear model". Do they assume their response variable is normally distributed? Or drawn from another distribution. Making model choices solely based on visual inspection of the predicted mutation probably along genes is not sufficient for me.

In one of their replies, the authors state "Importantly, we would like to emphasize that the significance of our discovery does not derive primarily from the effect size of the observed mutation bias (though some of the new analyses suggest that it might be at least comparable to other non-random evolutionary processes), but rather because it challenges prevailing assumptions about mutations occurring randomly with respect to fitness consequences. We agree that future research is warranted to more fully understand the limits and magnitude of the reported mutation bias". I strongly disagree with this perspective. The authors have all the results/model fits, etc. at hand. Why not just show the results to allow for a more fair assessment of their finding. Are we talking about predicted mutation biases that are relatively trivial relative to other evolutionary processes shaping natural variation, or are we looking at major effects that would fundamentally require us to reevaluate previous conclusions about genomic patterns of selection. To me, this difference determines if this ms is published in a high impact journal like Nature, or somewhere else.

2.

Related to this, all the plots in Fig. 2 showing the observed and predicted mutation frequency are represented in terms of mean values (calculated from all the genes), but show no information about the variance (i.e. variation around the mean pattern). A similar concern lingers for me in Fig. 3. My previous comments and those of another reviewer were critical of the decile plots shown in that figure. These plots convey a strong relationship between predicted mutation probability and various measures of evolutionary constraint. But looking at the correlation values (r) given in that figures, the relationship is actually very weak. In fact, the R^2 (correlation coefficient squared) would even be smaller. I had asked for an alternative representation that plots quantitative values of predicted mutation probability versus constraint measures. This would be a fairer representation and more consistent with the low correlation values. As an alternative I mentioned plotting 95% CI around the decile means. The authors did the latter. I now realized that the large sample sizes per decile would still lead to relatively small CI. So, after all, I think a quantitative bivariate plot with a regression fit is the fairest way to present the data (bivariate densities could be indicated via color code).

3.

It also occurred to me that the final selected predictor set for SNV and INDELS is very different, which raises the question if this relates to different biology and/or the fact that INDELS tend to occur in different parts of genes than SNVs so that they relate to different epigenomic features. I don't see any plots to show the observed mutation frequency distribution contrasting SNVs and INDELS. The authors also don't comment on this.

4.

The authors now included new PC analysis of the epigenetic marks they obtained from the PCSD. It remains unclear why they do this and not just plot the distribution individual marks along the gene (see Fig. 2A), or the chromatin states given in the PDSC database. If the authors think the PC dimensions are important, why don't they use the PC loadings as predictors in the model. The fact that various epigenetic modifications load onto the PC means that they are correlated. This brings me back to the variance inflation factors. The authors have performed VIF analysis (by the way I would not mention this in the main text as it is trivial and not a main result) and found little evidence for this. Why not show the VIFs for each predictor in a SI table so we can see more clearly.

5.

The authors bypassed my concern about their epiRIL data analysis by stating that " Because this result is not directly pertinent to the main conclusions of our paper, as the effects of cytosine methylation on mutation via deamination have been previously described, we have removed this extended data figure in order to make room for the new results we have added". Since the main conclusion of their revised paper is the same as their original submission, I do not see how the epiRIL analysis was pertinent before but is no longer. Space limitation in SI should not be an issue. The authors don't comment at all on whether I was wrong. I am happy to be proved wrong, but felt that the removal of their analysis bypasses potential issues in their original interpretation.

6.

Finally, I appreciate that the authors clarified the differences between "adaptive Mutation Bias" and "Adaptive Mutation bias". In their formulation, the patterns of reduced mutation probabilities in constraint genes increases fitness by avoiding deleterious mutations. In this sense they are "adaptive". Because this mutation bias increases fitness, this bias itself must be an evolving trait (and the authors also mention this in their discussion). There were a few comments from the reviewers how this bias could evolve. I acknowledge this this a deep and difficult question. But, the possibility that it can evolve (or that it is postulated to evolve) would require that this bias is variable in natural populations, as surely the predictor set the authors used will show variation among accessions (e.g. mCG is highly variable among natural accessions). The authors have now included additional MA lines that were established in different genetic background (i.e. derived from different founder accessions). This new data provides an opportunity to examine the range of

natural variation in mutation bias. In fact, going one step further: this new data would also allow the authors to assess a causal link to their predictor set by profiling the founder accessions for at least some of their strongest predictors. Such an analysis would provide some empirical hints as to how this bias does/can evolve, and would thus mitigate any potential philosophic debates and controversies that arises from their conclusions.

Referee #3 (Remarks to the Author):

I thank the authors for addressing my previous comments and for including many new analyses in this version. The manuscript has been strengthened through the revision. However, key issues remain unresolved.

Major comments:

1. Related to previous major comment 1.

(i) I asked the authors to use raw mutation numbers instead of predicted mutation probabilities to study mutation rates in different genes or genomic regions and to compare mutation with polymorphism. They used raw mutations in only one plot (lethal vs. non-lethal genes; Extended data Fig. 2c). But in all other plots, they still used predicted mutational probabilities. It appears that there are sufficient mutations in the data to allow using raw mutations. For instance, in 10 bins in Fig. 3, each bin would have hundreds of mutations. In Fig. 2b, for a 100-bp window, there should be >200 mutations. They now acquired a new mutation accumulation dataset (Fig. 2cd) that is larger than the primary dataset used (the exact mutation number in the new dataset is not provided in the manuscript). They could combine the two datasets to get even more mutations.

(ii) I complained that they did not tell us how well the model fit the data, and they still do not provide such statistics. It is interesting to note from the raw mutation numbers in Fig. 1d that SNV rate is not lower in coding regions than non-coding regions. Even for indels, the mutation rate is only slightly lower in coding regions than non-coding regions (Fig. 1d). This pattern, based on raw mutation numbers, gives a very different picture than the one based on predicted mutation probabilities (Fig. 2b). That is why it is important to use raw mutation numbers.

(iii) Regarding the model, the Methods section states that "The response variable was the untransformed (to avoid risk of increased false positives caused by transformation) detected mutation rate across every genic feature (upstream, UTR, coding, intron, downstream)." Does this mean that they used different models for different genic features (upstream, UTR, coding, etc)? What is the rationale of such modeling? Do they assume that the same epigenetic marker has different effects in different genic features and why? What result will they get if they use the same model for all genic features?

2. Related to previous major comment 2.

(i) Regarding the interpretation of their observation, authors state that their adaptive model is compatible with population genetic theory if L_{segment} is large, but do not provide concrete evidence that L_{segment} is indeed larger than that required by the theory. I suggest that they calculate, based on mutation rate and other parameters, the theoretically minimal L_{segment} for the adaptive hypothesis to work, and compare it with the true L_{segment} . Although the true value of L_{segment} is probably unknown, I believe one can estimate its order of magnitude.

(ii) Authors seem to agree that they cannot reject the intrinsic property hypothesis but is reluctant to discuss it because they felt that it would be "even more controversial" than the adaptive hypothesis and may be misused by non-scientists. In my view, drawing conclusions without considering/discussing alternative explanations is more dangerous.

3. This is a new major comment arising from their new analyses.

I am very confused by their new results on Tajima's D and minor allele frequencies. As far as I know, a change in minor allele frequencies does not indicate a change in mutation rate but a change in selection. So, the finding that genic regions have higher minor allele frequencies than non-genic regions does not tell us anything about mutation rate difference between these regions, but suggests that coding regions are subject to weaker purifying selection than non-coding regions, which is unexpected and deserves an explanation. Is this result consistent with previous population genomic analysis of the species? Is the same result obtained when raw mutations are used? Similarly, a less negative Tajima's D of genic regions than non-genic regions suggests a weaker purifying selection acting on genic than non-genic regions.

4. This is a new major comment arising from their newly presented results.

Because there is no variation in SNV mutation rate among genic features (Fig. 1d), are all the mutation rate differences in Figs. 2, 3, and 4 actually due entirely to indel mutations? If so, this should definitely be stressed because indel and point mutations are likely caused by different molecular mechanisms. Do polymorphisms in Figs. 2, 3, and 4 include both SNPs and indels? Also, at the end of page 3 and beginning of page 4, authors discussed that their results from modelling are consistent with previous findings. Are they consistent with previous findings on both SNVs and indels? If I am not mistaken, previous findings are almost exclusively on SNVs. Please clarify.

Minor comments

1. Previous minor comment 2. The pattern is still opposite to what was claimed in the manuscript. It shows that gene-poor regions (white parts in chromosomes) have lower mutation rates than gene-rich regions (black parts in chromosomes).

2. Previous minor comment 4. Authors did not answer how different the 10 deciles in Fig. 3 are in mutation rate (in absolute terms).

3. New minor comment. "We found a 70% reduction in synonymous variants in the raw de novo mutation dataset relative to random expectations". This sentence on page 4 appears contradictory to Fig. 1d, which shows virtually equal SNV mutation rates across genic features. Is the lowered synonymous rate compensated by an elevated nonsynonymous rate?

Referee #4 (Remarks to the Author):

In this revised version of the manuscript, the authors have made a lot of efforts to answer my comments and I greatly appreciated the effort and the positive tone of the answer.

I was nevertheless not fully satisfied with one part of the extra data presented in the extended material: the simulations made to look at the effect of deleterious mutations. The model was too simple and assume very large effect filtered at once. Because strong bottlenecks are imposed, what matters here is more the potential within host selection from gamete to gamete.

Let us assume that mutations within a given gene occur at rate μ and have a deleterious effect at the heterozygous state such that they hamper the reproduction rate of the affected cells through the development process by $(1-s)$. If there are N cellular generations within a plant generation, we have $\mu + \mu(1-s) + \mu(1-s)^2 + \dots + \mu(1-s)^{(N-1)}$ mutated cells because some cells have mutated at the last division and have not been counter selected, some have mutated at the generation before and have a $1-s$ probability of being a gamete, some two generations before and therefore have been counter selected for two cycles $(1-s)^2$ and so on. This gives $\mu(1-(1-s)^N)/s$ and if we compare to the number of neutral mutations we would expect μN ($s=0$) the counter selection of mutations occurring within the gene that have effect s is $(1-(1-s)^N)/(Ns)$.

If we assume 30 cellular generations between plant generations and a 1% fitness effect we expect a 15% underrepresentation, if the cost is of 2% it is a 25% reduction.

I have no clear idea of the values of N and s , but I think it is quite important to substantiate the modelling part in that respect. The argument that a 30% loss of mutation is unrealistic, is not that much unrealistic if we take into account the cumulative counter selection over the gamete to

gametes development that can turn a realistic small effect into a quite large counter selection. The outcome of this process could appear through the synonymous, non synonymous pattern, but may leave a marginal signal in term of significance, that may also be due to partially wrong mutational biases and other noise factors.

Interestingly if N is quite large and some s are important, this cellular process will also lead to a decreased mutation rate in important genes from generation to generation, and may also have some evolutionary implications that are worth mentioning.

Olivier Tenaillon

Author Rebuttals to First Revision:

Summary of revisions

As requested, the majority of results are now presented more quantitatively and in terms of mutation calls rather than probabilities derived from our linear model based on epigenomic features. We show that, as predicted, empirically observed *de novo* mutation rates in the absence of selection are lower in gene bodies (58% reduction), even lower in essential genes (a further 37% reduction), and genes subject to stronger purifying selection. We have, of course, also quantified how well the original model could predict some of the patterns discovered. For example, we asked, “how much of the pattern of mutation rate bias around gene bodies is explained by the epigenome-predicted mutation probabilities?”. We found that our model explains over 90% of the variance. We also show that modeling biased mutation rates can explain over 90% of the variance in polymorphism rates around genes in natural populations. We have added extensive simulations and a more thorough explanation of the results regarding Tajima’s D, which provide population genetic evidence that mutation bias is the dominant force shaping genome-wide patterns of sequence evolution around gene bodies. We find that all of the conclusions (low mutation rates in gene bodies and essential genes, etc.) are the same if we consider SNVs and InDels separately, so we present combined results and make this clearer in the text. We show that the reduction in the mutation rate of essential, lethal-effect, and constitutively expressed genes cannot be jointly explained by any hypothesized intrinsic mutation bias. We calculate the minimum L_{segment} necessary for these adaptive biases to evolve and show that this is much smaller than regions enriched for low mutation epigenomic features in essential genes.

We would like to note that while all of the new experiments, analyses, simulations, and theoretical considerations suggested by the reviewers since the original submission have supported the same conclusions - constrained loci experience lower mutation rates -, they have made our case even stronger and thus greatly improved the manuscript. This study has produced an enormous amount of results and data, and we are grateful to the reviewers for their time.

These are the highlights of our revision:

1. New/revised figures: Fig. 1, Fig. 2, Fig. 3, Fig. 4, Ext. Data Fig. 2, Ext. Data Fig. 3, Ext. Data Fig. 4, Ext. Data Fig. 5, Ext. Data Fig. 6, Ext. Data Fig. 7, Ext. Data Fig. 8, Ext. Data Fig. 9, Ext. Data Fig. 10.
2. Added large-scale simulations of population-scale genome evolution using SLiM - analyses of Tajima’s D confirm that mutation bias is the dominant force shaping patterns of sequence diversity around gene bodies (Ext. Data Fig. 6).
3. Added a more thorough explanation of Tajima’s D result to the main text and methods.
4. Added figure showing the density of gene bodies in relation to transcription start and termination sites of focal genes (Fig 1c).
5. Created a figure of aggregated observed mutation rates across all datasets around gene bodies (Fig. 2b).
6. Revised figure 1 to show the distribution of gene bodies, mutations and epigenomic features relative to all transcription start sites and termination sites more clearly.
7. Calculated reduction in gene body mutation rates from raw observed mutation data (Ext. Data Fig 4d).
8. Added scatterplot showing predicted and observed mutation of windows around gene bodies (Ext. Data Fig. 5).
9. Quantified how epigenome-predicted mutation probabilities compare to observed mutation rates around gene bodies (Ext. Data Fig. 5).
10. Added scatterplot of observed mutation density and polymorphisms of windows around gene bodies (Ext. Data Fig. 5).
11. Quantified the predictive power of observed mutation rates and observed levels of polymorphisms around gene bodies (Ext. Data Fig. 5).
12. Calculated mutation rate reduction in essential genes from raw mutation data.
13. Added confidence intervals to TSS/TTS figures (Fig. 2).
14. Calculated and visualized epigenomic enrichment for all features in essential genes, lethal, genes and constitutively expressed genes Fig. 3b, Ext. Data Fig. 8).
15. Added analysis and figures of raw (observed) mutation rates in essential genes, in introns of essential genes, in lethal- vs- non-lethal-effect genes, in introns of lethal- vs. non-lethal-effect genes, in constitutively expressed vs. tissue-specific genes, in introns of constitutively expressed vs tissue-specific genes (Fig. 3c, Ext. Data Fig. 8).
16. Added analysis and figures of raw mutations and evolutionary constraints.

17. Quantified genetic diversity for mutation bias in 8 mutation accumulation founders (Fig. 4b).
18. Rewritten the abstract to convey results more quantitatively.
19. Added specific quantification of minimum L_{segment} necessary for selection to act on mechanisms of hypomutation, and compared the minimum length to the length of regions enriched for epigenomic features associated with low mutation rates (Ext. Data Fig. 10).
20. Increased the number of percentiles from 10 to 50 in plots to show relationship between mutation rate and estimates of evolutionary constraint in more detail (Fig. 4b,c).
21. Changed percentile plots to show raw data rather than predicted values (Fig. 4 b,c).
22. Added a heatmap figure showing associations between epigenomic features and signatures of evolutionary rate and constraint (Fig. 4a).
23. Analyzed SNVs and InDels separately and confirmed that these led to the same conclusions (low mutation rates in gene bodies and essential genes).
24. Provided summary statistics of mutation-epigenome models (Materials and Methods).

Referee #1 (Remarks to the Author):

The authors have dealt with the one general issue I raised in the first review. I remain enthusiastic about this study given its possible impact on evolutionary thinking.

Thank you!

Referee #2 (Remarks to the Author):

Grey et al present a revised version of their now entitled paper “Mutation bias reflects natural selection in *Arabidopsis thaliana*”. The authors made substantial efforts trying to address the many technical and conceptual concerns raised by the four reviewers. I continue to think that it is an interesting paper, but I just don’t think it is quite enough to be published in Nature. Building on my previous comments, my major concern is the lack of sufficient quantification of their predictive model. In my view this is the heart of the paper, and drives all downstream conclusions/interpretations. Most of the results/conclusions currently rely too much on visual comparisons between predicted and observed patterns, rather than on quantitative measures such as “variance explained by the model”. Such model performance statistics should be provided. Without them, it remains impossible for me to judge the relevance/impact of their paper.

We are happy to hear that the reviewer finds our work in principle interesting and appreciates the substantial efforts we have made in the first round of revision. We interpret the reviewer’s statement to mean that they agree with us that our surprising conclusions – provided that they are sufficiently supported by evidence – should be communicated in a venue that reaches the broadest possible audience.

We have quantified how well the predictions of our model conform to observed *de novo* mutations: over 90% ($r^2 > 0.90$) of the variance in the genome-wide distribution of observed mutations around gene bodies is explained by epigenome-predicted mutation probabilities. Mutation bias, in turn, explains over 90% ($r^2 > 0.90$) of the variance in sequence polymorphism levels around genes.

Please note that the models merely play a minor role in the revised manuscript, as the majority of results are now presented in observed mutations. Please note that we had already substantially increased the number of observed mutations in the first revision, which in turn has reduced the need to rely primarily on predictions. That the conclusions have remained the same confirms the predictive power of the original model nevertheless.

The previous version mostly used verbal descriptions of patterns they are so striking (e.g., the distributions of predicted and observed mutation around gene bodies are nearly identical). Nevertheless, we agree that quantification makes our case even more robust, and we now provide the accompanying supportive statistics (e.g., r^2 between predicted and observed mutation rates around gene bodies > 0.90).

Here are my major concerns. These lean on some of my previous points, but also include additional questions in light of the revised version, the comments of the other reviewers, and me having had a bit more time to think about this.

1. The section “Predicting mutation probability across the genome” introduces their predictive model. The variance in observed mutation frequency by their final selected model is nowhere reported. Only the final predictor set along with the associated t-values of the coefficients are shown in Fig. 1. The t values are relatively large and should be highly significant. But as with anything in genomics, the input sample sizes are also large (all the genes), which renders standard errors of the coefficients small. A more meaningful quantity to report is something like the R2 value, which tells us how much of the variance in the mutation frequency patterns is explained by the model.

Also, standard statistical output such as values of the coefficients, t values and associated degrees of freedom, a plot of the distribution of the response variable(s) and perhaps residual diagnostic would be good to see (e.g. residuals versus fits plots). Again, it is still not clear to me how they set up their “generalized linear model”. Do they assume their response variable is normally distributed? Or drawn from another distribution. Making model choices solely based on visual inspection of the predicted mutation probably along genes is not sufficient for me.

In one of their replies, the authors state “Importantly, we would like to emphasize that the significance of our discovery does not derive primarily from the effect size of the observed mutation bias (though some of the new analyses suggest that it might be at least comparable to other non-random evolutionary processes), but rather because it challenges prevailing assumptions about mutations occurring randomly with respect to fitness consequences. We agree that future research is warranted to more fully understand the limits and magnitude of the reported mutation bias”. I strongly disagree with this perspective. The authors have all the results/model fits, etc. at hand. Why not just show the results to allow for a more fair assessment of their finding. Are we talking about predicted mutation biases that are relatively trivial relative to other

evolutionary processes shaping natural variation, or are we looking at major effects that would fundamentally require us to reevaluate previous conclusions about genomic patterns of selection. To me, this difference determines if this ms is published in a high impact journal like Nature, or somewhere else.

Incorporating the reviewer's comments has made the manuscript more impactful, as they have made us also think more about our data. Therefore: thank you!

The models of mutation as a function of epigenomic features are no longer the central result of the manuscript – conclusions are now drawn primarily from the empirically observed *de novo* mutations. Nevertheless, we appreciate the reviewer's suggestions to make these models clearer. We have added additional information to the manuscript with more details and summary statistics (e.g., degrees of freedom, r^2 , p-values for the models). In addition, the concerns about the summary statistics of the models from Fig. 1d, while now addressed, are no longer germane in light of the finding that our results have confirmed with analyses of empirically observed (rather than predicted) mutation rates (e.g., confirmed results shown in Fig. 2, Fig. 3, Fig. 4, Ext. Data. Figs.).

The effects we discovered are astonishingly large, with mutation bias explaining patterns of sequence evolution around genes to an even greater extent than selection does. This is evident by the pattern of natural polymorphism and Tajima's D around gene bodies in wild *Arabidopsis* accessions. We verify in the revised manuscript with extensive forward genome evolution simulations that these results are consistent with mutation bias being the dominant force behind patterns of natural variation around genes (Fig. 2, Ext. Data Fig. 5, Ext. Data Fig. 6). The revised manuscript also provides a more quantitative assessment of these mutation biases. Epigenome-derived mutation probabilities accurately predict mutation rates ($r^2 > 0.90$) around genes and levels of polymorphism in natural populations ($r^2 > 0.90$). We discover a 58% reduction in the mutation rate of gene bodies (compare this with estimates that only 30% of sites in gene bodies are plausibly under selection) and a further 37% reduction in essential genes. Given that a selection coefficient of 1% is generally considered large, these mutation bias effects are certainly not trivial.

In addition, we would like to point out that while the discovered effect sizes are large, even much smaller effect sizes would have required a rethinking of current models of the nature of mutation in evolution.

2. Related to this, all the plots in Fig. 2 showing the observed and predicted mutation frequency are represented in terms of mean values (calculated from all the genes), but show no information about the variance (i.e. variation around the mean pattern). A similar concern lingers for me in Fig. 3. My previous comments and those of another reviewer were critical of the decile plots shown in that figure. These plots convey a strong relationship between predicted mutation probability and various measures of evolutionary constraint. But looking at the correlation values (r) given in that figures, the relationship is actually very weak. In fact, the R^2 (correlation coefficient squared) would even be smaller. I had asked for an alternative representation that plots quantitative values of predicted mutation probability versus constraint measures. This would be a fairer representation and more consistent with the low correlation values. As an alternative I mentioned plotting 95% CI around

the decile means. The authors did the latter. I now realized that the large sample sizes per decile would still lead to relatively small CI. So, after all, I think a quantitative bivariate plot with a regression fit is the fairest way to present the data (bivariate densities could be indicated via color code).

The large sample size benefits our analyses, as it provides strong power to test deviations from the null model. Under the null hypothesis predicted by the traditional theory, r should be equal to 0. We have discovered that this is not the case for any of the relationships we have examined. All analyses show that the empirically observed data significantly deviate from null expectations, which would not change if r or r^2 were presented.

Quantile plots are a standard and widely accepted format. For example, recent and prominent papers published in *Nature* use such plots to visualize bivariate relationships (e.g., Karczewski et al. 2020, cited 1,515 times). To provide more precision and to mitigate concerns about visualizing non-linearities in the data, we have increased the number of quantiles from 10 to 50, to provide a higher resolution visualization of the data. As you will see, all our inferences still hold.

We have updated the figures requested with confidence intervals in the revised manuscript.

The patterns described since the first submission of this work – lower mutation rates in gene bodies and genes with conserved function – have now been confirmed with multiple independent mutation datasets that we have added since the original submission. These include: germline and somatic mutations called in the largest mutation accumulation experiment ever conducted in *Arabidopsis*; surveys of somatic mutation in 64 *Arabidopsis* leaves; and germline mutations discovered in bottlenecked *Arabidopsis* populations. Our new analyses of site frequency spectra

further confirm that the predicted effects of adaptive mutation bias are manifest in natural populations, with detectable effects on population genetic statistics such as the widely used Tajima's D.

3. It also occurred to me that the final selected predictor set for SNV and INDELS is very different, which raises the question if this relates to different biology and/or the fact that INDELS tend to occur in different parts of genes than SNVs so that they relate to different epigenomic features. I don't see any plots to show the observed mutation frequency distribution contrasting SNVs and INDELS. The authors also don't comment on this.

Please see the new comment in the manuscript "...results from all downstream analyses were effectively the same when SNVs and InDels were analyzed separately, with (i.e., both being less frequent in gene bodies and essential genes). Similarly, separate analyses of either germline or somatic *de novo* mutation calls led to the same conclusions. We thus report combined results." The inclusion of separate figures for SNV and InDels (eg. Ext. Data Fig. 4e) for all results would massively expand all figures and, since the analysis is giving similar results, would not provide new information. However, if the editor and reviewer disagree, we would be happy to provide these, as long as greatly increasing the number of Extended Data Figures is acceptable.

4. The authors now included new PC analysis of the epigenetic marks they obtained from the PCSD. It remains unclear why they do this and not just plot the distribution individual marks along the gene (see Fig. 2A), or the chromatin states given in the PDSC database. If the authors think the PC dimensions are important, why don't they use the PC loadings as predictors in the model. The fact that various epigenetic modifications load onto the PC means that they are correlated. This brings me back to the variance inflation factors. The authors have performed VIF analysis (by the way I would not mention this in the main text as it is trivial and not a main result) and found little evidence for this. Why not show the VIFs for each predictor in a SI table so we can see more clearly.

We agree that the PC results were confusing as they were added for illustration purposes only (to remind the reader that epigenomic marks often differ greatly around TSS and TTS). We have removed the VIF comment from the main text as well.

5. The authors bypassed my concern about their epiRIL data analysis by stating that "Because this result is not directly pertinent to the main conclusions of our paper, as the effects of cytosine methylation on mutation via deamination have been previously described, we have removed this extended data figure in order to make room for the new results we have added". Since the main conclusion of their revised paper is the same as their original submission, I do not see how the epiRIL analysis was pertinent before but is no longer. Space limitation in SI should not be an issue. The authors don't comment at all on whether I was wrong. I am happy to be proved wrong, but felt that the removal of their analysis bypasses potential issues in their original interpretation.

We agree with the reviewer that the epiRILs raise an interesting question about the indirect effects of gain/loss of cytosine methylation on genome stability. We removed the section, as a large number of new analyses has been added, and the greatly expanded set of empirically observed *de novo* mutations, which now underpin our main analyses, no longer require indirect validation of our epigenome-derived predictions. In light of our expanded and greatly improved analyses, it has become clear that appropriate treatment of the epiRILs requires its own full study, especially to properly incorporate the TE aspects in this population.

The final version of this paper is a more streamlined and well-focused report, thanks to the comments of all reviewers. Over the course of this project, we have generated an enormous amount of data, written over 15,000 lines of code, created over 300 figures, and essentially tripled the amount of results included in this paper since our original submission.

6. Finally, I appreciate that the authors clarified the differences between "adaptive Mutation Bias" and "Adaptive Mutation bias". In their formulation, the patterns of reduced mutation probabilities in constraint genes increases fitness by avoiding deleterious mutations. In this sense they are "adaptive". Because this mutation bias increases fitness, this bias itself must be an evolving trait (and the authors also mention this in their discussion). There were a few comments from the reviewers how this bias could evolve. I acknowledge this this a deep and difficult question. But, the possibility that it can evolve (or that it is postulated to evolve) would require that this bias is variable in natural populations, as surely the predictor set the authors used will show variation among accessions (e.g. mCG is highly variable among natural accessions). The authors have now included additional MA lines that were established in different genetic background (i.e. derived from

different founder accessions). This new data provides an opportunity to examine the range of natural variation in mutation bias. In fact, going one step further: this new data would also allow the authors to assess a causal link to their predictor set by profiling the founder accessions for at least some of their strongest predictors. Such an analysis would provide some empirical hints as to how this bias does/can evolve, and would thus mitigate any potential philosophic debates and controversies that arises from their conclusions.

Thank you for this excellent suggestion. We have performed additional analyses, which revealed evidence of heritable variation for mutation bias in the mutation accumulation datasets from 8 founders (Ext. Data Fig. 4b). The existence of this heritable variation indeed suggests opportunities for selection to act on mutation bias, and thus the potential for it to evolve in adaptive directions.

Referee #3 (Remarks to the Author):

I thank the authors for addressing my previous comments and for including many new analyses in this version. The manuscript has been strengthened through the revision. However, key issues remain unresolved.

Major comments:

1. Related to previous major comment 1.

(i) I asked the authors to use raw mutation numbers instead of predicted mutation probabilities to study mutation rates in different genes or genomic regions and to compare mutation with polymorphism. They used raw mutations in only one plot (lethal vs. non-lethal genes; Extended data Fig. 2c). But in all other plots, they still used predicted mutational probabilities. It appears that there are sufficient mutations in the data to allow using raw mutations. For instance, in 10 bins in Fig. 3, each bin would have hundreds of mutations. In Fig. 2b, for a 100-bp window, there should be >200 mutations. They now acquired a new mutation accumulation dataset (Fig. 2cd) that is larger than the primary dataset used (the exact mutation number in the new dataset is not provided in the manuscript). They could combine the two datasets to get even more mutations.

All results are now presented in terms of empirically observed *de novo* mutations.

(ii) I complained that they did not tell us how well the model fit the data, and they still do not provide such statistics. It is interesting to note from the raw mutation numbers in Fig. 1d that SNV rate is not lower in coding regions than non-coding regions. Even for indels, the mutation rate is only slightly lower in coding regions than non-coding regions (Fig. 1d). This pattern, based on raw mutation numbers, gives a very different picture than the one based on predicted mutation probabilities (Fig. 2b). That is why it is important to use raw mutation numbers.

We have corrected this figure. The bias in both SNVs and InDels is now clear. Both SNVs and InDels are lower in gene bodies, essential genes, etc. We show combined results for brevity and clarity of the manuscript.

(iii) Regarding the model, the Methods section states that “The response variable was the untransformed (to avoid risk of increased false positives caused by transformation) detected mutation rate across every genic feature (upstream, UTR, coding, intron, downstream).” Does this mean that they used different models for different genic features (upstream, UTR, coding, etc)? What is the rationale of such modeling? Do they assume that the same epigenetic marker has different effects in different genic features and why? What result will they get if they use the same model for all genic features?

We agree that this was unclear in the manuscript. “(upstream, UTR, coding, intron, downstream)” simply refers to which regions we are considering as genic. We did not use different models for each feature. We have clarified this in the text.

2. Related to previous major comment 2.

(i) Regarding the interpretation of their observation, authors state that their adaptive model is compatible with population genetic theory if $L_{segment}$ is large, but do not provide concrete evidence that $L_{segment}$ is indeed larger than that required by the theory. I suggest that they calculate, based on mutation rate and other parameters, the theoretically minimal $L_{segment}$ for the adaptive hypothesis to work, and compare it with the true $L_{segment}$. Although the true value of $L_{segment}$ is probably unknown, I believe one can estimate its order of magnitude.

Thank you for this comment, which we believe has considerably strengthened the manuscript. We have added the following:

Selection on intragenomic mutation rate variation will be effective (Lynch 2010; Hodgkinson and Eyre-Walker 2011) when

$$N_e * u * s * du * pd * L_{segment} > 1 \quad \text{eq.1}$$

where N_e is the effective population size, u is the mutation rate, s is the average selection coefficient on deleterious mutations, du is the degree of change in mutation rate, pd is the proportion of sites subject to purifying selection, and $L_{segment}$ is the region of the genome affected. Assuming an effective population size of ~300,000 (Cao et al. 2011; Gossmann et al. 2010; Moore and Purugganan 2003), a mutation rate of ~10⁻⁸ (Ossowski et al. 2010), an average

selection coefficient of 0.01 (Hodgkinson and Eyre-Walker 2011), an order-of-magnitude reduction in mutation rate (Hodgkinson and Eyre-Walker 2011), and functionally constrained regions where 20% of sites are under selection (Hodgkinson and Eyre-Walker 2011), the total length of the sequence affected, $L_{segment}$, would have to be at least ~200 Kb, which (accounting for differences in effective population size) is similar to previous estimates in humans (Hodgkinson and Eyre-Walker 2011). For perspective, this minimum $L_{segment}$ is considerably less (~1.5%) than the sum of coding regions with elevated levels of H3K4me1 (top quartile, ~13 Mb, or 15% of the genome), a feature we have shown to be enriched in gene bodies and essential genes and associated with lower mutation rates. Thus selection is expected to act with high efficiency on variants that cause DNA repair and protection mechanisms to preferentially target such regions. Please see Ext. Data Fig. 10 for more details.

(ii) Authors seem to agree that they cannot reject the intrinsic property hypothesis but is reluctant to discuss it because they felt that it would be “even more controversial” than the adaptive hypothesis and may be misused by non-scientists. In my view, drawing conclusions without considering/discussing alternative explanations is more dangerous.

We apologize for having not been fully clear on this point. We have considered and discussed among the authors such intrinsic properties from the onset of this study. After examining a large body of evidence, we believe that the intrinsic property interpretation will be more controversial because there is less evidence for it. Mutation bias, which is influenced by natural selection to favor interactions between DNA repair and features enriched in constrained regions, provides in our view a more parsimonious and complete explanation of the observed patterns, whereas an intrinsic property hypothesis cannot explain the full spectrum of our results (e.g., essential genes do not have higher expression, and do have greater gene body methylation). We mention this latter fact in the text. However, if this reviewer feels strongly that the alternative hypothesis, even if seems to provide a much less likely explanation for our results, must be discussed more extensively, we would of course be happy to add a few more sentences to the final paper.

3. This is a new major comment arising from their new analyses.

I am very confused by their new results on Tajima's D and minor allele frequencies. As far as I know, a change in minor allele frequencies does not indicate a change in mutation rate but a change in selection. So, the finding that genic regions have higher minor allele frequencies than non-genic regions does not tell us anything about mutation rate difference between these regions, but suggests that coding regions are subject to weaker purifying selection than non-coding regions, which is unexpected and deserves an explanation. Is this result consistent with previous population genomic analysis of the species? Is the same result obtained when raw mutations are used? Similarly, a less negative Tajima's D of genic regions than non-genic regions suggests a weaker purifying selection acting on genic than non-genic regions.

We have improved our explanation (main text, materials and methods, Fig. 2, Ext. Data Fig. 5, 9) of these results in the revised manuscript. The effect of mutation rate in non-equilibrium populations on the site frequency spectrum has been established in population genetics for at least 30 years (Tajima 1989; Morales-Arce et al. 2020). Sequences with high mutation rates are enriched for new and thus rare variants. Therefore regions with high mutation rates have many polymorphisms and more negative Tajima's D (the observed pattern in intergenic space in wild *Arabidopsis* accessions), while regions with low mutation rates have fewer polymorphisms and less negative Tajima's D (again, the observed pattern in gene bodies in wild *Arabidopsis* accessions).

We have also added a large set of forward analyses of population-scale genome evolution in SLiM. These confirm our theoretical predictions and show that reduced mutation rate in gene bodies fits our data.

4. This is a new major comment arising from their newly presented results.

Because there is no variation in SNV mutation rate among genic features (Fig. 1d), are all the mutation rate differences in Figs. 2, 3, and 4 actually due entirely to indel mutations? If so, this should definitely be stressed because indel and point mutations are likely caused by different molecular mechanisms. Do polymorphisms in Figs. 2, 3, and 4 include both SNPs and indels? Also, at the end of page 3 and beginning of page 4, authors discussed that their results from modelling are consistent with previous findings. Are they consistent with previous findings on both SNVs and indels? If I am not mistaken, previous findings are almost exclusively on SNVs. Please clarify.

We have corrected this figure. The bias in both SNVs and InDels is now clear. Both SNVs and InDels are lower in gene bodies, essential genes, etc. We show combined results for brevity and clarity of the manuscript.

Minor comments

1. Previous minor comment 2. The pattern is still opposite to what was claimed in the manuscript. It shows that gene-poor regions (white parts in chromosomes) have lower mutation rates than gene-rich regions (black parts in chromosomes).

Shown are only the mutations that we consider in our analyses of genic regions (that is, within 1 kb of genes); therefore, many observed mutations in gene-poor regions are not shown, because they are not used in our analyses. The rationale is that this paper tests hypotheses about the distribution of mutations within, around, and between genes. The legend states: “*De novo* mutations detected in genic regions (genes \pm 1,000 bp)”.

2. Previous minor comment 4. Authors did not answer how different the 10 deciles in Fig. 3 are in mutation rate (in absolute terms).

The absolute numbers are arbitrary because we combine multiple datasets, including somatic and germline mutations. This large, combined set of *de novo* mutations is one of the strengths of the investigation as it allows us to characterize mutation distributions at an unprecedented scale.

3. New minor comment. “We found a 70% reduction in synonymous variants in the raw *de novo* mutation dataset relative to random expectations”. This sentence on page 4 appears contradictory to Fig. 1d, which shows virtually equal SNV mutation rates across genic features. Is the lowered synonymous rate compensated by an elevated nonsynonymous rate?

Please see answer to major comment #3 above.

Referee #4 (Remarks to the Author):

In this revised version of the manuscript, the authors have made a lot of efforts to answer my comments and I greatly appreciated the effort and the positive tone of the answer.

I was nevertheless not fully satisfied with one part of the extra data presented in the extended material: the simulations made to look at the effect of deleterious mutations. The model was too simple and assume very large effect filtered at once. Because strong bottlenecks are imposed, what matters here is more the potential within host selection from gamete to gamete.

Let us assume that mutations within a given gene occur at rate μ and have a deleterious effect at the heterozygous state such that they hamper the reproduction rate of the affected cells through the development process by $(1-s)$. If there are N cellular generations within a plant generation, we have $\mu + \mu(1-s) + \mu(1-s)^2 + \dots + \mu(1-s)^{(N-1)}$ mutated cells because some cells have mutated at the last division and have not been counter selected, some have mutated at the generation before and have a $1-s$ probability of being a gamete, some two generations before and therefore have been counter selected for two cycles $(1-s)^2$ and so on. This gives $\mu(1-(1-s)^N)/s$ and if we compare to the number of neutral mutations we would expect μN ($s=0$) the counter selection of mutations occurring within the gene that have effect s is $(1-(1-s)^N)/(Ns)$.

If we assume 30 cellular generations between plant generations and a 1% fitness effect we expect a 15% underrepresentation, if the cost is of 2% it is a 25% reduction.

I have no clear idea of the values of N and s , but I think it is quite important to substantiate the modelling part in that respect. The argument that a 30% loss of mutation is unrealistic, is not that much unrealistic if we take into account the cumulative counter selection over the gamete to gametes development that can turn a realistic small effect into a quite large counter selection.

The outcome of this process could appear through the synonymous, non synonymous pattern, but may leave a marginal signal in term of significance, that may also be due to partially wrong mutational biases and other noise factors.

Interestingly if N is quite large and some s are important, this cellular process will also lead to a decreased mutation rate in important genes from generation to generation, and may also have some evolutionary implications that are worth mentioning.

Thank you, these are really interesting ideas. We have updated the manuscript to reflect this possibility. We now clarify that our simulation examines mutations removed by purifying selection, whether via heterozygous lethality or via somatic competition. Our results are thus robust to the possibility of strong somatic selection caused by intercellular competition. In our simulated selection, we assess the effects of 30% of gene body variants being lost due to purifying selection, which is the approximate upper limit of estimates of the percent of sites in gene bodies under any functional constraint (Haudry et al. 2013). The effect is still less than the observed mutation bias we discovered, consistent with our quantification of a 58% reduction in gene body mutations.

Olivier Tenaillon

Reviewer Reports on the Second Revision:

Referee #2 (Remarks to the Author):

The authors have (again) made substantial revisions to their manuscript. I would like to thank them for their efforts in responding to my many concerns. They now present more quantitative arguments throughout, which I find very convincing (especially since they confirm their initial conclusions).

In my view, the manuscript has improved substantially since the initial submission. The large amount of data, the diversity of analyses and data modalities, along with the thought provoking insights should make this work attractive to the wider readership of Nature.

Referee #3 (Remarks to the Author):

This version represents a significant improvement over the previous one, and I am largely satisfied. Below are some minor suggestions mainly about the presentation.

1. Fig. 1c, top panel: what do different shades of grey mean?
2. Tajima's D . Mutation rate is a scaler of Tajima's D . That is, when D is negative, a higher mutation rate makes D more negative, but when D is positive, a higher mutation rate makes D more positive. So, a general statement that a higher mutation rate makes D more negative is incorrect. Please correct this in main text and Methods.
3. Extended Data Fig. 10. The response to my comment regarding the new Extended Data Fig. 10 is good and should be included in either the legend to Extended Data Fig. 10 or Methods; otherwise, Extended Data Fig. 10 is hard to understand.
4. Finally, do the author think that their finding is specific to *A. thaliana* or is general, and why?

Referee #4 (Remarks to the Author):

The authors have responded to my comments appropriately.

Author Rebuttals to Second Revision:

Referee #3 (Remarks to the Author):

This version represents a significant improvement over the previous one, and I am largely satisfied. Below are some minor suggestions mainly about the presentation.

Thank you for all of the excellent suggestions and feedback throughout the review process. Your input has improved the quality of this work significantly!

1. Fig. 1c, top panel: what do different shades of grey mean?

Shading indicates the frequency of genomic regions that are at a specific distance (x-axis) from TSS or TTS annotated as inside a gene body. The lighter shaded regions, at TSS+3000 bp, for example, reflect that not all genes are ≥ 3000 bp in length. We have added clarification to the figure legend.

2. Tajima's D . Mutation rate is a scaler of Tajima's D . That is, when D is negative, a higher mutation rate makes D more negative, but when D is positive, a higher mutation rate makes D more positive. So, a general statement that a higher mutation rate makes D more negative is incorrect. Please correct this in main text and Methods.

We have added to the main text and methods statements to this effect, that the mutation rate is a scalar of Tajima's D.

3. Extended Data Fig. 10. The response to my comment regarding the new Extended Data Fig. 10 is good and should be included in either the legend to Extended Data Fig. 10 or Methods; otherwise, Extended Data Fig. 10 is hard to understand.

Thanks! We have added this comment to the legend of Extended Data Fig. 10.

4. Finally, do the author think that their finding is specific to *A. thaliana* or is general, and why?

Great question! The *Populus trichocarpa* Tajima's D patterns already suggest yes. We are now looking into this more fully, and initial tantalizing observations suggest that it is indeed not limited to *Arabidopsis*, but also that differences between species exist. In the meantime, to acknowledge that the current study is largely limited to *Arabidopsis* we have added "in *Arabidopsis*" to the conclusion of the abstract and "it will be important to test the degree and extent of mutation bias beyond *Arabidopsis*" to the conclusions of the paper.